# Comprehensive assessment of mRNA isoform detection methods for long-read sequencing data

Yaqi Su[1,2,9], Zhejian Yu[1,2], Siqian Jin[1,2], Zhipeng Ai[3], Ruihong Yuan[2], Xinyi Chen[1,2], Ziwei Xue[1,2], Yixin Guo[1,2], Di Chen[4,5], Hongqing Liang [3], Zuozhu Liu[6] & Wanlu Liu [1,2,7,8] ✉

The advancement of Long-Read Sequencing (LRS) techniques has significantly increased the length of sequencing to several kilobases, thereby facilitating the identification of alternative splicing events and isoform expressions. Recently, numerous computational tools for isoform detection using long-read sequencing data have been developed. Nevertheless, there remains a deficiency in comparative studies that systemically evaluate the performance of these tools, which are implemented with different algorithms, under various simulations that encompass potential influencing factors. In this study, we conducted a benchmark analysis of thirteen methods implemented in nine tools capable of identifying isoform structures from long-read RNA-seq data. We evaluated their performances using simulated data, which represented diverse sequencing platforms generated by an in-house simulator, RNA sequins (sequencing spike-ins) data, as well as experimental data. Our findings demonstrate IsoQuant as a highly effective tool for isoform detection with LRS, with Bambu and StringTie2 also exhibiting strong performance. These results offer valuable guidance for future research on alternative splicing analysis and the ongoing improvement of tools for isoform detection using LRS data.

Alternative splicing (AS) is a post-transcriptional regulation mechanism that splices a single kind of pre-mRNA into multiple distinct mature mRNAs, referred to as isoforms. The vast diversity of alternative splicing events significantly contributes to the complexity of the transcriptome and proteome. AS is prevalent in vertebrates, with an estimated 90% of human genes undergoing alternative splicing[1,2]. Moreover, AS has been observed in invertebrate, fungal, and plant genomes[3]. Accumulated evidence suggests AS plays a crucial role in

[1]Department of Orthopedic Surgery of the Second Affiliated Hospital, Zhejiang University School of Medicine, Zhejiang University, Hangzhou 310009 Zhejiang, China. [2]Centre of Biomedical Systems and Informatics of Zhejiang University-University of Edinburgh Institute (ZJU-UoE Institute), International Campus, Zhejiang University, Haining 314400 Zhejiang, China. [3]Division of Human Reproduction and Developmental Genetics, Women's Hospital, Zhejiang University School of Medicine, Zhejiang University, Hangzhou 310006 Zhejiang, China. [4]Center for Reproductive Medicine of the Second Affiliated Hospital Zhejiang University School of Medicine, Zhejiang University, Hangzhou 310009 Zhejiang, China. [5]Centre for Regeneration and Cell Therapy of Zhejiang University-University of Edinburgh Institute (ZJU-UoE Institute), International Campus, Zhejiang University, Haining 314400 Zhejiang, China. [6]Zhejiang University-Angel Align Inc. R&D Center for Intelligent Healthcare, Zhejiang University-University of Illinois at Urbana-Champaign Institute (ZJU-UIUC Institute), International Campus, Zhejiang University, Haining 314400 Zhejiang, China. [7]Future Health Laboratory, Innovation Center of Yangtze River Delta, Zhejiang University, Jiaxing 314100, China. [8]Alibaba-Zhejiang University Joint Research Center of Future Digital Healthcare, Zhejiang University, Hangzhou 310058 Zhejiang, China. [9]Present address: Department of Molecular and Cell Biology, University of California, Berkeley, CA 94720, USA. ✉e-mail: wanluliu@intl.zju.edu.cn

various biological processes, including cellular differentiation and organismal development, while its dysregulation has been implicated in numerous diseases, including cancer and neurological disorders[4,5].

The rapid development of next-generation sequencing (NGS) has revolutionized genome-wide investigations of alternative splicing and isoform characterization. However, AS detection using NGS-based RNA-seq relies on recovering splice-junction sites from short NGS reads. Consequently, it fails to provide the global picture of full-length transcripts, impeding the study of the functional consequences of AS. To overcome this limitation, Long-Read sequencing (LRS) technologies, such as those developed by Pacific Biosciences (PacBio) and Oxford Nanopore Technologies (ONT), have emerged[6,7]. PacBio platform employs Zero-Mode Waveguide (ZMW)-based single-molecule real-time (SMRT) technology, whereas ONT utilizes nanopores inserted in an electrically resistant membrane[6,7]. Both platforms enable profiling of full-length RNA transcripts through cDNA (complementary DNA) sequencing, while ONT also allows direct sequencing of native RNA[8–12]. PacBio sequencing includes two modes: (1) continuous long read (CLR), which yields reads longer than 30 kilobases (kb) but with a higher error rate (8-15%); (2) High-fidelity (HiFi) sequence reads data type, a circular consensus sequencing (CCS) that generates the highly accurate (>99%) HiFi reads with read lengths of 10 ~ 30 kb[12]. The progression of LRS technologies has markedly enhanced read length, thereby enabling the accurate delineation of isoform structures. Such technological advancements open new avenues for comprehensive analysis of transcriptome complexity and functional implications of AS.

As the advancement of long-read RNA-seq techniques continues, several computational tools have been developed for AS isoform detection. Nine cutting-edge tools, including StringTie2, FLAIR, FLAMES, Freddie, TALON, UNAGI, TAMA, Bambu, and IsoQuant, have been published[13–21] (Table 1 and Supplementary Table 1). These algorithms can be broadly classified into two categories: guided vs. unguided, depending on whether they require a reference annotation to guide the isoform identification. Among the nine tools, TALON and FLAMES are guided while Freddie, TAMA, and UNAGI are not. StringTie2, FLAIR, IsoQuant, and Bambu incorporate both guided and unguided modules into their algorithms. To ensure high-confidence isoform-calling, different software implements diverse filtering strategies. For instance, StringTie2 constructs an alternative splice graph for each gene locus and employs a maximum flow algorithm over de novo assembled super-reads. Bambu utilizes a machine-learning model for transcript discovery and enables context-aware quantification. IsoQuant builds an intron graph and utilizes an inexact intron-chain matching algorithm to call isoforms. FLAIR (guided) and FLAMES implement a step of re-aligning the raw reads to the initially assembled transcriptome, derived from splice site collapse, to filter out potential false positive events based on the abundance of support reads. TALON first labels reads with internal priming events, classifies each read as known and novel based on the given reference annotation, and then filters out novel reads with an abundance below a specific threshold. TAMA filters out the low-confidence splice junctions based on their ranking or the amount of mapping mismatch surrounding them. Freddie employs a split-segment-cluster strategy for the identification of isoforms, whereas UNAGI classifies multiple transcriptional boundaries and filters out splicing events that fall below a predetermined threshold of locus coverage. Additionally, the majority of methods perform sequencing error correction before the identification of isoforms, with the exception of FLAIR (unguided) and TALON.

With the increasing number of methods for detecting isoforms from LRS data, conducting comprehensive benchmark experiments is crucial to evaluate the applicability of different tools under various conditions. However, due to the lack of a ground-truth annotation, it is challenging to systemically analyze the accuracy of all nine methods across different scenarios. These analyses ought to incorporate a

**Table 1 | A brief overview of the software included for benchmarking**

| Software | Programming language | Version | Reference | Guidance required | Error correction | Functions |
|---|---|---|---|---|---|---|
| FLAMES | Python, C, C++ | 1.0 | Tian et al.[15] | Yes | Yes | Isoform detection from both bulk and single-cell long-read RNA-seq data. |
| FLAIR | Python | 1.5.0 | Tang et al.[14] | Yes/No | Yes/No | Can also perform alignment, quantification, differential expression, and differential splicing analysis. |
| Freddie | Python | 0.3.1 | Orabi et al.[16] | No | Yes | Provide a GTF with detected isoforms. |
| StringTie2 | C++ | 2.2.1 | Kovaka et al.[13] | Yes/No | Yes | Provide identified isoforms with abundance in the GTF format. Able to merge multiple sets of transcripts into a non-redundant set. |
| Bambu | R | 3.0.8 | Chen et al.[20] | Yes/No | Yes | For guided mode it provides a GTF extended with the novel discovered isoforms based on the reference annotation; for unguided mode, the output includes solely constructed transcripts. Both modes can output isoform quantification results. |
| TALON | Python | 5.0 | Wyman et al.[17] | Yes | No | Output a GTF containing identified isoforms and a TSV file of the abundance information. |
| TAMA | Python | b0.0.0 | Kuo et al.[19] | No | Yes | Can also merge different sets of transcripts and perform open reading frame/nonsense-mediated decay prediction. |
| UNAGI | C, C++ | 1.0.1 | Al Kadi et al.[18] | No | Yes | Output the detected isoforms in BED format. |
| IsoQuant | Python | 3.3.1 | Prjibelski et al.[21] | Yes/No | Yes | Can also perform alignment and quantification. |

consideration of multiple standards, including sensitivity and accuracy across datasets generated by disparate sequencing platforms, as well as computational efficiency for datasets of varying sizes. Consequently, there is a pressing need for a comprehensive assessment of existing isoform detection methods. Such evaluation would not only assist users in selecting the appropriate tools but also provide valuable guidance to bioinformaticians aimed at enhancing existing methods or developing novel approaches for isoform detection.

RNA sequins are synthetic RNA molecules utilized as internal controls in RNA-seq experiments[22]. These sequins exhibit intricate splicing events, making them ideal for benchmarking isoform detection software. Using ONT long-read RNA-seq technology along with synthetic, spliced, spike-in sequins RNAs[22], previous studies have compared the performance of several isoform detection software[23–25]. In this study, we systematically compared nine isoform detection tools, which correspond to a total of thirteen methods, using simulated, sequins, and experimental long-read RNA-seq data. These tools were selected based on their comprehensively detailed package documentation and the availability of published manuscripts. To generate Nanopore or PacBio long-read RNA-seq data for analyzing the accuracy of various methods, we have developed a long-read bulk RNA-seq simulation framework called YASIM (Yet Another SIMulator). YASIM operates at two levels: at the upper level, it employs statistical models to enable the simulation of novel AS events and realistic gene expression profiles; at the lower level, YASIM can simulate different sequencer models by utilizing Low-Level Read generators (LLRGs), which are DNA-seq simulators that introduce machine errors[26–29]. With these features, YASIM enables the simulation of datasets with user-defined read depths, novel AS events, sequencing error rates, read completeness, and various profiles of sequencing error models. We used the simulated long-read RNA-seq datasets to systematically evaluate the performance of 13 isoform detection methods. Additionally, we assessed the software performance using sequins long-read RNA-seq datasets, as well as experimental datasets from various species and cell types collected from previously published data. We also generated Nanopore long-read RNA-seq datasets from naïve and primed human embryonic stem cells (hESCs) in this study. With this paired naïve and primed hESCs long-read RNA-seq dataset, along with our previously published NGS RNA-seq datasets derived from the same conditions, we performed comprehensive downstream differential isoform usage (DIU) evaluation and experimentally validated one of the DIU isoforms from *RPL39L* (Ribosomal Protein L39 Like) gene using RT-qPCR (Reverse Transcription Quantitative Real-Time PCR)[30]. Furthermore, we assessed the computational performance of the software by evaluating time and memory consumption using an in-house developed profiler.

In conclusion, our results suggest that IsoQuant achieves the best performance for AS detection in long-read RNA-seq data, excelling in both precision and sensitivity. Additionally, Bambu and StringTie2 demonstrate commendable performance in these metrics. StringTie2, in particular, is distinguished by its superior computational efficiency. FLAIR, especially its guided mode, along with FLAMES, also merits recognition for its robust performance, augmented by comprehensive functional modules. These modules include upstream read alignment and downstream differential splicing/expression analysis within FLAIR, as well as applications to single-cell analysis in FLAMES. Our comprehensive analysis demonstrates the efficacy of prevalent long-read isoform detection methods and guides ongoing research in the AS field. Furthermore, it highlights the need for continuous refinement of isoform detection tools tailored to long-read RNA-seq datasets.

## Results

### An overview of the benchmark study

The overall workflow of the benchmark process is illustrated in Fig. 1. To address the challenge of lacking a ground truth reference for

evaluating the performance of different methods using experimental long-read RNA-seq data, we developed YASIM. YASIM facilitates the generation of long-read RNA-seq reads with novel AS events. It was specifically designed to support comprehensive benchmark by allowing users to specify parameters, including read depth, transcriptome complexity index representing number of isoforms per gene, sequencing read completeness, sequencing error rates, and reference annotation completeness. YASIM allows the generation of long-read RNA-seq datasets on different Nanopore and PacBio platforms (Supplementary Fig. 1).

In the data preparation stage, we utilized YASIM to generate simulated long-read RNA-seq raw data under various conditions, using the *Caenorhabditis elegans* genome as a reference. This approach allowed us to assess the precision and sensitivity of different methods. Additionally, publicly available sequins and experimental long-read RNA-seq datasets from various species were collected for validation. To enable direct comparison in downstream analysis, long-read RNA-seq data from naïve and primed hESCs were generated using the Nanopore cDNA-seq strategy.

Both the simulated, sequins, and experimental data were then aligned to the reference genome using minimap2[31] with platform-specific parameters. Isoform detection was then performed using different methods. In the comparative analysis, the results obtained from simulated data were compared against the ground truth. Precision and sensitivity values were calculated using GffCompare[32] to quantitatively evaluate the accuracy of different methods. For sequins and experimental data, the detected isoforms from each method were compared against the known reference annotation and classified into different categories of isoforms using SQANTI3[33]. Furthermore, a similarity analysis was conducted by comparing the isoforms detected by different methods using the Jaccard algorithm implemented in BEDTools[34].

Additionally, downstream differential isoform usage (DIU) analysis was performed using both simulated and experimental data to assess the accuracy and consistency of different methods. Finally, we developed an in-house profiler to evaluate the computational performance, including the time and memory requirements, of each method across varying scales of datasets.

### Generation of simulated data with YASIM

To comprehensively assess the software performance, we conducted a benchmark analysis under various influencing factors. Specifically, we generated five simulated scenarios, each representing different levels of sequencing depth, transcriptome complexity, sequencing read completeness, sequencing error rate, and reference annotation completeness. Our study encompassed six error models representing both Nanopore and PacBio sequencing technologies: Nanopore R103, reflecting the R10.3 sequencing chemistry released by Nanopore in 2020; Nanopore R94, representing the R9.4 Nanopore chemistry developed in 2016; PacBio Sequel and PacBio RSII, capturing the error profiles of the PacBio Sequel System released in 2015 and the PacBio RS II System developed in 2013; and PacBio CCS and CLR, representing the two distinct types of PacBio sequencing reads. To generate the simulated data representing these technologies, we utilized the LLRG, PBSIM2, and PBSIM3, driven by the YASIM framework[27,28].

For the generation of simulated data used for isoform detection, we generated three replicates of simulated datasets with the six error models under the five different simulation scenarios. We varied the sequencing depth for expressed isoforms, generating depths of 10X, 25X, 40X, 55X, and 70X. Quality control analysis of the sequencing depth for simulated datasets indicates that the majority of the data achieved the anticipated depth across all six error models (Supplementary Fig. 2A). For isoform per gene, we generated simulated data with different targeted transcriptome complexity indexes which positively correlated with the average isoforms per gene

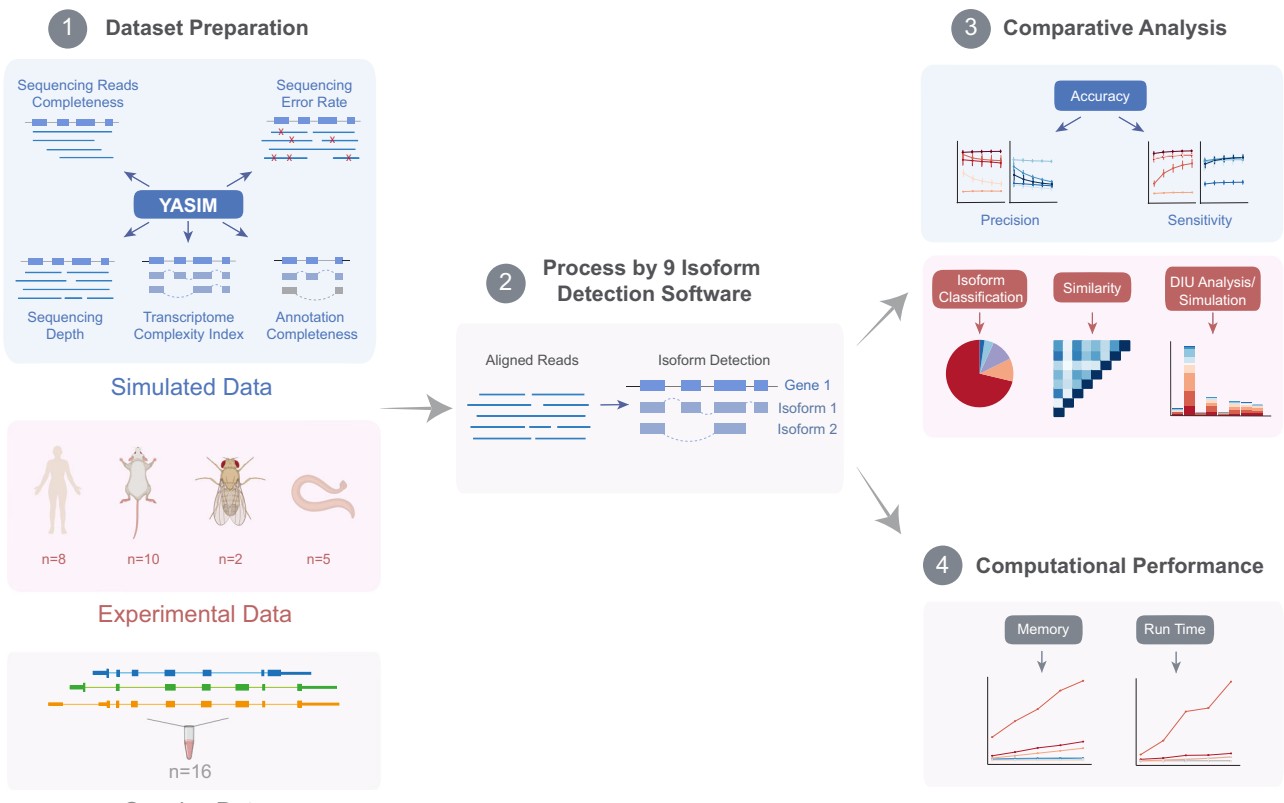

**Fig. 1 | Schematic workflow of the benchmark study (created with BioRender.com).** This figure illustrates the process undertaken for the benchmark study. Simulated datasets were prepared using YASIM, which simulated long-read RNA-seq datasets incorporating variations in sequencing depth, transcriptome complexity index, read completeness, sequencing error rates, and the completeness of reference annotation. Experimental datasets were sourced from publicly available long-read RNA-seq datasets for four species: *Homo sapiens*, *Mus musculus*, *Drosophila melanogaster*, and *Caenorhabditis elegans*. Sequins data were acquired from publicly available long-read RNA-seq datasets that had been spiked with sequins. Additionally, in-house long-read RNA-seq datasets were generated from human embryonic stem cells under both Naïve and Primed conditions. The performance of the software was evaluated from multiple perspectives, including the accuracy of isoform identification, classification of identified isoforms, pairwise similarity between results, and downstream analysis focusing on differential isoform usage (DIU). Computational resource consumption by each method was also analyzed.

(Supplementary Fig. 2B). We simulated reads with different levels of read completeness, including full length, 10% or 20% truncation from both 3′ and 5′ end, 20% or 40% truncation of 3′ end, and 20% or 40% truncation from 5′ end. The read completeness distribution for the simulated data under six error models displayed expected patterns (Supplementary Fig. 2C). Furthermore, we generated simulated long-read RNA-seq datasets characterized by diverse sequencing error rates (0%, 5%, 10%, 15%, 20%), and varying degrees of reference annotation completeness (20%, 40%, 60%, 80%), generating datasets that exhibit the anticipated attributes (Supplementary Fig. 2D, E). The read length within the simulated datasets exhibited consistency across various simulation conditions, except for datasets featuring truncated read completeness (Supplementary Fig. 3).

**Analyzing isoform detection accuracy using simulated data**
We benchmarked a total of thirteen modes from the above-mentioned nine tools, including those guided by a reference annotation (IsoQuant (guided), StringTie2 (guided), FLAIR (guided), FLAMES3, FLAMES10, TALON, and Bambu (guided)) and those independent of guidance (IsoQuant (unguided), StringTie2 (unguided), FLAIR (unguided), TAMA, UNAGI, Bambu (unguided), and Freddie). Software performance was assessed using default parameters to reflect typical use cases for users without specialized training[35]. Since the bulk RNA-seq module of FLAMES does not provide a default set of parameters, we adopted the configuration file used for running a provided test dataset and modified the threshold of support reads from 10 to 3, which aligns with the default threshold of a similar parameter in FLAIR. The results obtained with the number of supporting reads set to 3 or 10 were both included in our results and were denoted as "FLAMES3" and "FLAMES10", respectively.

When testing the performance of the software with variations in sequencing depth, IsoQuant (guided) and Bambu (guided) consistently achieved the highest precision, while TAMA and Freddie displayed the lowest precision across different depths (Fig. 2A and Supplementary Fig. 4A). Certain methods, such as FLAIR (unguided), TALON, UNAGI, and TAMA, showed a dramatic decrease in precision as sequencing depth increased, particularly on Nanopore and PacBio CLR datasets (Fig. 2A and Supplementary Fig. 4A). For PacBio CCS datasets, it appeared that the precision of the methods was not influenced by the changes in read depths, except for TALON and TAMA, which exhibited a decrease in precision as sequencing depth increased (Fig. 2A and Supplementary Fig. 4A). Regarding sensitivity, all methods demonstrated an overall improvement in sensitivity with increasing read depth across different sequencing platforms (Fig. 2B and Supplementary Fig. 4B). IsoQuant (guided) exhibited the highest sensitivity on the Nanopore and PacBio CLR datasets, while TAMA showed the highest sensitivity on PacBio CCS datasets (Fig. 2B and Supplementary Fig. 4B). Freddie and FLAMES10 exhibited the lowest sensitivity in all datasets (Fig. 2B and Supplementary Fig. 4B).

Next, we assessed the influence of variations in the number of isoforms, represented by the transcriptome complexity index (Supplementary Fig. 2B). Across all Nanopore and PacBio platforms, at

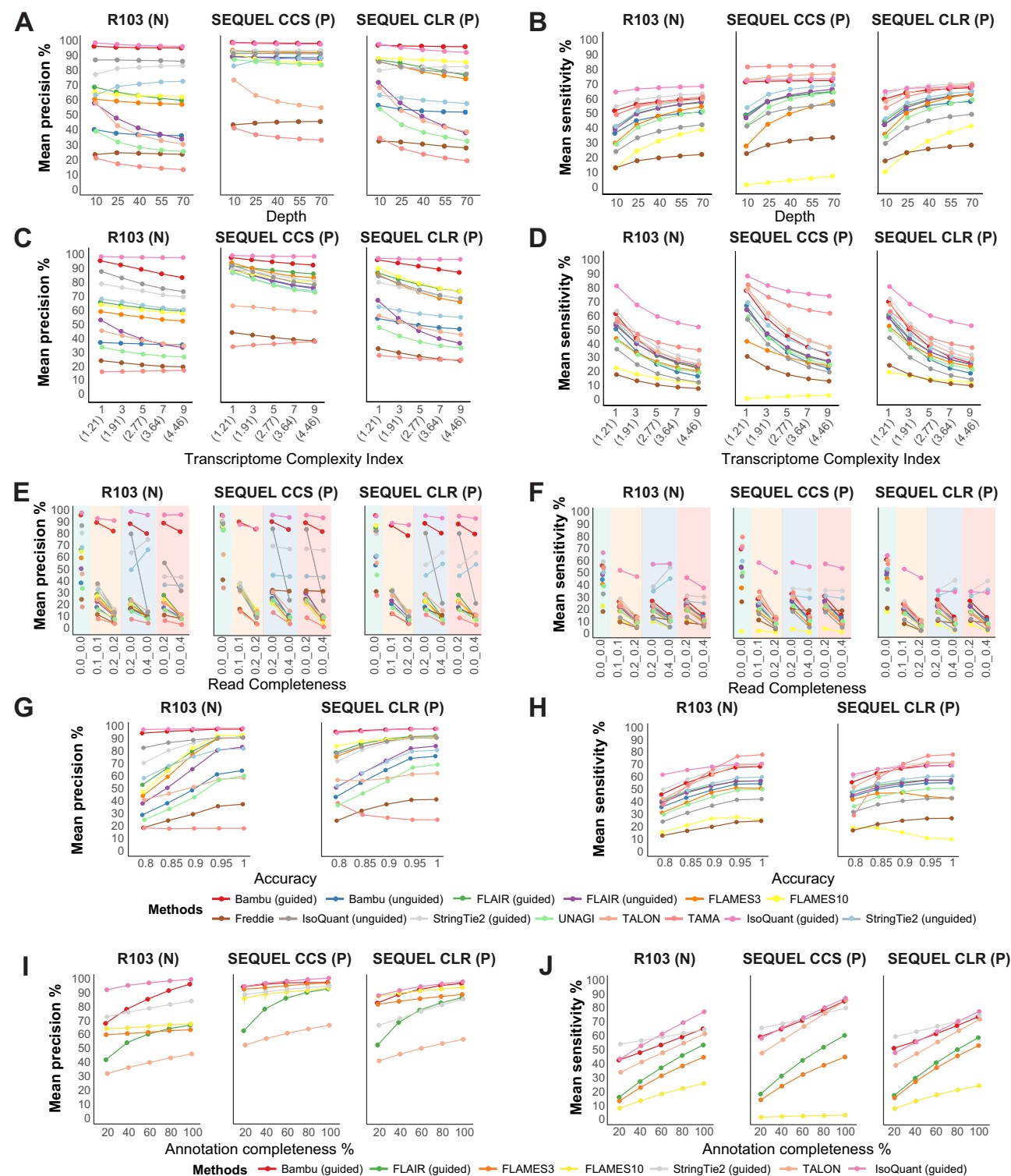

different levels of transcriptome complexity, IsoQuant (guided) and Bambu (guided) consistently achieved the highest precision. In contrast, TAMA and Freddie displayed the lowest precision. IsoQuant (guided) also exhibited the highest level of sensitivity across all sequencing platforms, except that TAMA showed the highest sensitivity in PacBio RSII CCS data. FLAMES10 and Freddie demonstrated the lowest sensitivity in all scenarios (Fig. 2C, D and Supplementary Fig. 4C, D). It is noteworthy that, across the majority of methodologies, an increase in the number of isoforms correlated with a likelihood of generating false positives. Notable exceptions to this trend included IsoQuant (guided), TAMA, and Bambu (unguided) for Nanopore

datasets, IsoQuant (guided) and TAMA for PacBio CCS datasets, and IsoQuant (guided) for PacBio CLR datasets (Fig. 2C and Supplementary Fig. 4C). The sensitivity of all methods, except for FLAMES10 in PacBio CCS datasets, experienced a significant decline as the transcriptome complexity increased (Fig. 2D and Supplementary Fig. 4D).

We also evaluated software performance under varying degrees of read completeness. In all Nanopore and PacBio datasets, most methods exhibited inferior performance with less-complete sequencing reads, although the precision of IsoQuant (guided) and Bambu (guided) were only slightly impacted. In addition, IsoQuant (guided) also consistently demonstrated the highest and most stable sensitivity

**Fig. 2 | Accuracy of software performance on simulated datasets of Nanopore R103, PacBio SEQUEL CLR, and CCS.** Precision (**A**) and sensitivity (**B**) of the tested methods on simulated data from Nanopore R103, PacBio SEQUEL CLR, and CCS across varying read depths (10X, 25X, 40X, 55X, 70X, with three replicates for each sequencing platform, totaling *n* = 45). Precision (**C**) and sensitivity (**D**) of the performance of tested methods obtained on simulated data of Nanopore R103, PacBio SEQUEL CLR, and CCS across different transcriptome complexity indices (1, 3, 5, 7, 9; values in parentheses denote the actual mean number of isoforms per gene simulated), with three replicates per sequencing platform (*n* = 45 in total). Precision (**E**) and sensitivity (**F**) of the performance of tested methods obtained on simulated data of Nanopore R103, PacBio SEQUEL CLR, and CCS with different read completeness (0.0_0.0: 100% complete, 0.1_0.1: 10% truncated from both ends; 0.2_0.2: 20% truncated from both ends; 0.2_0.0: 20% truncated from 5′ end;

0.4_0.0: 40% truncated from 5′ end; 0.0_0.2: 20% truncated from 3′ end; 0.0_0.4: 40% truncated from 3′ end, three replicates for each sequencing platform, *n* = 63 in total). Precision (**G**) and sensitivity (**H**) of the performance of tested methods on simulated data of Nanopore R103, PacBio SEQUEL CLR, and CCS with different read accuracy (0.8, 0.85, 0.9, 0.95, 1, three replicates for each sequencing platform, *n* = 30 in total). Precision (**I**) and sensitivity (**J**) of the performance of tested methods on simulated data of Nanopore R103, PacBio SEQUEL CLR, and CCS with different annotation completeness (20%, 40%, 60%, 80%, 100%, three replicates for each sequencing platform, *n* = 45 in total). N and P represent datasets generated from the Nanopore and PacBio platforms, respectively. All reported values are expressed as means, with Standard Deviation (SD) detailed in the Source Data file. Source data underlying (**A**–**J**) are provided as a Source Data file.

across different sequencing platforms (Fig. 2E, F and Supplementary Fig. 4E, F). The precision of IsoQuant (unguided) remained stable with 20% truncation from either the 3′ or 5′ end but decreased dramatically with 40% truncation at either end in both Nanopore and PacBio datasets. However, the precision level of IsoQuant (unguided) dropped significantly with 20% truncation at the 3′ end compared to 20% truncation at the 5′ end in Nanopore datasets (Fig. 2E and Supplementary Fig. 4E). StringTie2 (in both modes) exhibited reduced precision and sensitivity as reads became incomplete from both the 3′ and 5′ ends in all datasets. Compared to fully complete reads, StringTie2 showed reduced precision and sensitivity with 20% truncation from the either 3′ or 5′ end, while further truncation unexpectedly improved it in certain models (Fig. 2E, F and Supplementary Fig. 4E, F).

Long-read RNA-seq simulated data with different sequencing accuracy was also examined. For all datasets, most methods exhibited higher precision as sequencing accuracy increased, except TAMA (Fig. 2G and Supplementary Fig. 4G). IsoQuant (guided) and Bambu (guided) consistently demonstrated high precision. Increasing read accuracy had an overall positive impact on sensitivity, except for FLAMES10, which displayed a declining trend after reaching an accuracy level above 0.9 for Nanopore data and 0.85 for PacBio CLR data (Fig. 2H and Supplementary Fig. 4H). IsoQuant (guided) achieved optimal sensitivity for sequencing accuracy values below 0.9, whereas TAMA exhibited the highest sensitivity for sequencing accuracy values above 0.9 in all datasets (Fig. 2H and Supplementary Fig. 4H). String-Tie2 (guided) also showed superior sensitivity for sequencing accuracy values less than 0.9 in PacBio CLR datasets (Fig. 2H and Supplementary Fig. 4H). PacBio CCS data were not included in this analysis, as it can only produce highly accurate reads (>99%).

Considering that the completeness of the reference annotation may also impact isoform detection performance, we investigated the precision and sensitivity of different methods at varying levels of annotation completeness. We analyzed methods, including IsoQuant (guided), Bambu (guided), StringTie2 (guided), FLAIR (guided), FLAMES3, FLAMS10, and TALON, all of which allow the use of reference annotation. Overall, we observed that all methods exhibited improved performance with the increased annotation completeness (Fig. 2I, J and Supplementary Fig. 4I, J). IsoQuant (guided) consistently demonstrated the highest precision, while TALON exhibited the lowest precision. The precision detected by FLAIR (guided) was influenced more by the quality of the input reference across all sequencing platforms (Fig. 2I and Supplementary Fig. 4I). Regarding sensitivity, IsoQuant (guided) achieved the highest sensitivity when completeness exceeded 40%, whereas String-Tie2 (guided) showed the greatest sensitivity at lower completeness levels in Nanopore datasets (Fig. 2J and Supplementary Fig. 4J). For PacBio datasets, IsoQuant (guided), StringTie2 (guided), Bambu (guided), and TALON displayed comparable sensitivity with high annotation completeness, while StringTie2 (guided) generally outperformed others with lower annotation completeness (Fig. 2J and Supplementary Fig. 4J). FLAMES10 exhibited the lowest sensitivity across different sequencing platforms (Fig. 2J and Supplementary Fig. 4J).

## Analyzing isoform detection accuracy using sequins datasets

To conduct a comprehensive evaluation of the accuracy and efficacy of diverse computational approaches, we collated sixteen previously published sequins-based long-read RNA-seq datasets utilizing ONT technology (Supplementary Data 1)[23,36,37]. Read length, and GC content analysis demonstrated consistency across the samples, while samples generated by Zhu et al. displayed slightly lower read quality (Supplementary Fig. 5A–C). The majority of the datasets contained sequins that were spiked into human samples, with a subset of four datasets collected by Dong et al., consisting solely of sequins, resulting in the expected mapping rate and event types (Supplementary Fig. 5D, E)[23,36,37]. The sequins datasets exhibited an average coverage ranging from 300X to 13,000X (Supplementary Fig. 5F and Supplementary Data 1), with lower read completeness representing samples with lower read quality (Supplementary Fig. 6).

Using the sixteen sequins datasets as ground truth, we assessed the precision and sensitivity of twelve methods. Freddie was not included due to its substantial computational demands, possibly stemming from the extreme depths of the sequins datasets. Guided methods, such as IsoQuant (guided), StringTie2 (guided), Bambu (guided), FLAIR (guided), and FLAMES3 exhibited relatively high sensitivity. Unguided methods like UNAGI, IsoQuant (unguided), String-Tie2 (unguided), FLAIR (unguided), and Bambu (unguided) generally showed lower sensitivity, with TAMA exhibiting the lowest sensitivity. TALON, although a guided method, displayed comparatively lower sensitivity than the others (Fig. 3A). Regarding precision, methods like Bambu (guided), IsoQuant (guided), StringTie2 (guided), and FLAME3 consistently showed high precision. IsoQuant (unguided), Bambu (unguided), FLAIR (guided), and StringTie2 (unguided) presented reasonable precision, whereas UNAGI, TALON, TAMA, and FLAIR (unguided) displayed lower precision, indicative of a significant number of false positives (Fig. 3B).

Further analysis was conducted on the types of isoforms detected by different methods within the sequins datasets. UNAGI was excluded from this analysis because its output lacks the strand information required to classify isoform types. The detected isoforms were classified into five categories: full splice match (FSM), incomplete splice match (ISM), novel in catalog (NIC), novel not in catalog (NNC), and intergenic (Fig. 3C and Supplementary Data 2). When comparing the size of the sectors for each method across datasets, FLAIR (unguided) detected several thousand isoforms, while other methods typically identified tens or hundreds, with TAMA detecting the fewest number of isoforms (Fig. 3C and Supplementary Data 2). Reviewing the composition of isoform types detected by each method, guided methods generally identified a higher number of FSM isoforms compared to unguided methods, with IsoQuant (guided) and Bambu (guided) detecting the largest proportion of FSM, followed by StringTie2 (guided) and FLAMES3 (Fig. 3C). FLAIR (in both modes), TALON, and StringTie2 (unguided) reported a significant amount of NIC or NNC, indicating a potentially high level of false positives. Bambu (unguided) and IsoQuant (unguided) detected mostly FSMs or ISMs, suggesting

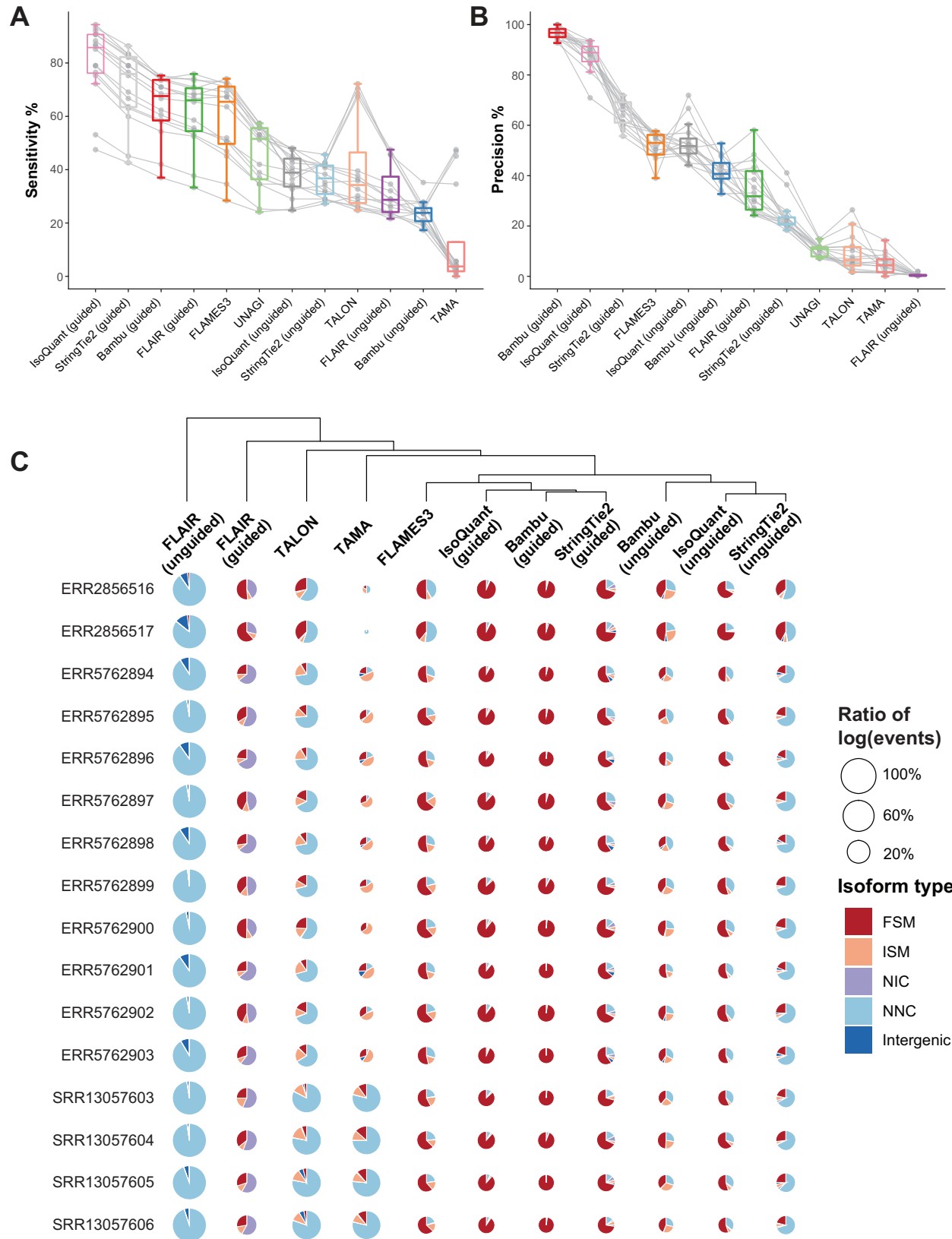

that these two methods may be quite conservative in calling novel isoforms. In conclusion, the performance on the sequins dataset indicates that guided methods generally outperform others, with Iso-Quant (guided), Bambu (guided), and StringTie2 (guided) emerging as the top performers. These results obtained from sequins datasets generally agreed with results based on simulated data.

**Comparative analyses with experimental datasets**

While sequins datasets may partially serve as a proxy for ground truth to a certain extent, they may not fully capture the complexity inherent in experimental datasets. To evaluate the software performance on experimental data, we collected a total of twenty-five experimental long-read RNA-seq datasets generated using Nanopore (GridION,

**Fig. 3 | Software performance on sequins datasets.** Precision (**A**) and sensitivity (**B**) of the performance of methods tested on previously published long-read RNA-seq datasets spiked-in with Sequins DNA (*n* = 16). Individual samples are denoted by gray dots, with gray lines connecting points corresponding to the same sample. The median is indicated by the central line, the boxes delineate the 25th (bottom) and 75th (top) percentiles, and the whiskers extend to the furthest points within 1.5 times the interquartile range from the box. **C** Pie charts display five different isoform types alongside the total counts of isoform events detected by the tested methods across 16 previously published long-read RNA-seq datasets spiked-in with Sequins DNA. FSM, ISM, NIC, and NNC correspond to Full Splice Match, Incomplete Splice Match, Novel In Catalog, and Novel Not In Catalog, respectively. The size of each circle is proportional to the logarithm of the count of isoform events detected by that method divided by the logarithm of the sample-level maximal isoform event count for each sample. Hierarchical clustering was applied to the results from these methods. Source data underlying (**A**, **B**) are provided as a Source Data file.

MinION, and PromethION) and PacBio (CLR reads, sequenced on RS, Sequel, and Sequel II platforms). These datasets were obtained from previous publications as well as datasets generated in our laboratory. They encompassed four different species, namely *Homo sapiens*, *Mus musculus*, *Drosophila melanogaster*, and *Caenorhabditis elegans*[38–48] (see "Methods" and Supplementary Data 1).

The quality control analysis of the average *Q*-Score indicated a diverse range of sequencing quality across the experimental data (Supplementary Fig. 7A). The distribution of read lengths for experimental data exhibited a wide range, spanning from several hundred base pairs to several kilobases (Supplementary Fig. 7B). Additionally, it is noted that the PacBio libraries were size-selected to enrich for longer cDNAs. This selection process results in a higher proportion of longer reads, which likely leads to a more even read length distribution (Supplementary Fig. 7B). Additionally, the GC content showed a generally consistent pattern across different samples (Supplementary Fig. 7C). Mapping status for the experimental data revealed an overall mapping rate of over 75%, although the proportion of primary and secondary alignment events varied among different samples (Supplementary Fig. 8A). Samples with lower read quality in general displayed a lower proportion of base-level match with the reference genome (Supplementary Fig. 8B). Depth analysis demonstrated that most long-read RNA-seq experimental data had depths ranging from 10X to 70X, consistent with the range generated by our simulated data (Supplementary Fig. 8C). Read completeness analysis revealed certain datasets (D2, M5, M6, M7, M8, H3, and H4) exhibited poor read completeness for transcripts longer than 5 kb (Supplementary Fig. 9).

To evaluate the software performance on experimental data, we included IsoQuant (both modes), StringTie2 (both modes), Bambu (both modes), FLAIR (both modes), Freddie, FLAMES, and TALON in these analyses. UNAGI and TAMA were not included due to their high computational resource requirements. As the ground truth for the experimental data are not known, we compared the results of different methods by aligning them side-by-side, employing the most widely used annotation as a reference for each species (Fig. 4A, B and Supplementary Data 3). Mono-exonic transcripts were excluded from classification since they lack splice junctions and may introduce bias into the results. Most methods detected a major proportion of FSM and ISM, except for FLAIR (unguided). Consistent with the sequins datasets, Bambu (guided) and IsoQuant (guided) reported the highest proportion of FSMs, while FLAIR (unguided) detected the largest number of isoforms classified as NNC, potentially due to a high number of false positives called (Fig. 4A, B). TALON and FLAIR (guided) detected a large proportion of NNC and NIC in sequins datasets, respectively. The performance of TALON on experimental data exhibited comparable levels of FSM and ISM detection as FLAMES3 and IsoQuant (unguided), whereas FLAIR (guided) showed similar proportions of FSM and ISM to StringTie2 (guided) (Fig. 4A, B). This discrepancy could potentially be explained by the distinct magnitudes of sequencing depths for sequins and experimental data, as our simulated data suggested TALON and FLAIR (guided) may detect more false positives with increasing sequencing depths (Fig. 2A and Supplementary Fig. 4A). Bambu (unguided) tends to detect a large proportion of ISM in both sequins and experimental datasets (Figs. 4A and 3C). NNC and NIC detected by Freddie remain questionable, given its low precision on simulated data. We also performed a comparative analysis of the results obtained from different methods and quantitatively assessed their similarities using Jaccard statistics, which represent the pairwise overlapping of detected isoforms at the base-pair resolution. StringTie2 (in both modes) and FLAIR (guided) exhibited a notable level of concordance, which may be partly attributed to the higher number of NIC categorizations (Fig. 4 and Supplementary Fig. 10). However, whether these NICs are true positives remains in doubt. After further reviewing the data using Integrative Genomics Viewer (IGV)[49], it appears Bambu (guided) and IsoQuant (guided) yield different sets of true positive isoforms. This discrepancy likely explains their fewer overlapping isoforms despite their commendable performance (Supplementary Fig. 10).

In our initial analysis when including the identification of mono-exonic transcripts, some methods exhibited a high proportion of NIC and intergenic isoforms, especially in datasets from mouse and human species. We hypothesized this might be attributed to the high proportion of transposable elements (TE) in the mammalian genome[50]. To validate this hypothesis, we analyzed the Nanopore long-read RNA-seq datasets from human naïve and primed embryonic stem cells, which were generated as part of this study. Previous studies have indicated that the dynamic expression of TEs could serve as a hallmark for human naïve and primed hESC[51,52], and our earlier research has highlighted their potential functional roles in hESC cell fate determination[30,53]. Therefore, we utilized this model to analyze the isoform types for mono-exonic transcript identification. Most unguided methods, as well as FLAIR (guided), detected a significant proportion of intergenic isoforms (Supplementary Fig. 11A). Compared to randomly shuffled genomic regions, the mono-exonic isoforms detected by FLAIR (in both modes), Bambu (unguided), and Freddie showed a significant enrichment for TE regions (54% to 60% vs. 47%) (Supplementary Fig. 11B). FLAIR (guided) and Freddie seem to detect slightly more expressed TEs. However, it should be noted that the identification of specific TE copies may not be accurate considering their high sequence similarity. Visualization of representative intergenic transcripts specific to naïve hESCs demonstrated a high degree of overlap with previously reported functional TE loci such as LTR5Hs and HERVH/LTR7Y, suggesting the capability of these methods to detect TEs[30,51,52] (Supplementary Fig. 11C, D).

## Differentially isoform usage analyses with both simulated and experimental datasets

Differentially isoform usage (DIU) analysis between groups facilitates the identification of key condition-related isoforms that may possess potential biological significance. To further investigate the impact of different isoform detection tools on downstream analysis, we performed comprehensive DIU analyses on both simulated and experimental long-read RNA-seq datasets from naïve and primed hESCs (Supplementary Fig. 12). As previously reported, there was no clear front-runner for downstream DIU analysis[24]. Therefore, we adopted a consistent downstream workflow using IsoformSwitchAnalyzeR for DIU calling, with the isoforms detected by different methods as input[54,55]. DIU analysis on the simulated naïve and primed hESCs datasets revealed that those methods that performed best in the previous isoform construction analysis, namely IsoQuant (guided), Bambu (guided), and StringTie2 (guided) also demonstrated the highest

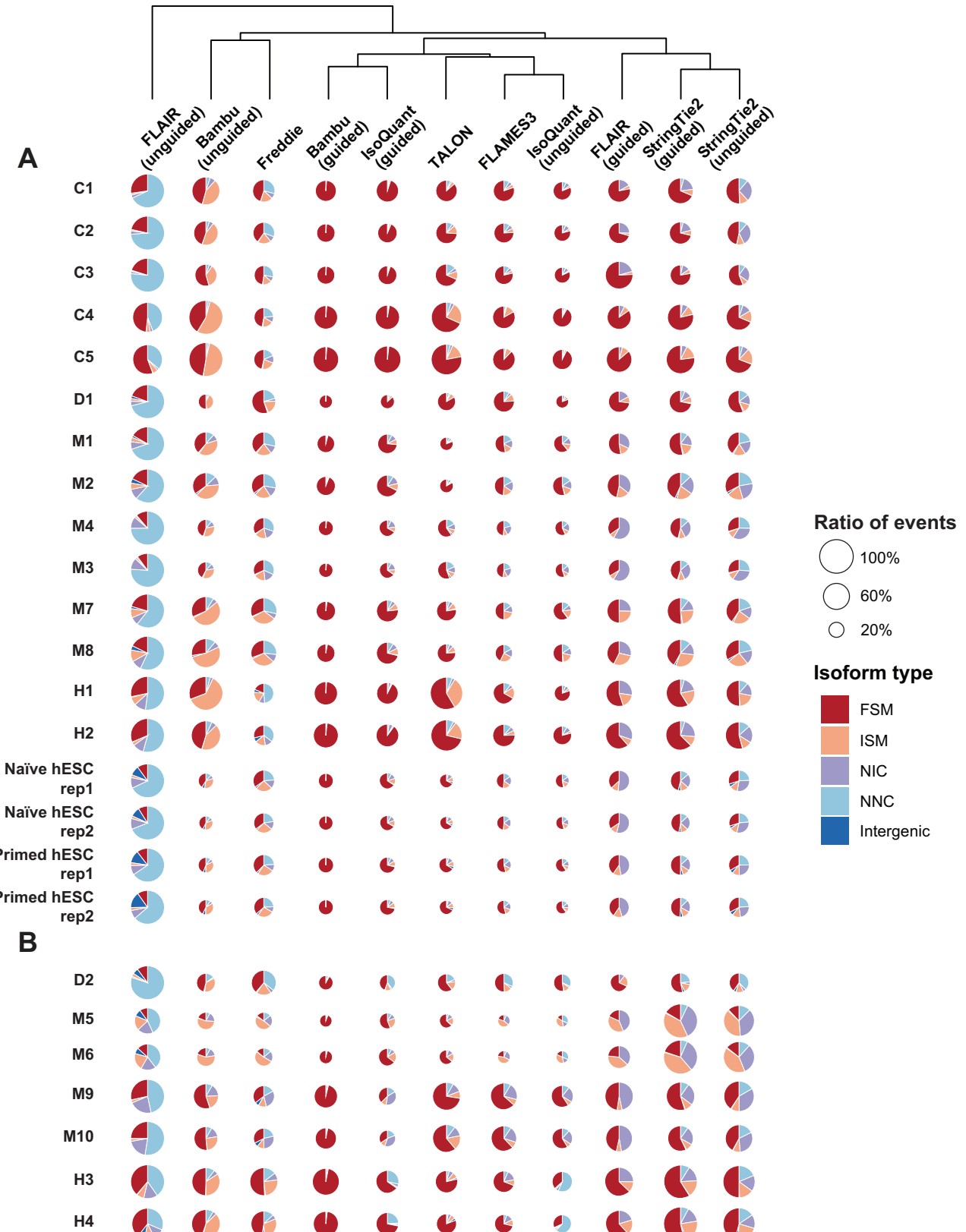

precision and sensitivity (Fig. 5A). Using previously published NGS RNA-seq datasets as a control, we performed DIU analysis on the naïve and primed hESCs applying a similar approach as in the DIU simulation. Isoforms called by different methods were utilized for LRS data, while the human reference genome was used for the NGS data. The results obtained from the experimental long-read RNA-seq datasets exhibited variations across all methods, including the number of differentially used isoforms, the distribution of AS event type, and the consequences of isoform switching (Fig. 5B, C). Moreover, the DIU results from guided methods exhibited varying degrees of overlap with each other, while unguided methods, particularly FLAIR (unguided), identified a large number of results that showed no overlap with

**Fig. 4 | Classification of isoforms detected by different methods from real datasets.** Five different isoform types and the total counts of isoform events detected by different methods across twenty-five experimental datasets collected using the Nanopore (**A**) or PacBio platform (**B**). FSM, ISM, NIC, and NNC represent full splice match, incomplete splice match, novel in catalog, and novel not in catalog, respectively. The size of each circle is proportional to the largest count of isoform events detected by the eleven different methods for each dataset. The results for different methods were hierarchically clustered. The publicly available experimental datasets originate from the following sources: C1: L1 larval stage of *Caenorhabditis elegans*, C2: mix stage of *Caenorhabditis elegans*, C3: young adult stage of *Caenorhabditis elegans*; C4: Wildtype *Caenorhabditis elegans* total RNA replicate 1, C5: Wildtype *Caenorhabditis elegans* total RNA replicate 2, D1: *Drosophila melanogaster*, D2: *Drosophila melanogaster* testis, M1: *Mus musculus* activated CD8 T cell, M2: *Mus musculus* naïve CD8 T cell, M3: *Mus musculus* retinal cells (control), M4: *Mus musculus* retinal cells (glaucomatous), M5: *Mus musculus* CD4SP cells, M6: *Mus musculus* CD8SP cells, M7: *Mus musculus* neural stem cells (E15.5), M8: *Mus musculus* neural stem cells (P1.5), M9: *Mus musculus* cerebral cells, M10: *Mus musculus* hippocampus cells. H1: *Homo sapiens* Beta cells, H2: *Homo sapiens* Beta cells treated with cytokines, H3: *Homo sapiens* Hela cells, H4: *Homo sapiens* iPSC cells. The long-read RNA-seq dataset on Naïve and Primed hESCs was generated in this study.

any other method, possibly due to the higher number of false positive isoforms they called (Fig. 5D, E). The DIU results of NGS data also showed the highest number of unique DIUs compared to results obtained from the LRS datasets (Fig. 5D, E).

We further performed experimental validation on *RPL39L*, which is one of the DIUs identified by both Bambu (guided) and StringTie2 (guided) in naïve and primed hESCs (Fig. 5F, G). The visualization of long-read RNA-seq tracks for *RPL39L* in naïve and primed hESCs suggested an up-regulation of the RPL39L-Long (*RPL39L-L*) isoform and the presence of a novel isoform structure of *RPL39L*, RPL39L-Unknown (*RPL39L-UN*), in primed hESCs compared to naïve hESCs, while the RPL39L-Short (*RPL39L-S*) isoform may express at a similar level in both conditions (Fig. 5F, G). We thus designed isoform-specific RT-qPCR primers and validated the existence of *RPL39L-UN*, as well as the differential usage of *RPL39L-L* isoform in naïve and primed hESCs (Fig. 5H and Supplementary Data 4).

### Computational performance analyses
We developed a profiler to evaluate the computational efficiency of the benchmarked methods, focusing on two key metrics: total run time and average memory consumption. Additionally, considering that transcriptome sizes can vary significantly among species, we analyzed the scalability of the tools using simulated datasets with varying sizes. Based on the results, StringTie2 (both modes) demonstrated the fastest speed, highest memory efficiency, and best scalability among the tested methods. FLAMES, FLAIR, Bambu, and IsoQuant (guided) also exhibited excellent computational performances (Fig. 6A, B). However, it is worth noting that some tools displayed high time and memory requirements, likely attributed to suboptimal algorithm design and data processing approaches, especially in handling SAM/BAM files (Fig. 6A, B).

## Discussion
In this study, we conducted a comprehensive analysis of nine computational tools implemented with thirteen different methods for isoform identification in long-read RNA-seq data. We evaluated their performances using a diverse range of simulated and experimental datasets. We also noticed the emergence of several new isoform discovery and quantification tools, such as ESPRESSO, isONform, TAGET, and IsoTools[56–59]. We decided to exclude ESPRESSO from our analysis due to its extremely high memory consumption. Additionally, it is important to note that isONform is specifically tailored for ONT cDNA sequencing data, while TAGET and IsoTools are designed for the analysis of full-length transcripts from PacBio Iso-Seq data. Since our study primarily focuses on methods compatible with both PacBio and Nanopore data, we did not include these specialized tools.

For the simulation analyses, we selected sequencing depth as a critical factor because this is commonly considered by most tools, especially when calculating which isoforms are likely to be false positives and which should be filtered out. Previous studies have indicated a positive correlation between the read coverage and the number of detected AS events, suggesting that reaching certain depths is necessary to detect most isoforms, particularly for those isoforms with modest expression level[60]. We also considered the number of isoforms per gene as a potential influencing factor to test the robustness of software under different data complexities. Genes with a higher number of splice variants pose challenges for accurate reconstruction, as the identification of branch points and systematic analysis of AS events become increasingly difficult as the number of isoforms per gene increases[61]. Additionally, the quality of reference annotation used by certain methods can significantly impact their performance. Inaccurate gene annotations can lead to erroneous isoform identification, whereas more complete annotations are likely to detect a larger proportion of expressed isoforms[62]. To account for the influence of incomplete reads on isoform identification, we simulated different levels of sequencing read completeness in our analysis. Incomplete reads introduce ambiguity in isoform assignment, posing challenges for accurate identification and analysis[63]. We also included sequencing error rate as a factor, considering that long reads, except for CCS reads, tend to have higher error rates (1–10%) compared to short reads. This high error rate presents challenges for alignment and the accurate detection of exon structures in isoforms[13].

Based on the results, we observed that increasing the sequencing depths did not evidently improve the precision of the methods. This finding can potentially be attributed to the unique characteristics of LRS data, such as its long read length, which allows a full span across isoforms and relatively even coverage of inter-exonic or intra-exonic regions[12]. It should also be noted that some tools exhibited an increase in the detection of false positive isoforms as the read depth increased. This phenomenon can be attributed to the relatively high sequencing error rate of LRS. Specifically, tools that were notably affected by changes in coverage (such as TALON, and FLAIR (unguided)) do not incorporate an error correction step before isoform detections. Conversely, all methods displayed less sensitivity to changes in read depth when processing highly accurate CCS reads. Furthermore, most methods demonstrated higher precision in CCS datasets compared to other error-prone reads, highlighting the advantage of the high accuracy provided by CCS reads. On the other hand, increasing the sequencing depth positively impacted the sensitivity of the software, enabling the identification of transcripts with relatively low expression levels.

We discovered that as the number of isoforms per gene increased the number of true positives detected by each method decreased. While most methods demonstrated improved performance with fewer erroneous reads, the sensitivity of FLAMES exhibited a declining trend as the sequencing accuracy increased under some circumstances. This can be attributed to one of its key parameters "min_sup_cnt", which determines the minimum number of aligned reads required for a transcript to be considered. In our analysis, we varied this parameter as 1 (FLAMES1), 3 (FLAMES3), and 10 (FLAMES10) and evaluated its impact on different levels of read accuracy. Increasing this parameter generally enhanced the precision under different degrees of read accuracy (Supplementary Fig. 13A). When assessing sensitivity for FLAMES1, FLAMES3, and FLAMES10, we observed that the decreasing trend in

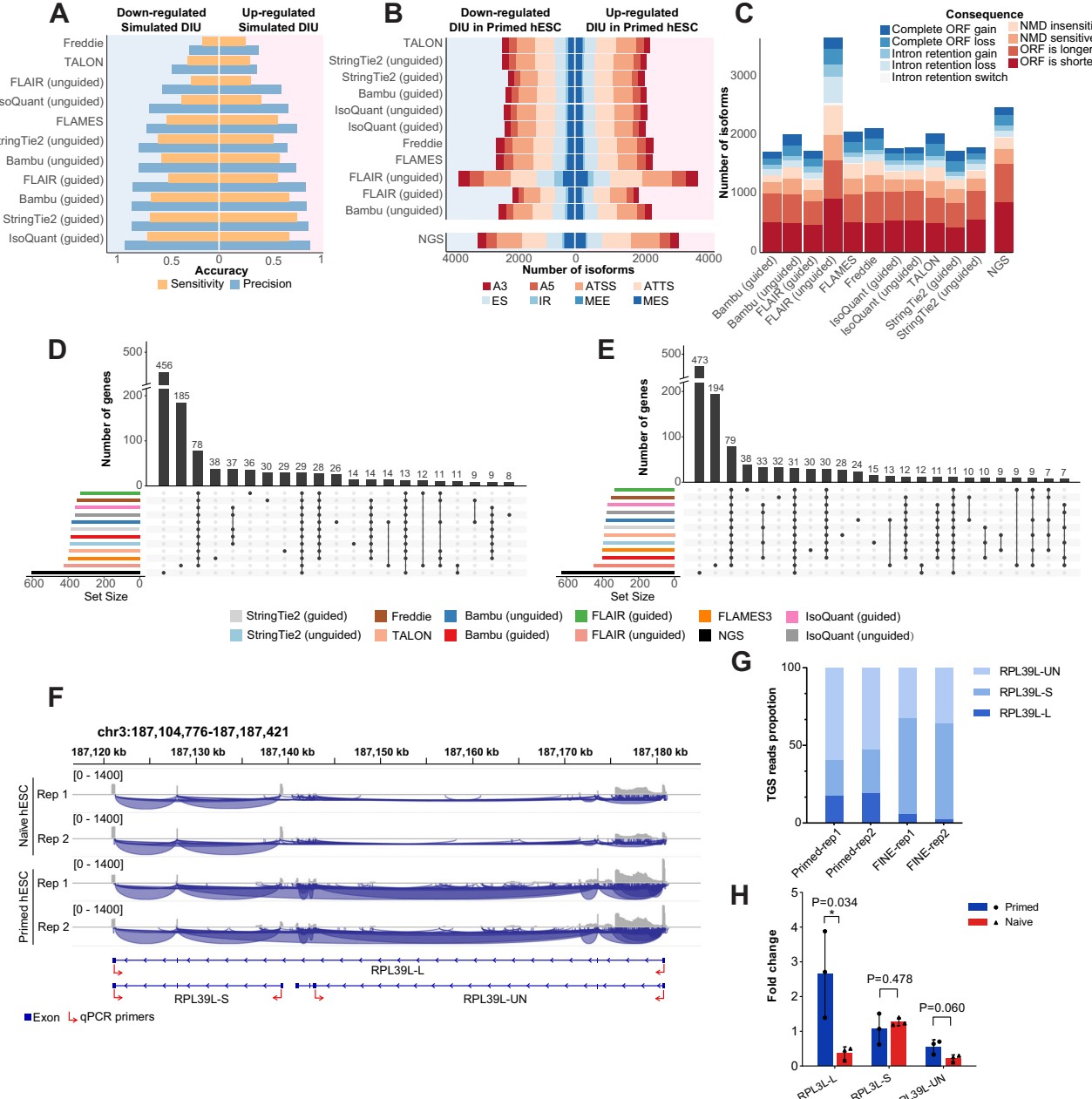

**Fig. 5 | DIU analyses with simulated and real data. A** Bar plot showing the DIU calling accuracy using simulated data with different methods. **B** Bar plot showing alternative isoform types for up-regulated and down-regulated DIU in Primed hESC compared with Naïve hESC detected using different methods. The analysis includes long-read RNA-seq datasets from Naïve and Primed hESC generated in this study, as well as NGS RNA-seq datasets from Naïve and Primed hESC. A3: alternative 3′ splice site, A5: alternative 5′ splice site, ATSS: alternative transcript start site, ATTS: alternative transcript terminated site, ES: exon skipping, IR: intron retention, MEE: mutually exclusive exon, MES: mutually exclusive splicing. **C** Bar plot showing the number of up-regulated and down-regulated DIU isoforms in Primed hESC with different switching consequences. NMD: nonsense-mediated decay, ORF: open reading frame. UpSet plot showing the number of overlapped DIU genes up-regulated (**D**) and down-regulated (**E**) in Primed hESC compared with Naïve hESCs identified by different methods and from the NGS data. **F** IGV screenshot displaying

long-read RNA-seq coverages and splicing junctions of *RPL39L* gene isoforms in Naïve and Primed hESC. The gene model for different transcript isoforms of *RPL39L* is shown under the tracks. *RPL39L-L* and *RPL39L-S* represent isoforms on the gene reference, and *RPL39L-UN* represents the novel isoform detected in this study. Red arrows indicate primers used in RT-qPCR validation experiments. **G** Bar plot representing the LRS read proportions of the three isoforms of *RPL39L* in Naïve and Primed hESC. **H** Bar plot representing the RT-qPCR results of the three different isoforms of *RPL39L*. RT-qPCR experiments were conducted to evaluate the three different isoform levels of *RPL39L* in primed and naïve hESC. Data were gathered from three independent experiments and results were presented as mean ± standard deviation of fold change, with *RPL39L-S* in Primed hESC serving as the control (the raw data presented in Supplementary Data 4). Each experiment's data was represented by dots. *p* values were determined using a two-sided two-sample *T*-test (**p* < 0.05). Source data underlying (**A**–**E**, **H**) are provided as a Source Data file.

sensitivity was less pronounced with a smaller "min_sup_cnt". Notably, when "min_sup_cnt" was set as 1, the sensitivity value increased as the read accuracy improved. This suggests that error-prone reads, normally excluded from analysis, may be ambiguously assigned to transcripts in FLAMES, leading to the identification of potentially false

positive isoforms (Supplementary Fig. 13B). Therefore, it is crucial for users to carefully select the parameter to strike a balance between precision and sensitivity.

The significant negative impact of read completeness on the performance of most tools underscores the importance of generating

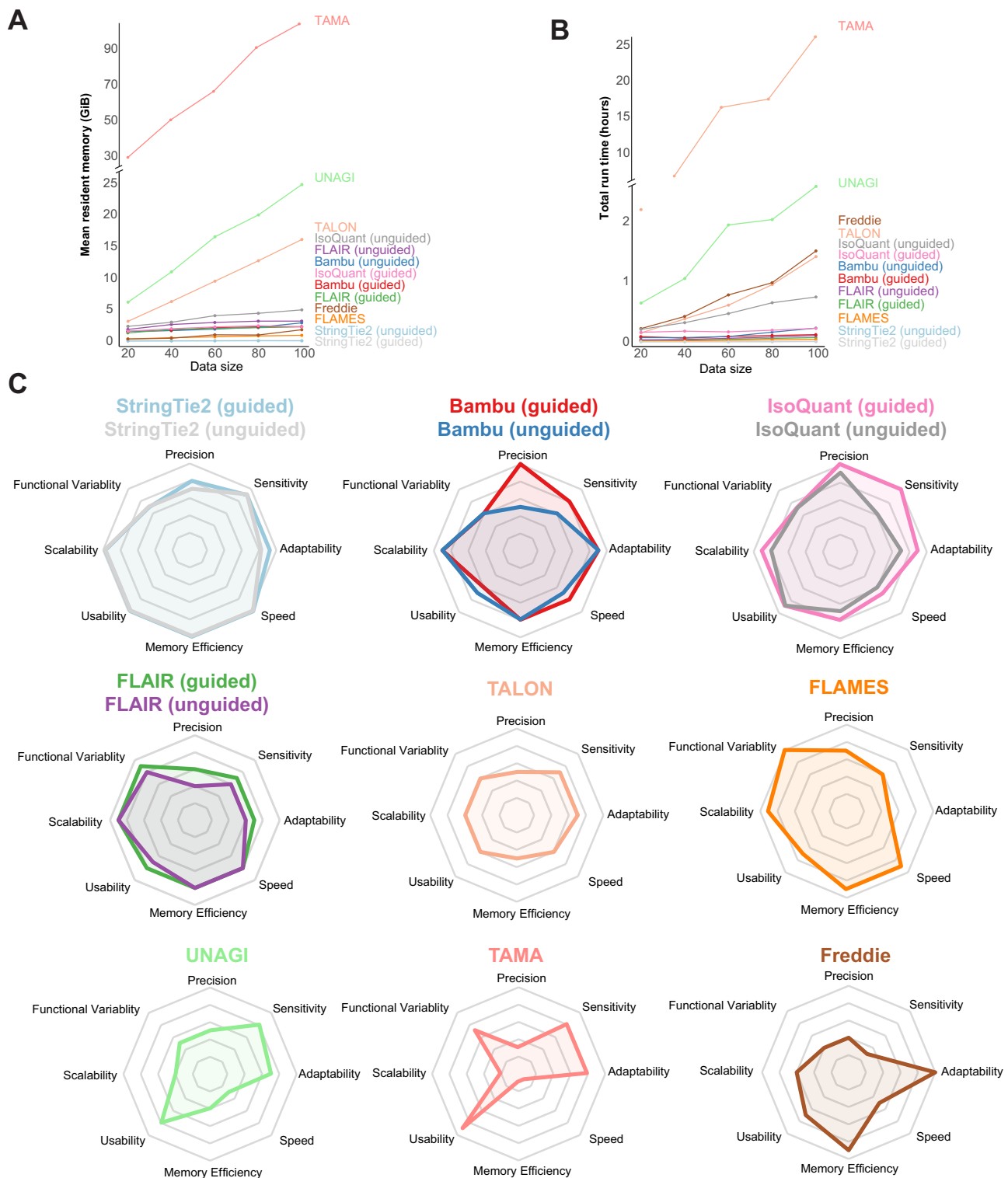

**Fig. 6 | Computational performance analyses and performance summary of the benchmarked methods.** Computational performance for mean resident memory consumed (**A**) and total time spent (**B**) by each method when processing different scales of data. The data size represents various simulated sequencing depths (20X, 40X, 60X, 80X, 100X) of the datasets used for testing. **C** Radar charts summarizing our evaluations of different methods across eight aspects, including precision, sensitivity, adaptability, speed, memory efficiency, usability, scalability, and functional variability. Source data underlying (**A**, **B**) are provided as a Source Data file.

more complete reads for accurate isoform identification. The distinct trends observed in both the precision and sensitivity of StringTie2 under single-end truncated reads may be attributed to its usage of splice graphs for isoform detection. To further investigate this, we tested the influence of the "-R" parameter in StringTie2, which enables read cleaning and collapsing without constructing splice graphs. The

results revealed that StringTie2 (unguided with "-R") exhibited poor performance on truncated reads, similar to other methods (Supplementary Fig. 14A, B). Interestingly, the accuracy of StringTie2 (guided with "-R") during single-end truncation remained relatively unchanged, which is likely due to how StringTie2 refers to the annotated transcripts during its isoform reconstruction process (Supplementary

Fig. 14A, B). It is hypothesized that StringTie2 considers a transcript present in the reference as a true positive if the reads overlap with it to a certain percentage and if the transcription start site (TSS) or transcription termination site (TTS) are matched. For StringTie2 (in both modes) and IsoQuant (guided), our analysis revealed comparable performance in handling either 5' or 3' single-end truncations, as demonstrated using datasets produced by PBSIM3 (PacBio). However, those methods exhibited inconsistent trends for data generated by PBSIM2 (Nanopore). After investigating simulated reads tracks from PBSIM2 and PBSIM3, we hypothesized this discrepancy may be attributed to 5' and 3' degradation in PBSIM2-generated reads, resulting in unmatched TSS and TTS sites, while PBSIM3 did not exhibit this degradation (Supplementary Fig. 14C, D). This difference in degradation could potentially influence the process of splice graph/intron graph traversal and subsequent isoform reconstruction.

It is worth noting that all the methods that rely on a known reference annotation demonstrated increased detection for true positives when provided with a higher-quality reference annotation. Specifically, methods like FLAIR and FLAMES3, which utilize the guidance annotation for error correction, appeared to be more susceptible to changes in the quality of reference annotation. This may be attributed to the fact that more true positive reads could be erroneously "corrected" or filtered out when the provided guidance annotation is incomplete.

Interestingly, it was observed that certain isoform detection tools, such as FLAIR, Bambu, and Freddie can also identify potential TE fragments. The results revealed differences between the TE sequences identified from LRS data and the reference TE annotation, particularly at TE boundaries. This suggests the potential improvement of using long-read RNA-seq data for more precise identification of TE structures.

While we benchmarked the performance of various upstream isoform detection tools on downstream DIU calling, Dong et al. tested different tools specifically designed for downstream DIU calling and noted limited consistency between methods[24]. The inconsistent DIU results we observed also emphasizes the significant impact of isoform detection on downstream analyses. It is worth noting that a substantial proportion of the DIUs identified from the paired NGS data were not detected by any other LRS methods. This could be attributed to the fact that these DIUs were called based on the complete human reference genome annotation for the analysis of NGS datasets, which contains numerous inactive transcripts that may increase ambiguity in read assignment during isoform quantification. Moreover, the nature of NGS data itself, characterized by short read lengths, may increase the likelihood of false positive results. Interestingly, the mouse paralog of the validated DIU *RPL39L* human gene has been proposed to be essential in sperm formation[64], raising the possibility of further investigating whether these DIU isoforms in naïve and primed hESC have functional implications in the transition between human naïve and primed pluripotency.

In a recent study, Dong et al. applied sequins long-read RNA-seq datasets to evaluate the performance of six methods and identified Bambu and StringTie2 as the best isoform detection tools[24]. In our analysis, we included additional tools such as IsoQuant, Freddie, UNAGI, and TAMA, assessed on simulated, sequins, and experimental data. Overall, it can be concluded that among all methods requiring guidance, IsoQuant and Bambu achieved top performance in precision, and IsoQuant and StringTie2 exhibited the best sensitivity. In addition, StringTie2 demonstrated the best performance in terms of usability and computational efficiency (Fig. 6C). FLAMES and FLAIR showed slightly lower accuracy but showcased better functional versatility, such as supporting LRS reads mapping, quantification, or single-cell long-read RNA-seq analysis (Fig. 6C). Among unguided methods, IsoQuant and StringTie2 generally outperformed the others. It is worth noting that StringTie2 employs a splice graph approach for

isoform identification, which represents AS events in a gene as a directed acyclic graph. This approach ensures similar coverage of each exon within an isoform during transcript assembly, thereby avoiding parsimonious yet incorrect results. The splice graph also enables the investigation of AS patterns even under incomplete reference annotation, contributing to the superior performance of StringTie2 in sensitivity and under the unguided mode[65]. Moreover, the use of splice graphs is computationally efficient by compressing transcriptome data into graph structures, which further contributes to the high computational performance of StringTie2. IsoQuant utilizes an intron graph construction influenced by and adapted from the splice graph. This approach offers several advantages, including simplified graph traversal, easier detection of incorrectly spliced sites, and graph simplification. IsoQuant also incorporates a series of optimizations in its algorithm design, contributing to its satisfying performance. These optimizations include read assignments to known isoforms through intron-chain matching, consideration of potential sequencing errors in exonic overlap detections, spliced alignment correction, and refinement of terminal positions. Bambu stands out with its unique utilization of machine learning models for transcript discovery, enabling context-specific isoform quantification. It also introduces a precision-focused threshold called the novel discovery rate (NDR), which is calibrated to provide a reproducible maximum false discovery rate across various analyses, thereby avoiding arbitrary per-sample thresholds commonly employed by other isoform detection methods[20]. Our analyses demonstrate that these algorithm designs indeed have the potential to improve the precision of the isoform construction without significantly sacrificing sensitivity for long-read RNA-seq data.

It is important to acknowledge that the benchmark results obtained from simulated data may not fully capture the complexity of experimental data, there may also be additional factors influencing the performance of isoform detection tools that were not analyzed in this study, such as GC content[66]. Another potential limitation of our study is the use of a uniform model by YASIM to simulate read completeness. This approach simplifies the simulation compared to the non-uniform incompleteness observed in experimental data, where missing portions are likely to be pronounced in longer transcripts. Future iterations of YASIM will explore the integration of a more sophisticated distribution model to better simulate the variation in incomplete reads. Additionally, while default parameters were employed for all evaluated methods in this study, it is conceivable that certain tools may achieve improved results when fine-tuned with settings optimized for particular datasets. Nevertheless, this benchmark study still offers valuable insights into the comparative effectiveness of most published methods for identifying isoform structures from long-read sequencing data. The findings can serve as a guide for the future development of isoform detection algorithms and investigations into alternative splicing events using LRS data.

## Methods
### Data simulation
We utilized YASIM (version 3.2.0) to simulate long-read RNA-seq reads containing AS events, which served as the simulated data in this benchmark study. YASIM enables the generation of realistic long-read RNA-seq with a representative expression profile and AS events, based on distribution models derived from empirical data. YASIM takes reference genome GTF (Gene Transfer Format) and reference genome FASTA as input and generates corresponding realistic FASTQ sequences as output, along with the ground truth annotation GTF and ground truth expression matrix. The overall workflow of YASIM is as follows: the simulator first selects a specific set of genes as expressed genes and generates new AS events. The number of AS events of each type is derived from empirical data, and the resulting information is written into a GTF file referred to as ground truth GTF (gtGTF). The

gtGTF is then transcribed to ground truth cDNA according to the reference FASTA. Additionally, an expression profile consisting of transcripts and their corresponding expression levels is generated above the gtGTF. long-read RNA-seq reads are generated by LLRGs according to the error profile that resembles the error profile of a specific sequencer. In this study, we employed PBSIM3 (at commit b6a68f2d, for PacBio Sequel CCS, Sequel CLR, RSII CCS, RSII CLR data simulation) and PBSIM2 (at commit eeb5a194, for Nanopore R94, R103 data simulation) as LLRGs for simulating PacBio and Nanopore data[27,28]. The *Caenorhabditis elegans* genome was chosen for simulation, and the ce11 UCSC genome version was used as the reference genome.

Isoform-, Gene- and Sample-Level Depth of simulated data is calculated and defined as follows. To calculate depths from simulated and experimental data, the reads are initially unspliced and aligned to the reference transcriptome of corresponding species using BWA (NGS data, defaults) or minimap2 (LRS data, defaults). The depth of each isoform is calculated by dividing the number of primarily aligned bases by the transcribed length of the corresponding isoform. To simplify the calculation process, the depth of each gene is determined by taking the arithmetic mean of the depths of the expressed isoforms within the corresponding gene. Similarly, the depth of each sample is calculated by computing the arithmetic mean of the depths of the expressed isoforms within each sample. The depth of simulated data is calculated using a similar approach, with the exception that, in the simulation, all input isoforms or genes are expressed.

To control the simulated long-read RNA-seq datasets within the dynamic range of gene and isoform expression similar to experimental data, we applied the Gaussian Mixture Model (GMM) and Zipf's distribution. This allowed us to control the overall RNA abundance, varying it by $10^5$- or $10^6$-fold. In the first step, we generated the gene-level depth of each expressed gene. Given targeted mean sequencing depth, this step randomly drew values from a GMM estimated from several experimental *Caenorhabditis elegans* long-read RNA-seq and NGS RNA-seq datasets within a specific range (Supplementary Data 5). By appropriately setting the higher limit, we could generate a distribution with a gene expression variation up to 1000-fold. The second step involved the generation of isoform-level depth within each gene. For this, a Zipf's distribution was assigned as an isoform-level expression to each isoform of a gene. The mean of the distribution was equal to the pre-assigned gene-level depth, and the inequality was controlled by the parameter "--alpha". We set "--alpha" to 4, which allowed for a 1000-fold variation among most multi-isoform genes without significantly affecting the means. By applying a similar filtering strategy, we ensured a 1000-fold variation in isoform expression. The third step encompassed the stranded transcription of gtGTF to FASTA, with the isoform name serving as the sequence ID. This was achieved by retrieving stranded exonic sequences from reference genome sequences and concatenating them together. Additionally, this step generated a tab-separated file that recorded statistics for each isoform (e.g., length, GC content, etc.) and a directory where each isoform was stored as a separate FASTA. The final step involved the generation of raw reads. This step consisted of two substeps: the generation of reads for each isoform (referred to as "sequencing", performed by LLRGs) and the assembly of all generated reads into a single file (referred to as "assembling", performed by an assembler). Firstly, the LLRGs adapter received the isoform sequence, depth, and other customized arguments (e.g., error rate). The adapter then invoked LLRGs, performed cleanup, and passed the generated sequence file to the assembler. The assembler reformatted the read ID, performed additional clipping (either from the 5' or 3' end) if specified, wrote the reads into a single file, and recorded statistics such as the actual number of reads generated.

The detailed simulation process in this study was as follows: firstly, YASIM and *Caenorhabditis elegans* references were installed. AS events were then generated with a transcriptome complexity index set to two as the base gtGTF for all simulations, except for those involving different transcriptome complexity. The default depth was set as 20, complexity as 2, and the error rate as 15% (i.e., 0.85 accuracy) except for those involving varying levels of depths or error rates. The default setting for read truncation was set to 0 and the default setting for reference annotation completeness was set to 100% unless specified. "--low_cutoff" was set for 0.01 for PBSIM2 LLRG and 1 for PBSIM3 LLRG with "--high_cutoff_ratio" set as 200. To simulate long-read RNA-seq data with different numbers of isoforms per gene, a parameter named transcriptome complexity index ("--complexity" parameter) was applied. To simulate long-read RNA-seq data with varying depths, the "-mu" parameter within YASIM was adjusted accordingly. To simulate long-read RNA-seq data with different read completeness, the "--truncate_ratio_5p" or "--truncate_ratio_3p" parameters were adjusted to clip a proportion of reads from 5' end or 3' end, respectively. For simulating long-read RNA-seq reads with different error rates, the YASIM "--accuracy-mean" parameter, which internally set the error rate parameter within PBSIM2/3, was adjusted. To simulate reference annotation with different completeness, the "--percent" parameter was applied to randomly discard a certain proportion of reference annotation in the "sample_transcript" module of labw_utils.bioutils.

All simulated sequencing data was mapped using minimap2 (version 2.17-r941)[31] with "-ax splice -MD" parameters. The resulting SAM files were sorted, converted to BAM format, and indexed using SAMtools (version 1.15.1)[67]. The sorted BAM files were then processed by StringTie2, FLAMES, FLAIR, Bambu, and Freddie, whereas the sorted SAM files were processed by TALON and TAMA. UNAGI, on the other hand, takes FASTQ and the reference genome as inputs, as it is embedded with the alignment process.

For the DIU simulation part, YASIM is currently only compatible with UCSC reference genomes. We used the UCSC release of hg38 National Center for Biotechnology Information (NCBI) RefSeq reference annotation to obtain the count matrix for each isoform of naive and primed hESCs long-read RNA-seq datasets. We used featureCounts to generate the count matrix (parameters "-O -L -t transcript -g transcript_id") (version 2.0.0)[54]. Then, we directly called 96 DIU genes using IsoformSwitchAnalyzeR (version 1.8.0) based on the count matrix obtained from featureCounts[55]. The corresponding isoform annotations of these 96 genes were extracted from the reference GTF as ground truth. Additionally, we randomly selected another 100 genes from the reference annotation, excluding the 96 DIU genes, and mixed them into the gtGTF. We then extracted the isoform counts from these 196 genes from the isoform count matrix generated by featureCounts. This resulting isoform count matrix, containing the 196 genes, served as the input for generating expression profiles in YASIM. We generated four corresponding simulated long-read RNA-seq datasets using the Nanopore R103 error model. The simulated reads were then aligned using minimap2 (version 2.17-r941) and processed with the nine benchmarked methods[31]. The assembled transcriptome was extracted from the hg38 reference genome by GffRead (version 0.12.7) based on the isoform annotation provided by each method[32]. Quantification was performed using Salmon (version 1.8.0) with the parameters of "--ont -l U", and DIU analysis was conducted using IsoformSwitchAnalyzeR[55,68]. To account for any biases introduced by Salmon, we also directly called the DIUs using Salmon directly on the gtGTF of the 196 genes. Finally, we compared the DIUs called based on the quantification results obtained from the transcriptome constructed by isoform detection methods with ground truth DIUs to obtain the precision and sensitivity.

## Process of sequins datasets

Reference sequence and annotation of sequins artificial chromosome were retrieved from https://github.com/novoalab/Nano3P_Seq/tree/master/references. Raw data from previous studies[23,36,37] (accession: ERR2856516, ERR2856517, SRR13057603–SRR13057606, ERR5762894–ERR5762903) (Supplementary Data 1) were retrieved using prefetch and fastq-dump from sra-tools (version 3.0.9) package. The sequences were firstly aligned to the sequins artificial chromosome using minimap2 (version 2.17-r941) with "-x spliced -a" parameters. Pre- and post-alignment quality control was performed as described in the process of experimental datasets. Reads that are successfully aligned were extracted and used for downstream processing using benchmarked tools with the parameters detailed as described below. GFFCompare (version 0.12.6) and SQANTI3 (version 4.2) were utilized to compare the precision, sensitivity, and isoform classes.

## Process of experimental datasets

Pre- and post-alignment quality control of LRS datasets were performed using the following method. The GC content and read length of each read were extracted from FASTQ using the "describe_fastq" module of labw_utils.bio_utils. The phred score of reads was extracted using seqkit (version 2.6.1) "fx2tab" command except for datasets from PacBio Sequel. The alignment rate including the number of secondary and supplementary alignments was detected by the "describe_sam" module of labw_utils.bioutils with default arguments. To calculate base-level match/mismatch events and read completeness, LAST (version 1449) was utilized to align raw LRS reads to reference transcriptome[69]. The aligned MAFs (Multiple Alignment Format) were used as input for the "extract_quality_from_maf" and "extract_read_length_from_maf_gp" modules from yasim_scripts to calculate base-level match/mismatch events and read completeness. To calculate the odds ratio for mono-exonic isoforms over TE regions, we randomly permutated the original intergenic mono-exonic isoforms identified by FLAIR, Bambu (unguided), and Freddie within the intergenic regions with BEDTools shuffle module[34]. We obtained the overlapping regions between test regions (mono-exonic isoforms and their paired random shuffled regions) and TE regions using the BEDTools intersect module[34]. The odd ratios and corresponding p values were determined using the Fisher exact test embedded in R.

The Nanopore direct RNA-seq data was aligned with minimap2 (version 2.17-r941) using the parameters "-ax splice -uf -k14 --MD"[31]. The Nanopore cDNA data was aligned using "-ax splice --MD" parameters using minimap2[31]. For the PacBio data, the alignment was performed using "-ax splice:hq -uf --MD" parameters in minimpa2 (version 2.17-r941)[31]. Quality control of NGS naïve and primed hESCs data was performed using FastQC (version 0.11.8) (https://www.bioinformatics.babraham.ac.uk/projects/fastqc/). The first 10 bp of both paired-end reads were trimmed by Cutadapt (version 2.9)[70]. STAR (version 2.7.1e) was used for the alignment of NGS RNA-seq data with parameters set as "--outFilterMultimapNmax 1000, --outFilterMismatchNmax 3, --outSAMmultNmax 1"[71]. All resulting SAM files were sorted, converted to BAM, and indexed with SAMtools (version 1.15.1)[67].

## Cell culture and RT-qPCR validation

The cell lines used in this study were H1 primed hESC (Wicell Research Institute, Cat# WA01-pcbc) and naïve hESC derived from H1 primed hESC[30]. H1 primed ESC was maintained in mTeSR1 medium (STEMCELL Technologies, Cat#85851), on Matrigel-treated (Corning, Cat#354277) plates and routinely passaged every 4–5 days by using Dispase I (STEMCELL Technologies, Cat#07923), with a split ratio of 1:4 to 1:10. Primed hESC was cultured in normal $O_2$, 5% $CO_2$ condition. Before induction to the naïve stage, primed hESC was cultured on reduced growth factor Matrigel-coated (Corning, Cat# 354230) plates for 2–3 days. When reaching around 60% confluency, culture media was changed to naïve culture media for another 5 days in the incubator with 5% $O_2$ and 5% $CO_2$ (naïve culture media: 24 ml DMEM/f-12 (Nacalai Tesque, Cat#08460-95) and 24 ml Neurobasal (Gibco, Cat#21103049) media, with 500 μl supplement, 500 μl B2 supplement, 500 μl L-Glutamine (Gibco, Cat#25030081), 500 μl Non-essential amino acids (Gibco, Cat#11140050), 0.1 mM B-mercaptoethanol (Sigma, Cat#21985023), 62.5 ng/ml BSA (Sigma, Cat#V900933), supplemented with 0.1 μM Dasatinib (Selleckchem, Cat#S1021), 0.1 μM AZD5438 (TOCRIS, Cat#3968), 0.1 μM SB590885, 1 μM PD0325901, 10 μM Y-27632 (STEMCELL Technologies, Cat#72308), 20 ng/ml human recombinant LIF (Peprotech, Cat#300-05-50UG), 20 ng/ml Activin A (STEMCELL Technologies, Cat#78001.1) and 8 ng/ml of bFGF (Gibco, Cat#PHG0023))[30,72]. Cells were passaged every 4–5 days onto reduced growth factor Matrigel-coated plates using Tryple Express (Gibco, Cat#12604021). Naïve stage cell morphology could be observed after the 2–3 passages. A low $O_2$ incubator (5% $O_2$) was necessary for naïve hESC maintenance.

To perform long-read RNA-seq, total RNA was extracted, reverse transcribed, and quality controlled. The Nanopore cDNA RNA-seq library was constructed according to the Nanopore Ligation Sequencing Kit 1D (PM) using pore type R9.4.1, and sequenced on the PromethION platform by Novogene Co., Ltd. For the RT-qPCR validation experiments, total RNA was extracted by TRIzol RNA isolation reagents (Thermo, Cat#15596026). The cDNA was synthesized by reverse transcription of 0.5 μg RNA using the RT-PCR kit (Vazyme, Cat#R222). RT-qPCR analysis was performed using the RT-qPCR SYBR-green kit (Vazyme, Cat#Q712-03) following the manufacturer's protocol, and each sample contained three replicates for the elimination of technical errors. Isoform-specific primers were designed to assess the expression of different *RPL39L* isoforms using *GAPDH* as the internal control. The primer sequences were listed in Supplementary Data 6. The RT-qPCR experiments were performed using the LightCycler® 480 System (Roche).

## Computational performance analyses

The computational performance analyses were conducted on a workstation with an AMD Ryzen Threadripper 3970 × 32-Core Processor and 256 GiB 2133MT/S DDR4 memory. The system ran Ubuntu 20.04 LTS with the latest updates and a 5.13.0-44-generic kernel. Memory consumption for each method was measured based on the residential set size, a metric commonly used by other profiler tools. The run time of each software was recorded from the start to the termination of the execution.

The profiler used in this study, proc_profiler, is a process-level profiler that collects metrics such as CPU utilization or memory consumption of processes and child processes in an asynchronous manner. It is implemented in Python (version 3.8) on top of psutil library (version 5.9). The profiler is designed to be executed on GNU/Linux systems only. The workflow of the profiler is as follows. It starts by targeting the command line using the subprocess module of Python and records the process ID (PID) of the targeted process. Then, it generates a dispatcher over this process, which further generates several tracers. Each tracer is a single thread that asynchronously probes various aspects of the process, including CPU usage, memory usage, I/O operations, file descriptor, and current status, using the psutil library. The information collected by the tracers is appended to separate GZipped Tab-Separated Values (TSV) files using appenders and reported to a Command-Line Interface (CLI) frontend. The dispatcher also monitors if the process has spawned new processes. If new processes are detected, a new dispatcher is started to monitor these processes. The dispatcher or tracer terminates when the process it is monitoring terminates, and the program exits when the main dispatcher ends. A system dispatcher, which manages tracers that track system-level metrics, is started and terminated at the same time as the main dispatcher. The output of the profiler consists of a folder

containing multiple GZipped TSVs files, each capturing different metrics on the monitored process(es).

## The benchmark evaluation process

To ensure the objectivity of the benchmark study, every evaluated method was tested with default parameters. Namely, StringTie2 (version 2.2.1) was executed with "-L" which allows the processing of long reads. Bambu (version 3.0.8) was executed in custom wrapping scripts "bambu_guided.R" and "bambu_unguided.R" for guided and unguided modes respectively. FLAMES (version 1.0) was executed using its "bulk_long_pipeline.py". Since it does not provide a default set of parameters for its bulk RNA-seq module, we used the configuration file included in the software for running a test dataset. We modified the threshold of support read (controlled by the "min_supp_cnt" parameter), from 10 to 3. In other words, the software discarded transcripts with less than 3 aligned reads. This threshold is also the default threshold used by FLAIR, indicated by the "--support" parameter. The results obtained using FLAMES with read support equals to 10 were included in the results and were denoted as "FLAMES10". Freddie (at commit 501d9f08) was executed with all default parameters with a Groubi license to S.Y. and X.Z. TALON (version 5.0) was executed with default parameters and additional "--ar 20" in labeling SAM files and "--maxFracA 0.5 --minCount 5 --minDatasets 1" in filtering produced transcripts. TAMA (version b0.0.0) was executed with an additional "-x no_cap" parameter. UNAGI (version 1.0.1) was executed with all default parameters. IsoQuant was executed with an additional "--report_novel_unspliced true" to allow the report of mono-exonic transcripts. For the unguided mode of FLAIR (version 1.5), the correction step was not performed as it requires a reference annotation for guidance. The collapse step was conducted directly without the input of a reference annotation. Results of TAMA and UNAGI were also transformed into GTF before being further analyzed. We used GffCompare (version 0.12.6) with default parameters to compare the accuracy of the results obtained with simulated data against ground truths[32]. The transcript level statistics were used as the reference metric. SQANTI3 (4.2) was run in default mode for isoform classification on mono-exonic transcripts excluded by either a custom R script "filter_mono_exon.R" or GFFRead (version 0.12.6) with "-U --gtf" parameter. SQANTI3 was executed with additional "--skipORF" which suppresses ORF prediction[33]. For the DIU analysis on naïve and primed hESCs datasets, isoforms detected from different samples with each software were first merged guided by the reference annotation. The merge function in StringTie2 was used with the parameters "--merge -L -G {reference GTF}", and featureCounts (version 2.0.0) was then used to quantify the isoform expression with "-O -L -t transcript -g transcript_id" parameters based on the merged annotations[54]. For the NGS primed and naïve hESCs data, featureCounts was used to quantify the isoform expression based on the GRCh38.105 reference annotation, using the parameters "-O -t transcript -g transcript_id". IsoformSwitchAnalyzeR (version 1.8.0)[55] was used to perform the DIU analysis based on the expression matrix provided by featureCounts. The isoform switching consequences were obtained using the "extractConsequenceSummary" function, considering selected consequences such as "intron_retention", "NMD_status", and "ORF_seq_similarity". The distribution of AS events for isoforms of differential usage was provided by the "extractSplicingSummary" function.

The method of calculating precision and sensitivity is as follows: TP refers to "true positives", which in this investigation refers to the isoforms detected that match the transcript records in the corresponding ground truth annotation. FN ("false negatives") are transcripts present in the ground truth but missed in the isoforms identified by the software, while FP refers to "false positives" that are found in the detected isoforms but not recorded by any ground truth. Precision is calculated as TP divided by the sum of TP and FP. Sensitivity is calculated as TP divided by the sum of TP and FN. An identified isoform was considered matched with the ground truth (i.e., true positive) if it shared all splice site boundaries exactly compared to an annotated transcript. Regarding the classification of isoforms, FSM refers to the query isoforms having the same number of exons and matched internal junctions with the reference whereas the 5' start or 3' end of the first and last exon can vary, while ISM includes isoforms with fewer 5' or 3' exons but still matched internal junctions compared to the reference. The exact 5' start and 3' end can differ by any amount both for FSM and ISM. NIC includes isoforms without an FSM or ISM match but uses a combination of known donor or acceptor sites, whereas NNC refers to isoforms without FSM/ISM and have at least one unannotated donor or acceptor site. Intergenic means the query isoform is in the intergenic region.

Figure 6C is a summary diagram presented to provide users with a quick and intuitive understanding of the software's performance. Each method was scored based on its ranking in different evaluated aspects, with the top-ranked method receiving the highest score, and the score decreasing as the ranking drops. The score penalty is intensified if there is a significant difference in performance between two consecutive rankings, as observed in the case of TAMA's memory and speed efficiency. Precision, sensitivity, speed, memory, scalability, and adaptability were all ranked based on the results of this benchmark study. Adaptability reflects how significantly the method's performance is influenced by changes in the five different factors investigated. Functional variability mainly demonstrates the diversity of tasks that the software can perform, with higher scores awarded to software with more complex functions. For example, despite having only one additional functional module, FLAMES received the highest score for functional variability due to the complexity of its extra capacity, such as isoform detection from single-cell long-read RNA-seq data. Usability is primarily assessed based on factors such as the smoothness of software installation, ease of usage (whether it is a one-line commander or requires additional scripting/contains multiple steps), and the frequency of encountering bugs during data processing.

## Reporting summary

Further information on research design is available in the Nature Portfolio Reporting Summary linked to this article.

## Data availability

Detailed information on the experimental datasets used in this study can be found in Supplementary Data 1. The datasets used in this study are available in National Centre for Biotechnology Information (NCBI) database under accession code SRR8568873, SRR8568871, SRR13762843, and SRR13762841[38,39]; SRR14630760, SRR14630758, SRR12800923, SRR12800924, SRR22522188, SRR22522033, SRR17960971, SRR17960979, SRR19257398, and SRR19257401[40–44]; ERR3588905, and SRR13494726[45,46]; SRR8929006, SRR8929005, SRR8929004, SRR19055922, and SRR19055924[47,48]; SRR14073786, SRR14073787, SRR14073792, and SRR14073793[30]. The version of reference genomes used in this study are GRCh38.105 (*Homo sapiens*) and GRCm39.105 (*Mus musculus*) from Ensembl, BDGP6.32.53 (*Drosophila melanogaster*), and WBcel235.55 (*Caenorhabditis elegans*) from Ensembl Metazoa. The four naïve and primed hESCs long-read RNA-seq datasets in this study have been deposited in the NCBI's Gene Expression Omnibus (GEO) under GEO Series accession number GSE227911. The simulated datasets generated in this study are available upon request, as the size of these datasets (>600 GB) limited deposition in standard data-sharing platforms. Please contact the corresponding author (wanluliu@intl.zju.edu.cn) of this paper for access, and we will respond within a week. There are no specific restrictions on the use of the simulated datasets. The data for generating figures in this study are provided in the Source Data file. Source data are provided with this paper.

## Code availability

Code for YASIM can be found on GitHub via https://github.com/WanluLiuLab/yasim/[73] or on PYPI https://pypi.org/project/yasim/. Documentation of YASIM can be found at https://labw.org/yasim-docs/. Code for the profiler can be accessed via https://github.com/WanluLiuLab/labw_proc_profiler. Customized analysis code performed in this study can be found on GitHub via https://github.com/WanluLiuLab/2024_LRS_AS_Benchmark_Code[74].

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

## Acknowledgements

We express our gratitude to Dr. Aaron Trent Irving at ZJU-UoE Institute for helping with the language editing. We would also like to acknowledge Dr. Chaochen Wang, and all lab members of the Liu lab at ZJU-UoE Institute for their helpful discussions. We also extend our thanks to the students of the ZJU-UoE Institute BMI-2019 class and staff from the computational biology and system biology (2022-CBSB3) course at ZJU-UoE Institute of Zhejiang University for their valuable suggestions on YASIM. We would also like to acknowledge the technical support provided by the Core Facilities of ZJU-UoE Institute. This work is supported by the National Natural Science Foundation of China (32170551 to W.L.; 32270835 to D.C.; 32250610202 to H.L.), National Key Research and Development Program of China (2022YFC2703503 to H.L.), Fundamental Research Funds for the Central Universities 226-2022-00134 (to W.L.), Alibaba Cloud (to W.L.), and Student Research Training Program (SRTP) of Zhejiang University Y202104390 (to Y.S.).

## Author contributions

Y.S. and W.L. conceived the study and designed experiments. Y.S., Z.Y. and W.L. wrote the manuscript. Z.Y., Y.S., Z.L. and W.L. designed YASIM. Z.Y., Y.S., R.Y., X.C., Z.X. and Y.G. implemented and compiled the documentation for YASIM. Z.Y. implemented the in-house profiler. D.C., H.L. and W.L. designed the hESCs experiments. S.J. and Z.A. performed the hESCs experiments. All authors contributed to the review and correction of the manuscripts.

## Competing interests

The authors declare no competing interests.
