## [Peer Review File · Nature Communications]

Comprehensive Assessment of mRNA Isoform Detection Methods for Long-Read Sequencing DataReviewer #1 (Remarks to the Author):

Overall, gene isoform detection is a fundamental problem in RNA research. The authors aim to benchmark the performance of Third Generation Sequencing data for this goal. Here The authors consider different metrics, including computational efficacy, and also the impact of coverage. Both simulation and real data are used. Here are a few major concerns to address:

1. The benchmark includes real and simulated datasets. The real dataset does not have a known ground truth and simulated datasets are not realistic. The authors may further validate their results by synthetic datasets like SIRV spike-in and other experimental validation like PCR. These have been widely used in previous studies.
2. The authors used the default parameters for the benchmarking. However, some software like Minimap2 provides different parameters to map ONT and PacBio reads to the reference genome considering the different error rates of these two sequencing platforms. They need to further tweak their workflow for benchmarking on different sequencing technologies.
3. The evaluation strategy does not distinguish different sequencing platforms and technologies, while the authors do have both Nanopore and PacBio real data available. Thus, the paper cannot provide a comprehensive guideline for software selection based on different data scenarios.
4. For real data evaluation, the authors only conduct a side-by-side comparison of performance. They can consider using short reads as the baseline performance.
5. The guideline is not categorized into different sequencing technologies. It is unlikely that users only generate one type of data.
6. The authors also do not investigate the impact of factors like different read lengths and error rates on the performance. As they have suggested the error rate has a major impact on the performance, a comprehensive evaluation of different error rates by either simulated or real datasets seems essential.

Reviewer #2 (Remarks to the Author):

Su et al have benchmarked a number of tools for detecting gene isoforms from long-read RNA-seq datasets. Working out which of the growing number of long-read isoform detection methods performs best will be important for the field. As such this manuscript is timely. While the results are quite clear and accurately described there is a lot of important information (particularly about the simulated data) that is not present. This leaves significant concerns as to the appropriateness of the simulated data (described in detail below). The authors have also not tested a recently described long-read isoform detection method (bambu), which a recent preprint found is the best tool in the field. To ensure the usefulness and significance of their manuscript, the authors should include this tool in their revised analysis.

Main points:

Acknowledgement of prior literature and omission of bambu:

- The present study omits citation or discussion of relevant literature including papers which have compared/benchmarked isoform detection methods. This is particularly relevant to the authors claim on line 85 that there aren't other systematic benchmarking studies.

Dong et al 2021 (doi: 10.1093/nargab/lqab028) - compared 3 tools (FLAMES, FLAIR and TALON) for transcript discovery with spike-ins. While not a comprehensive benchmarking of isoform detection methods, this study and it's findings should be cited by the present manuscript

Dong et al, bioRxiv July 22 (<https://doi.org/10.1101/2022.07.22.501076>) - "Benchmarking long-read RNA-sequencing analysis tools using in silico mixtures" - benchmarked 6 methods for isoform detection and found Bambu was the best. Plus it also compared DIU methods. While this is still a pre-

print, the present manuscript should cite and reference the results of this study.

- The present manuscript does not include the current best performing isoform discovery method in its benchmarking. First introduced by Chen et al. 2021 (10.1101/2021.04.21.440736), bambu was found by Dong et al 2022 to be the best isoform discovery method. A manuscript specifically describing bambu & including benchmarking was also recently posted on bioRxiv (Chen et al 2022, 10.1101/2022.11.14.516358). While all these manuscripts are currently in pre-print and therefore don't affect the novelty of the present manuscript. The omission of bambu does affect the significance of the results. I highly recommend the authors also include bambu in their benchmarking.

- When describing why they developed YASIM the authors state on line 120: "However, currently no pipeline has been generated for TGS transcript data simulation". This is seemingly at odds with the large number of simulators for long-read data including, NanoSim (Yang et al Giga Science 2017), PBSIM2 (Ono et al, Bioinformatics, 2021), Badread (Wick, 2019) and others. In addition, if I understand the methods section "Data Simulation" correctly, PBSIM2 and Badread are actually part of YASIM. The authors need to improve the way how they cite, acknowledge and discuss prior literature as well as ensuring they are providing clear and accurate descriptions of what is novel about YASIM.

Description and appropriateness of the simulated data:

- The authors simulated the impact of 5 different sequencing error models. However what these are and the impact of sequence error rate/model on isoform detection software performance are not described in the methods/results. Please add these.

- Given that the authors hypothesize high sequencing error rate affected the results from simulated data by causing false +ves, it is essential the error rates being used represent current long-read sequencing characteristics. For example, PacBio Iso-seq data is now all HiFi reads which are highly accurate (>99.9%). However from what information is available it looks like the authors used a PacBio model from 2016 ("pacbio2016"). This is very unlikely to be representative of current PacBio data. The same may be true for Nanopore data but it is not currently possible to tell.

- An important aspect to the simulated data is the dynamic range of gene and isoform expression. I.e: How much do the expression levels of each gene and isoform differ with respect to each other. A realistic simulation would see isoform abundance within a gene vary by up to 1000 fold, and total expression from each gene differ by a similar amount. Overall RNAs in the dataset should vary by in abundance by 10^5 or 10^6 fold.

I can find no information regarding this aspect of the simulated data, but it is very important to describe. If the dynamic range of the simulated data was too low it would be a significant concern for the appropriateness of the data.

- Even in long-read data, many, sometimes the majority of reads are not of full-length RNAs. With longer genes having higher proportions of incomplete reads. Incomplete reads impact on isoform detection and quantification as they create reads where the isoforms they belong to is ambiguous. How was this modeled in the simulated data? What proportion of reads were incomplete? Currently I don't think the manuscript mentions this at all.

- Simulated read depth - The lowest mean simulated read depth is 20x coverage. How does this compare to real data sets, such as those tested in the manuscript?

I am concerned about the possibility the simulated data is considerably higher coverage than that found in real data and the possible implications of that for the benchmarking. If the read depth coverage in the real data is lower, then additional, lower coverage, simulated data should be created

and tested as well.

- Figure 2 shows software isoform detection sensitivity barely affected by read depth. Conceptually this is surprising. Why is this? Could it suggest the lowest read depth is too high? Or that expression differences between genes and isoforms not great enough?

- FLAIR unguided in figure 3 (real data) uniquely has a very high proportion of novel isoforms (NNC & intergenic). While FLAIR guided shows similar results, including many intergenic transcripts, in mouse. These are likely lots of false +ves. Therefore why did FLAIR not show low precision in the simulated data? This suggests something about simulated data is not representative of real data in an important way. In a similar vein, FLAMES found few isoforms in mouse samples, why?

Differential isoform usage (DIU):

Why use Strawberry RNA-seq data for this and only for this? It is not made clear. Why was DIU performance not also tested with simulated data, where a ground truth for detection methods could be known?

References:

Text uses written references ie: Wang et al. 2008. However the References section uses numbered references. The reference list does not match the order of references in the text, nor are they in any obvious order (like alphabetical). As currently written the Reference section is unsuitable.

YASIM:

This simulator looks useful for testing software performance against a simulated and bespoke transcriptome, thereby allowing various aspects of isoform detection software performance to be examined.

The YASIM Github repo needs improvement to give an overview of what it is and what it can be used for. The current readme covers how to use the various options and parameters but needs the addition of information so a potential user could better understand it's purpose and when it is relevant to use.

Writing:

- Clarity of writing needs improvement and I highly recommend having it read and editing by someone for whom English is their first language. Overall the manuscript is quite clear, however there are numerous examples of incorrect or unusual word choices, which should be corrected. To take the first sentence of the abstract as an example: "managed", "leads to a bright future" & "reserved" are all incorrect or unusual word/phrase choices given the context.

- The manuscript abstract describes the work but doesn't actually inform the reader of the results of the benchmarking. I'd like the authors report the best tool(s) in the abstract.

Minor points:

From what I understand the simulated data simulated a C.elegans transcriptome. This is mentioned in the methods but I would like this reported in the results as well to make it clearer to the reader.

Line 34-35: The authors state: "more than 90% of genes encode multiple protein isoforms (Wang et al., 2008)". This is not what Wang et al (2008) reported. Wang (08) reported that "92-94% of human

genes undergo alternative splicing", but alternative splicing at the RNA level does not correspond 1:1 with alternative protein isoforms. Please fix this claim.

Line 53, the authors state there two main types of long-read RNA-seq are PacBio SMRT-seq and Nanopore direct RNA sequencing. ONT offer 3 different types of RNA-seq (PCR-cDNA, direct cDNA and direct RNA) and direct RNA is almost certainly not the most popular. The authors should revise this sentence.

Figure S2A - shows 5 different "data types", but what they are or mean are not described. Please add this to the figure legend.

Figure S2B&C - reuses the colour scheme of the data-types from S2A, but I don't think the colours refer to the data types. This is a bit confusing. Please change the colour scheme.

FigS4 - Please overlay the FLAMES results shown in figure 2 so the difference in results when the FLAMES read support threshold is changed can be seen.

Please provide some summary statistics for the real data used (perhaps in the sup table). Please include information on the number of reads, no. of pass reads, real quality (ie: Q-scores etc) etc etc.

Please provide the data used to generate Figure 3 in the supplementary information. Ie: The number of isoforms in the different categories.

Figure 6 is useful but what were the performance characteristics required for a tool to meet the various categories?

Response to reviewers:

Dear editor:

We thank the reviewers for these constructive comments on our manuscript. Our point-by-point response draft is below. Reviewers' comments are in normal black type and our response is in normal blue type.

#Reviewer 1

Overall, gene isoform detection is a fundamental problem in RNA research. The authors aim to benchmark the performance of Third Generation Sequencing data for this goal. Here The authors consider different metrics, including computational efficacy, and also the impact of coverage. Both simulation and real data are used. Here are a few major concerns to address:

1. The benchmark includes real and simulated datasets. The real dataset does not have a known ground truth and simulated datasets are not realistic. The authors may further validate their results by synthetic datasets like SIRV spike-in and other experimental validation like PCR. These have been widely used in previous studies.

We thank the reviewer for this constructive suggestion. We agree that validating these results with experimental method is crucial for our analysis. In order to address this issue, we utilized the human embryonic stem cells (hESCs) model established in our lab. In our revised manuscript, we first generated TGS RNA-seq datasets with Nanopore cDNA strategy from human embryonic stem cells (hESCs) in naïve and primed condition (replicates=2 in each condition, n=4 in total). We then performed the differential isoform usage (DIU) analysis on our newly sequenced naïve and primed hESCs datasets and have validated one of the DIU gene *RPL39L* with qRT-PCR.

The DIU analysis from naïve and primed hESCs TGS RNA-seq can be found in figure below and in our revised Figure 5B-E.

Figure 5

B. Bar plot showing alternative isoform types for up-regulated and down-regulated DIU in Primed hESC compared with Naïve hESCs detected using nine methods. The analysis includes TGS RNA-seq dataset from Naïve and Primed hESC generated in this study, as well as NGS RNA-seq dataset from Naïve and Primed hESC. A3: alternative 3' splice site, A5: alternative 5' splice site, ATSS: alternative transcript start site, ATTS: alternative transcript terminated site, ES: exon skipping, IR: intron retention, MEE: mutually exclusive splicing, MES: mutually exclusive splicing. **C.** Bar plot showing the number of up-regulated and down-regulated DIU isoforms in Primed hESC with different switching consequences. NMD: nonsense-mediated decay; ORF: open reading frame. **D, E.** UpSet plot showing the number of overlapped DIU genes up-regulated (**D**) and down-regulated (**E**) in Primed hESC compared with Naïve hESCs identified by different methods and from the NGS data.

Experimental validation with qRT-PCR can be found in figure below and in our revised Figure 5F-H.

Figure 5

F. IGV screenshot displaying TGS RNA-seq coverages and splicing junctions of *RPL39L* gene isoforms in Naïve and Primed hESC. The gene model for different transcript isoforms of *RPL39L* is shown under the tracks. *RPL39L-L* and *RPL39L-S* represent isoforms on the gene reference, and *RPL39L-UN* represents the novel isoform detected in this study. Red arrows indicate primers used in RT-qPCR validation experiments. **G.** Bar plot representing the TGS read proportions of the three isoforms of *RPL39L* in Naïve and Primed hESC. **H.** Bar plot representing the RT-qPCR results of the three different isoforms of *RPL39L*. The RT-qPCR was performed using the isoform-specific primers shown in **F**. Each group involves three biological replicates, and each bar represents the fold change relative to the expression of *RPL39L-S* in Primed hESC. Error bars represent standard deviation. Statistical analysis was performed with a two-sided T-test ($*P < 0.05$).

The revised description for those results can be found on page 12 lines 436-457.

2. The authors used the default parameters for the benchmarking. However, some software like Minimap2 provides different parameters to map ONT and PacBio reads to the reference genome considering the different error rates of these two sequencing platforms. They need to further tweak their workflow for benchmarking on different sequencing technologies.

We thank the reviewer for pointing out this issue. As suggested by the reviewer, we carefully investigated parameters provided in Minimap2 and have tweaked the parameters for real datasets according to the corresponding parameters recommended by minimap2 as follows: 1) For ONT cDNA sequencing data: “-ax splice”; 2) For ONT direct RNA seq: “-ax splice -uf -k14”; 3) For PacBio Iso-seq: “-ax splice:hq -uf”. In our revised manuscript, all real data were pre-processed with the parameters listed above. The revised description for those parameters have been updated in our method section on pages 20-21 lines 760-764.

3. The evaluation strategy does not distinguish different sequencing platforms and technologies, while the authors do have both Nanopore and PacBio real data available. Thus, the paper cannot provide a comprehensive guideline for software selection based on different data scenarios.

We thank the reviewer for this great suggestion. We have now split all our results according to their sequencing platforms (either Nanopore or PacBio) as suggested by the reviewer. For simulated data, we now divided our results according to different error models mimicking various sequencing technologies and platforms as shown in the revised Figure 2 and Figure S4 (i.e., R94: Nanopore R9.4, R103: Nanopore R10.3, RSII CLR: PacBio RS II System (CLR (continuous long read reads), RSII CCS (circular consensus sequencing): PacBio RS II System (CCS reads), SEQUEL CLR: PacBio Sequel System (CLR reads), SEQUEL CCS: PacBio Sequel System (CCS reads)). For real datasets, we have also separated the results according to different sequencing technology (i.e., Nanopore/PacBio), as denoted in our revised Figure 3 and Figure 4.

Analyzing the performance of different software on PacBio CLR and Nanopore platforms show overall consistent conclusions. Since the accuracy for PacBio CCS are usually high (>99%), the performance of different software on PacBio CCS are slightly different from Nanopore and PacBio CLR reads. The results for different software performance on simulated Nanopore, PacBio CLR, and PacBio CCS data can be found on pages 6-9 lines 237-338.

4. For real data evaluation, the authors only conduct a side-by-side comparison of performance. They can consider using short reads as the baseline performance.

This is a great suggestion. Since we included the new TGS RNA-seq dataset from

naïve and primed hESC, we utilized our previously published NGS RNA-seq from the same conditions as dataset for baseline performance (Ai et al., 2022, *Cell Reports*). DIU analysis involved applying the isoforms identified by different methods, while the human reference genome was used for the NGS RNA-seq datasets, followed by featureCounts and isoformSwitchAnalyzeR analysis.

Overall, we observed consistent shared DIU events in NGS and TGS, and the outcomes of DIU analysis using guided methods were observed to have varying degrees of overlap with one another. Conversely, unguided methods, particularly FLAIR (unguided), detected a considerably higher number of non-overlapping results compared to other methods, possibly due to the greater number of false positives generated. The analysis of NGS data revealed the highest count of unique DIUs when compared to the results obtained from TGS datasets. This could be attributed to the fact that these DIUs were identified based on the complete human reference genome annotation, which encompasses a large number of inactive transcripts, leading to ambiguity in read assignment during isoform quantification. Additionally, the short read lengths of NGS data could also potentially contribute to an increased likelihood of identifying false positives.

The results can be found in the figure below and our revised Figure 5B-E, and the description for this part can be found in results session on page 12 lines 436-446, and discussion session on page 16 lines 598-606.

Figure 5

B. Bar plot showing alternative isoform types for up-regulated and down-regulated DIU in Primed hESC compared with Naïve hESCs detected using nine methods. The analysis includes TGS RNA-seq dataset from Naïve and Primed hESC generated in this study, as well as NGS RNA-seq dataset from Naïve and Primed hESC. A3: alternative 3' splice site, A5: alternative 5' splice site, ATSS: alternative transcript start site, ATTS: alternative transcript terminated site, ES: exon skipping, IR: intron retention, MEE: mutually exclusive splicing. **C.** Bar plot showing the number of up-regulated and down-regulated DIU isoforms in Primed hESC with different switching consequences. NMD: nonsense-mediated decay; ORF: open reading frame. **D, E.** UpSet plot showing the number of overlapped DIU genes up-regulated (D) and down-regulated (E) in Primed hESC compared with Naïve hESCs identified by different methods and from the NGS data.

5. The guideline is not categorized into different sequencing technologies. It is unlikely that users only generate one type of data.

We thank the reviewer for this comment. In evaluating the effectiveness of the software, we have considered various factors such as precision, sensitivity, adaptability, speed, memory efficiency, usability, scalability, and functional variability. We have found that apart from precision and sensitivity, different sequencing platforms do not have a significant impact on the performance of the software. As suggested by the reviewer, we have discussed the effects of different sequencing platforms on precision and sensitivity in our revised Figures 2 and S4. The results for different software performance on simulated Nanopore, PacBio CLR, and PacBio CCS data can be found on pages 6-9 lines 237-338.

6. The authors also do not investigate the impact of factors like different read lengths and error rates on the performance. As they have suggested the error rate has a major impact on the performance, a comprehensive evaluation of different error rates by either simulated or real datasets seems essential.

As suggested by the reviewer, we have now added a parameter in our YASIM to consider the error rates for simulated TGS RNA-seq data. With this new parameter

incorporated, we simulated TGS RNA-seq data with different error rates (80%, 85%, 90%, 95%, 100%) on four sequencing platform (R103 (Nanopore), SEQUEL CLR (PacBio), R94 (Nanopore), and RSII (PacBio)). PacBio CCS data were not included for the analysis since it can only produce highly accurate reads (>99%).

For the Nanopore datasets, Bambu (guided) demonstrated consistently high precision while TAMA consistently exhibited low precision with the increased sequencing accuracy. Other methods showed higher precision as sequencing accuracy increased (Figure 2G; S4G). Notably, FLAMES3, FLAMES10, and FLAIR (both modes) appeared to be more affected by sequencing accuracy compared to other methods for Nanopore datasets (Figure 2G; S4G). StringTie2 (guided) achieved optimal sensitivity for sequencing accuracy values below 0.9, whereas TAMA exhibited the highest sensitivity for sequencing accuracy values above 0.9 on Nanopore datasets (Figure 2H; S4H). Increasing read accuracy overall had a positive impact on software sensitivity, except for FLAMES10, which displayed a declining trend after reaching an accuracy level above 90% (Figure 2H; S4H). For PacBio CLR datasets, similar to the Nanopore datasets, increasing read accuracy had a positive impact on the precision performance of different software, except for Bambu (guided), TALON, and TAMA (Figure 2G; S4G). The results for simulation with different accuracy can be found below and in Figure 2G,H and Figure S4G,H. The description for those results can be found on pages 8-9 lines 296-311.

We have now added the error rates as a parameter in the revised YASIM, and the description for this parameter method section on page 19 lines 721-723.

Figure 2

G, H. Precision (G) and sensitivity (H) of the performance of tested methods on simulated data of Nanopore R103, Pacbio SEQUEL CLR, and CCS with different read accuracy (0.8, 0.85, 0.9, 0.95, 1, three replicates for each sequencing platform, n=30 in total).

Supplementary Figure 4

G, H. Precision (G) and sensitivity (H) of the performance for tested methods on simulated data of Nanopore R94, Pacbio RSII CLR, and CCS with different read accuracy (0.8, 0.85, 0.9, 0.95, 1, three replicates for each sequencing platform, n=30 in total)

The length of reads is controlled by the model of the low level read generator (LLRG), and typically the read length parameters provided by the LLRG are suitable for DNA sequencing because the length of DNA contigs is much longer than the sequencing read length. However, for RNA sequencing, most isoforms are shorter than the default sequencing read length produced by the sequencer (approximately 1200-2500), so specifying an average read length may not have a significant impact on the actual generated read length. Considering that the read length fundamentally affects the completeness of the sequencing reads, we further investigated the impact of different read completeness levels on software performance. Thus, we simulated reads with different levels of read completeness, including full length, 10% or 20% truncation from both 3' and 5' end, 10% or 20% truncation of 3' end, and 10% or 20% truncation from 5' ends of the TGS reads.

The results showed that StringTie2 achieved the best performance under incomplete reads assumption. Moreover, with the exception of StringTie2, all other methods demonstrated poor performance with less complete sequencing reads, irrespective of the sequencing platform used. When analyzing Nanopore data, StringTie2's precision and sensitivity decreased as read completeness reduced from the 3' and 5' ends; however, its performance remained stable when reads were cut from the 5' end. Interestingly, StringTie2's accuracy improved as reads became more incomplete from the 3' end. For PacBio CLR data, StringTie2's performance varied depending on read completeness in a manner distinct from that observed for Nanopore datasets. Specifically, as read completeness decreased from either end (3' or 5'), both precision and sensitivity increased. In contrast, neither precision nor sensitivity were affected by changes in read completeness from either end in PacBio CCS data, which differed from the results obtained for either CLR or Nanopore data. The results for this part could be found in Figure below, and in Figure 2E, F and Figure S4E, F. The description and discussion for read completeness simulation and results can be found on page 8 lines 277-294.

We have now added the read completeness as a new parameter in the revised YASIM, and the description for this parameter method section on page 19 lines 719-721.

Figure 2

Supplementary Figure 4

Reviewer #2 (Remarks to the Author):

Su et al have benchmarked a number of tools for detecting gene isoforms from long-read RNA-seq datasets. Working out which of the growing number of long-read isoform detection methods performs best will be important for the field. As such this manuscript is timely.

We thank the reviewer for the positive comments!

While the results are quite clear and accurately described there is a lot of important information (particularly about the simulated data) that is not present. This leaves significant concerns as to the appropriateness of the simulated data (described in detail below). The authors have also not tested a recently described long-read isoform detection method (bambu), which a recent preprint found is the best tool in the field. To ensure the usefulness and significance of their manuscript, the authors should include this tool in their revised analysis.

Main points:

1. Acknowledgement of prior literature and omission of bambu:
The present study omits citation or discussion of relevant literature including

papers which have compared/benchmarked isoform detection methods. This is particularly relevant to the authors claim on line 85 that there aren't other systematic benchmarking studies.

Dong et al 2021 (doi: 10.1093/nargab/lqab028) - compared 3 tools (FLAMES, FLAIR and TALON) for transcript discovery with spike-ins. While not a comprehensive benchmarking of isoform detection methods, this study and it's findings should be cited by the present manuscript

Dong et al, bioRxiv July 22 (<https://doi.org/10.1101/2022.07.22.501076>) - "Benchmarking long-read RNA-sequencing analysis tools using in silico mixtures" - benchmarked 6 methods for isoform detection and found Bambu was the best. Plus it also compared DIU methods. While this is still a pre-print, the present manuscript should cite and reference the results of this study.

We thank the reviewer for those constructive feedbacks. We have now added those citations suggested by the reviewer on page 4 lines 113-115.

2. The present manuscript does not include the current best performing isoform discovery method in its benchmarking. First introduced by Chen et al. 2021 (10.1101/2021.04.21.440736), bambu was found by Dong et al 2022 to be the best isoform discovery method. A manuscript specifically describing bambu & including benchmarking was also recently posted on bioRiXV (Chen et al 2022, 10.1101/2022.11.14.516358). While all these manuscripts are currently in pre-print and therefore don't effect the novelty of the present manuscript. The omission of bambu does affect the significance of the results. I highly recommend the authors also include bambu in their benchmarking.

We thank the reviewer for the great suggestions. We have included Bambu for both simulated data and real data in our revised manuscript. According to our updated results, the guided mode of Bambu obtained the highest precision under every simulate scenario and sequencing platform, and it also performed relatively well in aspect of sensitivity. We have also added related discussion for Bambu in our discussion section on page 17 lines 628-633. During our revision, we also noticed a new isoform discovery and quantification software called ESPRESSO. We tried to included ESPRESSO in our benchmark analysis. However, it is excessively time-consuming on our simulated and experimental data. Due to the large size of our datasets and limited computational resources, we have excluded it from this benchmark analysis. The discussion for ESPRESSO can be found in discussion section on page 13 lines 476-479.

3. When describing why they developed YASIM the authors state on line 120: "However, currently no pipeline has been generated for TGS transcript data simulation".

This is seemingly at odds with the large number of simulators for long-read data including, NanoSim (Yang et al Giga Science 2017), PBSIM2 (Ono et al, Bioinformatics, 2021), Badread (Wick, 2019) and others. In addition, if I understand the methods section "Data Simulation" correctly, PBSIM2 and Badread are actually part of YASIM.

The authors need to improve the way how they cite, acknowledge and discuss prior literature as well as ensuring they are providing clear and accurate descriptions of what is novel about YASIM.

We thank the reviewer for pointing out these issues. As mentioned by the reviewer, there are several TGS Low-Level Read Generators (LLRGs) such as PBSIM1/2/3, NanoSim, and Badread that could be applied to simulate TGS reads. In our study, we developed an upper-level TGS RNA-seq reads simulator, YASIM, which is compatible with multiple previous TGS LLRGs such as PBSIM1/2/3, NanoSim, and Badread. YASIM allows for the simulation of datasets with user-defined read depths, novel AS events, sequencing error rate, reads completeness, and various profiles of sequencing error models. Using YASIM, we were able to generate simulated TGS RNA-seq datasets to systematically evaluate the performance of eleven isoform detection methods under different scenarios. The revised description for these LLRGs as well as the novelty for YASIM has been updated in the introduction section on page 4, lines 118-126.

4. Description and appropriateness of the simulated data:

- The authors simulated the impact of 5 different sequencing error models. However what these are and the impact of sequence error rate/model on isoform detection software performance are not described in the methods/results. Please add these.

We thank the reviewer for this great suggestion. We have now split all our results according to their sequencing platforms (either Nanopore or PacBio) as suggested by the reviewer. For simulated data, we now divided our results according to different error models mimicking various sequencing technologies and platforms as shown in the revised Figure 2 and Figure S4 (i.e., R94: Nanopore R9.4, R103: Nanopore R10.3, RSII CLR: PacBio RS II System (CLR (continuous long read) reads), RSII CCS (circular consensus sequencing): Pacbio RS II System (CCS reads), SEQUEL CLR: PacBio Sequel System (CLR reads), SEQUEL CCS: Pacbio Sequel System (CCS reads)). For real datasets, we have also separated the results according to different sequencing technology (i.e., Nanopore/PacBio), as denoted in our revised Figure 3 and Figure 4.

Analyzing the performance of different software on PacBio CLR and Nanopore platforms show overall consistent conclusions. Since the accuracy for PacBio CCS are usually high (>99%), the performance of different software on PacBio CCS are slightly different from Nanopore and PacBio CLR reads. The generation of

simulated data on different sequencing platforms can be found on page 5-6, lines 187-221. The results different software performance on simulated Nanopore, PacBio CLR, and PacBio CCS data can be found on page 6-9 lines 223-338.

5. Given that the authors hypothesize high sequencing error rate affected the results from simulated data by causing false +ves, it is essential the error rates being used represent current long-read sequencing characteristics. For example, PacBio Iso-seq data is now all HiFi reads which are highly accurate (>99.9%). However from what information is available it looks like the authors used a PacBio model from 2016 ("pacbio2016"). This is very unlikely to be representative of current PacBio data. The same may be true for Nanopore data but it is not currently possible to tell.

We thank the reviewer for this suggestion. As suggested by the reviewer, we have now included the PacBio HiFi reads (CCS) in our simulated dataset and we've also included two recent types of sequencing chemistry for Nanopore including R9.4 (released 2016) and R10.3 (released 2020). In addition, we have now split all our evaluation metrics by the sequencing platform.

In addition to incorporating recent model from Nanopore and PacBio in our simulated data, we have now added a parameter in our YASIM to consider the error rates for simulated TGS RNA-seq data. With this new parameter incorporated, we simulated TGS RNA-seq data with different error rates (80%, 85%, 90%, 95%, 100%) on four sequencing platform (R103 (Nanopore), SEQUEL CLR (PacBio), R94 (Nanopore), and RSII (PacBio)). PacBio CCS data were not included for the analysis since it can only produce highly accurate reads (>99%).

For the Nanopore datasets, Bambu (guided) demonstrated consistently high precision while TAMA consistently exhibited low precision with the increased sequencing accuracy. Other methods showed higher precision as sequencing accuracy increased (Figure 2G; S4G). Notably, FLAMES3, FLAMES10, and FLAIR (both modes) appeared to be more affected by sequencing accuracy compared to other methods for Nanopore datasets (Figure 2G; S4G). StringTie2 (guided) achieved optimal sensitivity for sequencing accuracy values below 0.9, whereas TAMA exhibited the highest sensitivity for sequencing accuracy values above 0.9 on Nanopore datasets (Figure 2H; S4H). Increasing read accuracy overall had a positive impact on software sensitivity, except for FLAMES10, which displayed a declining trend after reaching an accuracy level above 90% (Figure 2H; S4H). For PacBio CLR datasets, similar to the Nanopore datasets, increasing read accuracy had a positive impact on the precision performance of different software, except for Bambu (guided), TALON, and TAMA (Figure 2G; S4G). The results for simulation with different accuracy can be found in figure below and in Figure 2G,H and Figure S4G,H. The description for those results can be found on page 8-9 lines 296-311.

Figure 2

G, H. Precision (G) and sensitivity (H) of the performance of tested methods on simulated data of Nanopore R103, Pacbio SEQUEL CLR, and CCS with different read accuracy (0.8, 0.85, 0.9, 0.95, 1, three replicates for each sequencing platform, n=30 in total).

Supplementary Figure 4

G, H. Precision (G) and sensitivity (H) of the performance for tested methods on simulated data of Nanopore R94, Pacbio RSII CLR, and CCS with different read accuracy (0.8, 0.85, 0.9, 0.95, 1, three replicates for each sequencing platform, n=30 in total)

6. An important aspect to the simulated data is the dynamic range of gene and isoform expression. I.e: How much do the expression levels of each gene and isoform differ with respect to each other. A realistic simulation would see isoform abundance within a gene vary by up to 1000 fold, and total expression from each gene differ by a similar amount. Overall RNAs in the dataset should vary by in abundance by 10^5 or 10^6 fold.

I can find no information regarding this aspect of the simulated data, but it is very important to describe. If the dynamic range of the simulated data was too low it would be a significant concern for the appropriateness of the data.

We thank the reviewer for the constructive feedbacks. We agree considering the dynamic range of gene and isoform expression is important for our simulation. Thus, in our revised YASIM algorithm, we applied Gaussian Mixture Model (GMM) and Zipf's distribution to control the simulated TGS RNA-seq datasets within the dynamic range of gene and isoform expression similar to real data. This allowed us to control the overall RNA abundance, varying it by 10^5 or 10^6 fold. The detailed description could be found in method section on pages 18-19, lines 682-696.

With our new algorithm, we analyzed the isoform-level and gene-level expression variation. As shown below in "Response To Reviewer Figure 1", we measured the variation of the top 10% expressed genes and calculated their pairwise fold change of expression level relative to the lowest 10% expressed genes (panel A) and

demonstrated the variation of genes can be up to 1000 fold. Variation distribution for the pairwise fold change of expression level of top 10% expressed isoforms vs. the lowest 10% expressed isoforms also showed the that the variation of isoforms can reach 1000 fold (panel B). Variation of fold change for the expression levels of all isoforms within genes with multiple isoforms also demonstrated the isoform expression variation can be up to 1000 fold (panel C).

Response To Reviewer Figure 1

A-C. Violin plots represent depth variation among the highest 10% vs. the lowest 10% genes (A), highest 10% vs. the lowest 10% isoforms (B), and all isoforms within multi-isoformal genes (C) under different simulation scenarios for sequence platforms. N and P represent datasets generated from the Nanopore and Pacbio platforms, respectively.

7. Even in long-read data, many, sometimes the majority of reads are not of full-length RNAs. With longer genes having higher proportions of incomplete reads. Incomplete reads impact on isoform detection and quantification as they create reads where the isoforms they belong to is ambiguous. How was this modeled in the simulated data? What proportion of reads were incomplete? Currently I don't think the manuscript mentions this at all.

We thank the reviewers for this constructive feedback. We've added a new simulated scenario considering different read completeness including full length, 10% or 20% truncation from both 3' and 5' end, 10% or 20% truncation of 3' end, and 10% or 20% truncation from 5' ends of the TGS reads. The results showed that StringTie2 achieved the best performance under incomplete reads assumption. Moreover, with the exception of StringTie2, all other methods demonstrated poor performance with less complete sequencing reads, irrespective of the sequencing platform used. When analyzing Nanopore data, StringTie2's precision and sensitivity decreased as read completeness reduced from the 3' and 5' ends; however, its performance remained stable when reads were cut from the 5' end. Interestingly, StringTie2's accuracy improved as reads became more incomplete from the 3' end. For PacBio CLR data, StringTie2's performance varied depending on read completeness in a manner distinct from that observed for Nanopore datasets. Specifically, as read completeness decreased from either end (3' or 5'), both precision and sensitivity increased. In contrast, neither precision nor

sensitivity were affected by changes in read completeness from either end in PacBio CCS data, which differed from the results obtained for either CLR or Nanopore data. The results for this part could be found in Figure below, and in Figure 2E, F and Figure S4E, F. The description for read completeness simulation and results can be found on page 6 lines 210-215, and page 8 lines 277-294.

We have now added the read completeness as a new parameter in the revised YASIM, and the description for this parameter method section on page 19 lines 719-721.

Figure 2

E, F. Precision (E) and sensitivity (F) of the performance of tested methods obtained on simulated data of Nanopore R103, Pacbio SEQUEL CLR, and CCS with different read completeness (0.0_0.0: 100% complete, 0.1_0.1: 10% truncated from both ends; 0.2_0.2: 20% truncated from both ends; 0.2_0.0: 20% truncated from 5' end; 0.4_0.0: 40% truncated from 5' end; 0.0_0.2: 20% truncated from 3' end; 0.0_0.4: 40% truncated from 3' end, three replicates for each sequencing platform, n=63 in total).

Supplementary Figure 4

E, F. Precision (E) and sensitivity (F) of the performance for tested methods on simulated data of Nanopore R94, Pacbio RSII CLR, and CCS with different read completeness (0.0_0.0: 100% complete, 0.1_0.1: 10% truncated from both ends; 0.2_0.2: 20% truncated from both ends; 0.2_0.0: 20% truncated from 5' end; 0.4_0.0: 40% truncated from 5' end; 0.0_0.2: 20% truncated from 3' end; 0.0_0.4: 40% truncated from 3' end, three replicates for each sequencing platform, n=63 in total).

8. Simulated read depth - The lowest mean simulated read depth is 20x coverage. How does this compare to real data sets, such as those tested in the manuscript? I am concerned about the possibility the simulated data is considerably higher coverage than that found in real data and the possible implications of that for the benchmarking. If the read depth coverage in the real data is lower, then additional, lower coverage, simulated data should be created and tested as well.

We thank the reviewer for this great suggestion. After investigating the read depths of all the real data collected in this manuscript including both previously published data and the new TGS RNA-seq dataset from naïve and primed human embryonic

stem cells generated by our lab, we found the read depths of most real datasets range from 5X to 80X. These results can be found in figure below and in our revised Figure S6C.

Supplementary Figure 6

C. Quality control analyses of experimental datasets was performed to assess read depths (C). The publicly available experimental datasets originate from the following sources: C1: L1 larval stage of *C. elegans*, C2: mix stage of *C. elegans*, C3: young adult stage of *C. elegans*; C4: Wildtype *C. elegans* total RNA replicate 1, C5: Wildtype *C. elegans* total RNA replicate 2, D1: *Drosophila*, D2: *Drosophila testis*, M1: mouse activated CD8 T cell, M2: mouse naïve CD8 T cell, M3: mouse retinal cells (control), M4: mouse retinal cells (glaucomatous), M5: mouse CD4SP cells, M6: mouse CD8SP cells, M7: mouse neural stem cells (E15.5), M8: mouse neural stem cells (P1.5), M9: mouse cerebral cells, M10: mouse hippocampus cells. H1: human Beta cells, H2: human Beta cells treated with cytokines, H3: human Hela cells, H4: human iPSC cells. The TGS RNA-seq dataset on Naïve and Primed hESCs was generated in this study.

Based on this, in our revised manuscript, we varied the sequencing depth for expressed isoforms, generating depths of 10X, 25X, 40X, 55X, and 70X. The results for the performance of different software under various depths could be found in figure below and Figure 2A, B, and Figure S4A, B.

Figure 2

A, B. Precision (A) and sensitivity (B) of the performance of tested methods on simulated data of Nanopore R103, Pacbio SEQUEL CLR, and CCS with different read depths (10X, 25X, 40X, 55X, 70X with three replicates for each sequencing platform, n=45 in total).

Supplementary Figure 4

A, B. Precision (A) and sensitivity (B) of the performance for tested methods on simulated data of Nanopore R94, Pacbio RSII CLR, and CCS with different read depths (10X, 25X, 40X, 55X, 70X with three replicates for each sequencing platform, n=45 in total).

9. Figure 2 shows software isoform detection sensitivity barely affected by read depth. Conceptually this is surprising. Why is this? Could it suggest the lowest read depth is too high? Or that expression differences between genes and isoforms not great enough?

We thank the reviewers for pointing out this issue. In our initial version, we applied a uniform distribution to simulate the gene and isoform expression variation. As pointed out by the reviewer in the question 6, we now have adapted GMM and Zipf's distribution to simulate a more realistic isoform and gene expression profile which allows the generation of RNAs varied in abundance by 10^5 or 10^6 . The revised results show that software's sensitivity increased as the read depth increased (Figure 2B, S4B and in Figure below), indicating that those transcripts with low abundance are more likely to be recovered with higher sequencing coverage. The description has been revised in the results and discussion section on page 7 lines 227-254.

Figure 2

B. Sensitivity (B) of the performance of tested methods on simulated data of Nanopore R103, Pacbio SEQUEL CLR, and CCS with different read depths (10X, 25X, 40X, 55X, 70X with three replicates for each sequencing platform, n=45 in total).

Supplementary Figure 4

B. Sensitivity (B) of the performance for tested methods on simulated data of Nanopore R94, Pacbio RSII CLR, and CCS with different read depths (10X, 25X, 40X, 55X, 70X with three replicates for each sequencing platform, n=45 in total).

10. FLAIR unguided in figure 3 (real data) uniquely has a very high proportion of novel isoforms (NNC & intergenic). While FLAIR guided shows similar results, including many intergenic transcripts, in mouse. These are likely lots of false +ves. Therefore why did FLAIR not show low precision in the simulated data? This suggests something about simulated data is not representative of real data in an important way. In a similar vein, FLAMES found few isoforms in mouse samples, why?

We thank the reviewer for pointing out this problem. To investigate this issue, we hypothesized that the intergenic isoforms may be attributed to the high proportion of transposable elements, especially in the mammalian genome (Deniz et al., 2019, *Nat Rev Genet*). To validate this, we first analyzed mono-exonic transcripts identified by different software independently from our newly generated TGS RNA-seq data from naïve and primed hESCs as shown below and figure below. From this analysis, we found FLAIR (guided/unguided), Freddie, Bambu (unguided), Stringtie2 (unguided) would identify extremely high proportion of intergenic isoforms.

Supplementary Figure 8

A. Pie chart showing the different types of mono-exonic isoforms detected by different methods in Naive and Primed hESC TGS RNA-seq datasets. FSM, ISM, NIC, NNC represent full splice match, incomplete splice match, novel in catalog, and novel not in catalog, respectively.

Previous studies have indicated that the dynamic expression of TEs could serve as a hallmark for human naïve and primed hESCs (Theunissen et al., 2016, *Cell Stem Cell*; Pontis et al., 2019, *Cell Stem Cell*), and our earlier research has highlighted their potential functional roles in hESC cell fate determination (Xiang et al., 2022, *Nat Commun*; Ai et al., 2022, *Cell Rep*). The visualization of representative intergenic transcripts specific to naïve hESC demonstrated a high degree of overlapped with previously reported functional TE loci such as LTR5Hs and HERVH/LTR7Y (Figure S8B, C and figure shown below) (Theunissen et al., 2016, *Cell Stem Cell*; Pontis et al., 2019, *Cell Stem Cell*; Ai et al., 2022, *Cell Rep*).

Supplementary Figure 8

B, C. IGV screenshots displaying Naive and Primed hESC TGS RNA-seq tracks over representative transposable element sites, specifically LTR5_Hs_dup3 (B) and LTR7Y_dup2 (C). The green bar represents the reference annotation for transposable elements, blue bars represent the mono-exonic transcripts identified by FLAIR (guided), FLAIR (unguided), Bambu (unguided), and Freddie in Naive hESC rep1, and brown bars represent the mono-exonic transcripts identified by FLAIR (guided), FLAIR (unguided), Bambu (unguided), and Freddie in Naive hESC rep2. The text under each bar represents the transcript ID assigned by different methods, and the white arrows on each bar represent the annotated transcript direction by each method.

To avoid potential biases introduced by batch effects, we also included more real TGS RNA-seq datasets from different species in the revised version. After filtering out the mono-exonic isoforms identified by each method and including more read datasets, results showed that there were much less results classified as “intergenic” for all datasets, even though FLAIR (unguided) still reported a significant proportion of NNC (as shown in figure below and in our revised Figure 3). FLAMES now detected similar number of isoforms compared to other methods except for FLAIR (unguided).

Figure 3

Figure 3. Classification of isoforms detected by different methods from various experimental datasets.
A, B. Five different isoform types and the total counts of isoform events detected by nine different methods across 25 experimental datasets collected using the Nanopore (A) or Pacbio platform (B). FSM, ISM, NIC, NNC represent full splice match, incomplete splice match, novel in catalog, and novel not in catalog, respectively. The size of each sector is proportional to the largest count of isoform events detected by the nine different methods for each dataset. The results for the nine methods were hierarchically clustered. The publicly available experimental datasets originate from the following sources: C1: L1 larval stage of *C. elegans*, C2: mix stage of *C. elegans*, C3: young adult stage of *C. elegans*, C4: Wildtype *C. elegans* total RNA replicate 1, C5: Wildtype *C. elegans* total RNA replicate 2, D1: *Drosophila*, D2: *Drosophila* testis, M1: mouse activated CD8 T cell, M2: mouse naive CD8 T cell, M3: mouse retinal cells (control), M4: mouse retinal cells (glaucomatous), M5: mouse CD4SP cells, M6: mouse CD8SP cells, M7: mouse neural stem cells (E15.5), M8: mouse neural stem cells (P1.5), M9: mouse cerebral cells, M10: mouse hippocampus cells. H1: human Beta cells, H2: human Beta cells treated with cytokines, H3: human Hela cells, H4: human iPSC cells. The TGS RNA-seq dataset on Naive and Primed hESCs was generated in this study.

We reasoned the the FLAIR (unguided) detected NNC isoforms are likely to be false positives, given its relative low precision and precision assessed with the new simulated dataset as shown in Figure 2 and S4.

The description for above mentioned new analysis has been revised in the results section on page 10-11 lines 363-405.

11. Differential isoform usage (DIU):

Why use Strawberry RNA-seq data for this and only for this? It is not made clear. Why was DIU performance not also tested with simulated data, where a ground truth for detection methods could be known?

We thank the reviewer for this great suggestion. In our previous version, we cannot find a TGS RNA-seq dataset with good control to perform DIU analysis. To overcome this issue, in our revised version we generated TGS RNA-seq dataset with Nanopore cDNA strategy from human naïve and primed hESCs (replicates=2 in each condition, n=4 in total). In our revised manuscript, we performed the DIU analysis on this well controlled naïve vs. primed hESC TGS RNA-seq datasets. In addition, we utilized our previously published NGS RNA-seq from the same conditions as dataset for baseline performance (Ai et al., 2022, *Cell Reports*). The DIU analysis from naïve and primed hESCs TGS RNA-seq can be found in figure below and in our revised Figure 5B-E.

Figure 5

B. Bar plot showing alternative isoform types for up-regulated and down-regulated DIU in Primed hESC compared with Naïve hESCs detected using nine methods. The analysis includes TGS RNA-seq dataset from Naïve and Primed hESC generated in this study, as well as NGS RNA-seq dataset from Naïve and Primed hESC. A3: alternative 3' splice site, A5: alternative 5' splice site, ATSS: alternative transcript start site, ATTS: alternative transcript terminated site, ES: exon skipping, IR: intron retention, MEE: mutually exclusive exon, MES: mutually exclusive splicing. **C.** Bar plot showing the number of up-regulated and down-regulated DIU isoforms in Primed hESC with different switching consequences. NMD: nonsense-mediated decay; ORF: open reading frame. **D, E.** UpSet plot showing the number of overlapped DIU genes up-regulated (D) and down-regulated (E) in Primed hESC compared with Naïve hESCs identified by different methods and from the NGS data.

Meanwhile, as suggested by the reviewer, we also added DIU (Differentially Isoform Usage) simulation analysis. The overall simulation process is described in the method section on page 20, lines 734-757. As shown in figure below and Figure

5A, DIU analysis on the simulated naïve and primed hESC datasets revealed that those methods which performed best in the previous isoform construction analysis, namely StringTie2 (guided) and Bambu (guided), also demonstrated the highest accuracy in DIU analysis. The results for DIU simulation can be found on page 12 lines 422-426.

Figure 5

A. Bar plot showing the DIU calling accuracy using simulated data with nine different methods.

12. References:

Text uses written references ie: Wang et al. 2008. However the References section uses numbered references. The reference list does not match the order of references in the text, nor are they in any obvious order (like alphabetical). As currently written the Reference section is unsuitable.

We thank the reviewer for pointing out this problem. We've now revised the references to the format of Nature Communications.

13. YASIM:

This simulator looks useful for testing software performance against a simulated and bespoke transcriptome, thereby allowing various aspects of isoform detection software performance to be examined.

The YASIM Github repo needs improvement to give an overview of what it is and what it can be used for. The current readme covers how to use the various options and parameters but needs the addition of information so a potential user could better understand it's purpose and when it is relevant to use.

We thank the reviewer for this suggestion. We have now revised the documentation and added quick start tutorial for simulated TGS RNA-seq data with YASIM. The documentation page can be accessed via <https://labw.org/yasim-docs/>, and the source code can be found via <https://github.com/WanluLiuLab/yasim/> or <https://pypi.org/project/yasim/>. If the reviewer has further suggestions on our documentation page, we are more than happy to review it.

14. Writing:

- Clarity of writing needs improvement and I highly recommend having it read and

editing by someone for whom English is their first language. Overall the manuscript is quite clear, however there are numerous examples of incorrect or unusual word choices, which should be corrected. To take the first sentence of the abstract as an example: "managed", "leads to a bright future" & "reserved" are all incorrect or unusual word/phrase choices given the context.

We thank the reviewer for pointing out this problem. We have extensively revised the grammar issues in the revised manuscript and have invited our colleague Dr. Aaron Trent Irving to help us edit the language of our manuscript.

15. The manuscript abstract describes the work but doesn't actually inform the reader of the results of the benchmarking. I'd like the authors report the best tool(s) in the abstract.

We thank the reviewer for this suggestion. We've added the description of the best tools (i.e. StringTie2 and Bambu) in the abstract on page 2 lines 36-38.

Minor points:

1. From what I understand the simulated data simulated a C.elegans transcriptome. This is mentioned in the methods but I would like this reported in the results as well to make it clearer to the reader.

We thank the reviewer for this suggestion. We've added this information in the result section on page 5 line 162-164.

2. Line 34-35: The authors state: "more than 90% of genes encode multiple protein isoforms (Wang et al., 2008)". This is not what Wang et al (2008) reported. Wang (08) reported that "92-94% of human genes undergo alternative splicing", but alternative splicing at the RNA level does not correspond 1:1 with alternative protein isoforms. Please fix this claim.

We thank the reviewer for pointing out this issue. We've revised this claim as suggested by the reviewer on page 2 lines 46-47.

3. Figure S2A - shows 5 different "data types", but what they are or mean are not described. Please add this to the figure legend.

We thank the reviewer for this suggestion. We've now separated the results according to different error models as shown in Figure 2 & figure S4. The generation of different types of simulated data have now been revised in result section on page 5-6 lines 187-221.

4. Figure S2B&C - reuses the colour scheme of the data-types from S2A, but I don't

think the colours refer to the data types. This is a bit confusing. Please change the colour scheme.

We thank the reviewer for this suggestion. We have now revised the color scheme to avoid confusions, as shown in Figure S2 and S3.

5. FigS4 - Please overlay the FLAMES results shown in figure 2 so the difference in results when the FLAMES read support threshold is changed can be seen.

We thank the reviewer for this suggestion. We've added the results in Figure 2 as denoted as "FLAMES10".

6. Please provide some summary statistics for the real data used (perhaps in the sup table). Please include information on the number of reads, no. of pass reads, real quality (ie: Q-scores etc) etc etc.

We thank the reviewer for this constructive feedback. We've added the quality control of experimental datasets used in the study in Figure S5, S6, S7 including the average Q-Score (Figure S5A), read length (Figure S5B), GC level (Figure S5C), mapping status (Figure S6A), sequencing errors (Figure S6B), depth (Figure S6C), and read completeness (Figure S7). The details of the metrics can also be found in supplementary table 1 (Table_S1.csv).

7. Please provide the data used to generate Figure 3 in the supplementary information. Ie: The number of isoforms in the different categories.

We thank the reviewer for this suggestion. We've included the data in supplementary table2 (Table_S2.csv).

8. Figure 6 is useful but what were the performance characteristics required for a tool to meet the various categories?

We thank the reviewer for pointing out this problem. We've added the description of how the methods are evaluated and scored to generate the radar plot as shown in the revised Figure 6C, which can be found in the discussion section on pages 23-24 lines 876-892.

Reviewer #2 (Remarks to the Author):

I have read over the revised manuscript and the authors responses to the previous concerns. The authors have done a good job of responding to most of these concerns and the manuscript is much improved.

I still have several concerns regarding the clarity/accuracy of some analysis and the description of some results which do not agree well with the presented data. I have described these remaining issues below:

1. Read completeness in simulated data:

The authors description of the results page 8, lines 277-294 should be revised. There are a number of statements which are not an accurate description of the results. The key comparison is how tools perform compared to 100% complete transcripts.

The authors state "For the Nanopore datasets, most methods, except for Bambu (guided) and StringTie2 (both modes), exhibited inferior performance with less-complete sequencing reads". – I don't think this is correct. For example, StringTie2 does much worse in both precision and sensitivity when reads are incomplete at both ends. I think a more accurate overall summary would be that "most methods exhibited inferior performance with less-complete sequencing reads, although the precision of Bambu (guided) was only slightly impacted".

Regarding Stringtie2 the authors state: "performance remained relatively stable when reads were truncated from the 5' end and its accuracy even improved noticeably when reads were truncated from the 3' end". The 1st claim is not backed up by the data in Fig2E&F, in almost all cases StringTie performance was significantly decreased with 5' truncation (compared to 100% complete transcripts). The later claim should also be revised, 20% 3' truncation decreased StringTie2 performance and while further truncation unexpectedly improved it in some models, as far as I can see, this improvement did not match the original performance on 100% complete RNAs.

"for the PacBio CCS datasets, the precision and sensitivity for StringTie2 (both modes) was not significantly impacted by the changes in read completeness from either the 3' or 5' truncation " – Again this does not agree with the data presented, performance in truncated datasets was significantly reduced compared to 100% complete data.

Page 6 lines 212, 213 state the authors tested a "10% or 20% truncation of 3' end, and 10% or 20% truncation from 5' ends of the TGS reads". However, figure 2 and S4 show a 20% or 40% truncation. Please correct figure or text to correct values.

From what I understand the author's transcript incompleteness model is uniform in the percentage of transcript missing from the 5' and/or 3' end. This is a simplification, as real data incompleteness is not uniform per read and the proportion missing increases as transcript length increases. I suggest the authors note this limitation in discussion and they may also wish to note that a simulator selecting a level of incompleteness based on a distribution would be more realistic.

2. Proportion of novel isoforms and TEs

This section is improved and the hypothesis that many intergenic isoforms represent the detection of expressed transposable elements (TEs) is interesting. However, the author don't present the result that would help pin-point TEs as the cause. This should be achievable by calculating what proportion

of intergenic transcripts overlap TEs. Is it the majority as the author appear to hypothesize?

A second important point here is if these TEs are real detections? TEs are prone to multi-mapping and false +ve detections. Removing mono-exonic TE transcripts implies the author think they are false +ves, but the paragraph starting line 391 suggests the opposite. The authors should be explicit here and if they believe these are real detections, (as a class) then some evidence to support this is required.

In their response to reviews the author appear to agree that the large proportion of novel transcripts found by FLAIR (both forms), and to a lesser extent by Freddie and StringTie2 (both forms), which remain even after removing mono-exonic transcripts likely contain many false positives. However, I don't think the authors actually state this. Instead, they largely imply the opposite in the discussion. I suggest the authors revise their discussion due to these 3 reasons.

1. Bambu (guided) has the highest precision and usually a high sensitivity only just below StringTie2 guided in their simulations.
2. Dong et al 2022 (who should be cited in this discussion section), showed Bambu has the best performance on ground truth spike-in data with higher true positives and lower false positives in their full dataset.
3. Figure 3 shows that StringTie2 (both forms), Freddie and FLAIR (both forms) commonly find ~50-75% of identified isoforms not to be full splice matches in mouse and humans. We should ask ourselves if this seems likely to be true.

Taken together the most logical conclusion is many of these novel transcripts are false +ves, while at the same time Bambu is perhaps quite conservative in calling novel isoforms.

3. Differential isoform usage (DIU):

From reading the method lines (734-757), I'm not clear on how the ground truth DIU data was generated. I would like to think the authors directly created isoform counts that both did and did not correspond to DIU. As shown by Dong et al 2022 (ref 22) tools for calling DIU are not very accurate, so if the authors used experimental data and/or DIU calling tools to decide on a ground truth DIU dataset there is a danger this will give inaccurate results. This could then call into question the accuracy of the DIU benchmarking. The authors should consider how they can better explain what they did and ensure their DIU ground truth results are valid.

4. What is the base simulation that each analysis varies a component of. Transcriptome complexity of 2 is specified in the Methods but the other default parameters don't seem to be (read depth, error rate). Please add.

5. Transcriptome complexity index

Figure S2B shows the actual number of isoforms generated per gene for the "transcriptome complexity index" was a lot lower than targeted. I.e: the mean no. of isoforms per gene was 2.8 isoforms when 5 was targeted, 4.5 when 9 were targeted etc. I suggest it would be more accurate to name the transcriptome complexity index by the actual mean complexity, not the targeted complexity.

Result re: transcriptome complexity index. Lines 256-275. The authors should revise parts of this section to ensure they are focusing on the key results and to avoid duplication. Eg: The author do not mention that Bambu (guided) has the highest precision in all simulated datasets. Eg: The authors

state that TAMA and Bambu (unguided) for Nanopore & TAMA and TALON for PacBio CCS may increase slightly in precision with increased complexity, however these are some of the worst performing tools, so this isn't all that relevant. Eg: When different sequencing models show equivalent results the authors should combine their descriptions instead of describing each individually.

6. Sequencing accuracy section. Again revise to ensure main result is stated and duplicate information is removed. I suggest:

Line 297 change "For the Nanopore datasets" to "For all datasets"

Line 307 – Delete sentence beginning "For PacBio CLR datasets..."

7. Q-scores

Line 346/ Fig S5A – I'm concerned about the accuracy of the Q score values presented. All the nanopore values, except those from Li et al., 2020 are almost certainly too high. For example, the authors own hESC datasets all have mean Q-scores above 20 (see Sup Table 1), which represents a mean accuracy of over 99%. This is unrealistic for R9.4.1 flowcells, which the authors report using. As another example, the H1 and H2 samples are reported to have a Q-score >18 (accuracy >98%), but Fig S6A shows an error rate of ~10% or more. It would appear there is an error in the authors calculations. Can this please be identified and fixed.

8. Read length distributions.

FigS5B – Many PacBio samples show a flat distribution. This suggests that either these samples have been fractionated by length to enrich longer cDNAs, or the length includes sub-reads. The authors should check which of these applies and consider if this alters any of their other results or conclusions. They should also mention the explanation on line 352.

9. Calculating read completeness in real data

The authors state (line 359-61) that most libraries had "high read completeness". I'm not convinced this is supported by Fig S7. There are established metrics for quantifying the proportion of reads that "fully" cover a transcript (ie: see Gleeson et al 2022 in NAR). The authors should calculate this so they have a range of values to report.

10. YASIM online documentation is much improved. I suggest a couple of further improvements:

Please add clear links between the labw.org documentation page and the github page. As much of the necessary information is found on one of the other.

YASIM (when you include the associated LLRG simulators) is a complex program with many options. I recommend expanding the FAQ to enable users to understand what important options do and how to choose the best value for their experiment. Examples would be "How do I specify read completeness?", "How should I decide on the complexity of AS events (Transcriptome Complexity Index)? "How should I choose the Sequencing Depth of Isoforms?" etc.

Minor corrections:

Line 64,65 – remove "the most common PacBio data type". This was previously true but I don't believe it is any more.

Line 414. High similarity between unguided StringTie2 and Bambu does not suggest their results are "robust". Simulated data makes it clear these unguided versions are not very accurate. Please revise.

Line 495. Please add a citation for: "Incomplete reads introduce ambiguity in isoform assignment, posing challenges for accurate identification and analysis".

Line 498. Error rate of ">8-10%" is quite vague. "1-10%" would be reasonable. The authors could also define this from the error rates in their data.

Line 550 – Sentence doesn't make sense. Missing word?

Line 787. Nanopore TGS, please confirm which kit was used to generate libraries as the ligation sequencing kit (LSK) is not usually used for whole transcriptome. Was polyA purification, or rRNA depletion performed on RNA? If so please specify.

Line 859 – FN should be "False Negatives"

Line 869 – Can't ISM be isoforms missing either 5' or 3' exons?

Line 870 – revise to make it clear that for FSM that the 5' start or 3' end of the first and last exon can vary so long as the internal splice site stays the same.

Reviewer #3 (Remarks to the Author):

Su et al have modified their manuscript. While many comments (comments 3, 4 and 5) have clearly been sufficiently addressed, there are serious concerns that remain.

1) question 1: the entire section on experimental data is unfortunately in large parts uninformative and can even misguide readers. This is because there is no ground truth. In reality, extremely high FSM numbers (and low NNC numbers) can be achieved by trusting the annotation to a large extent and by placing little value on the data. The key was to use spike-in data (as suggested by reviewer 1). Without this, the section can do more harm than good. In the worst case, a very high number of qPCRs (say >20) could be used to establish a ground truth, on which the software tools can be compared.

2) By now, there are multiple new pipelines (in order of appearance: Isoquant, ESPRESSO, IsoTools, Bambu, isONform, TAGET). Only Bambu seems to have been incorporated. To be useful, the authors would have to employ a complete and updated tool set. The authors would also have to ensure that all tools are run in an optimal way.

3) The authors mention using published long-read sequencing data. Most of the data collected from publications is outdated (published in 2022 or earlier, thus produced in 2021 or earlier). Long-read sequencing has dramatically changed since then. It is not always clear, which data is used. As a rough guideline, no conclusions should be drawn on data produced before summer of 2022.

4) Regarding the use of -k14 in minimap ... it is not clear that this makes much of a difference from the default. Also, the use of distinct parameters should correlate with the used kits (and therefore quality that is achieved). That would require distinct parameters for different types of cDNA library.

Response to reviewers:

Dear editor:

We thank the reviewers for these constructive comments on our manuscript. Our point-by-point response draft is below. Reviewers' comments are in normal black type and our response is in normal blue type.

Reviewer #3:

1) question 1: the entire section on experimental data is unfortunately in large parts uninformative and can even misguide readers. This is because there is no ground truth. In reality, extremely high FSM numbers (and low NNC numbers) can be achieved by trusting the annotation to a large extent and by placing little value on the data. The key was to use spike-in data (as suggested by reviewer 1). Without this, the section can do more harm than good. In the worst case, a very high number of qPCRs (say >20) could be used to establish a ground truth, on which the software tools can be compared.

We sincerely appreciate the constructive feedback provided by the Reviewer. Following your valuable suggestion, we have incorporated previously published sequins TGS RNA-seq datasets from various sources, including Zhu et al.¹, Wright et al.², and Dong et al.³, into our analysis. These datasets were analyzed alongside the methods we previously benchmarked, as well as with a novel method, IsoQuant, which we tested in both guided and unguided modes.

In Figures S5 and S6 of the revised manuscript, we comprehensively evaluate key characteristics of these sequins datasets, such as the average quality score, read length, GC content, alignment status, depth, and read completeness. Utilizing these sixteen sequins datasets as ground truth, we have assessed the precision and sensitivity of the different methods under study.

Supplementary Figure 5

Supplementary Figure 5. Quality Control for Sequins Datasets.

A-F. Quality control analysis of Sequins TGS RNA-seq datasets was performed on average read quality score (Q-Score from FASTQ file) distribution (A), read length (B), GC level distribution (C), read alignment status (D), type of base-level matches and mismatches (E), and read depths (F). Source data underlying D and F are provided as a Source Data file.

Supplementary Figure 6

Supplementary Figure 6. Read Completeness Distribution of Seq uins Datasets.
Distribution of read completeness and transcript length for different Seq uins TGS RNA-seq datasets.

In the revised Figure 3, we present findings demonstrating that IsoQuant (guided), StringTie2 (guided), and Bambu (guided) consistently exhibit superior sensitivity and precision. In contrast, FLAIR (unguided) appears to identify a considerable number of false positives. This pattern is consistent with the observations from both our simulated and real data. The revised text for this part can be found on pages 9–10 lines 323–370.

Figure 3

Figure 3. Software Performance on Sequins Datasets.

A, B. Precision (A) and sensitivity (B) of the performance of methods tested on previously published TGS RNA-seq datasets spiked-in with Sequins DNA. Individual samples are denoted by gray dots, with gray lines connecting points corresponding to the same sample. The median is indicated by the central line, the boxes delineate the 25th (bottom) and 75th (top) percentiles, and whiskers extend to the furthest points within 1.5 times the interquartile range from the box. **C.** Pie charts display five different isoform types alongside the total counts of isoform events detected by the tested methods across 16 previously published TGS RNA-seq datasets spiked-in with Sequins DNA. FSM, ISM, NIC, and NNC correspond to Full Splice Match, Incomplete Splice Match, Novel In Catalog, and Novel Not In Catalog, respectively. The size of each circle is proportional to the logarithm of the count of isoform events detected by that method divided by logarithm of the sample-level maximal isoform event count for each sample. Hierarchical clustering was applied to the results from these methods. Source data underlying A and B are provided as a Source Data file.

While we concur with the reviewer's perspective on the value of sequins in providing ground truth for our analysis, we also wish to emphasize the importance of the simulated datasets generated by YASIM, as well as the experimental datasets we have collected. YASIM enables the simulation of datasets incorporating a wide range of potential influencing factors, thereby systematically evaluating the performance of different methods. However, it is important to note that both simulated datasets and sequins may not fully capture the complexity inherent in experimental data. Therefore, in our revised manuscript, we have included results from YASIM-simulated, sequins TGS RNA-seq, and experimental datasets across various species and sequencing platforms. Our goal is to provide a comprehensive evaluation of the performance of different tools under diverse conditions. We believe that this holistic approach significantly strengthens the robustness and validity of our findings.

2) By now, there are multiple new pipelines (in order of appearance: Isoquant, ESPRESSO, IsoTools, Bambu, isONform, TAGET). Only Bambu seems to have been incorporated. To be useful, the authors would have to employ a complete and updated tool set. The authors would also have to ensure that all tools are run in an optimal way.

We are grateful for your constructive feedback. Following your suggestions, we have included both the guided and unguided versions of IsoQuant in our analysis of simulated, sequins, and experimental datasets. Our findings indicate that IsoQuant (guided) exhibits the best performance in terms of both sensitivity and precision, while the unguided version of IsoQuant also demonstrates high precision among the unguided methods. We have updated our conclusions throughout the manuscript to reflect these findings.

Regarding the other methods you suggested, we had initially tested ESPRESSO⁴ in our previous manuscript version. However, we decided not to include it due to its extremely high memory consumption. Furthermore, we noted that isONform⁵ is specifically designed for ONT cDNA sequencing data, while TAGET⁶ and IsoTools⁷ are tailored for analyzing full-length transcripts from PacBio Iso-seq data. Our benchmark analysis primarily focuses on methods compatible with both PacBio and Nanopore data. Consequently, these tools did not align with the scope of our study. We have discussed these tools in our revised discussion section on page 14 lines 506–512.

While we acknowledge the importance of running tools in an optimal manner, we believe it is unrealistic to test every parameter in each software to find the optimal setting. This approach would require exploring a vast array of possible combinations and entail significant time and resource commitments. Additionally, since our goal is to provide unbiased suggestions of tools for untrained users, we have used the default parameter settings for all methods we tested, as previously suggested⁸. Specific reasons for choosing default parameters include: (1) Default parameter settings are usually recommended by the software developers, having been established after their own testing and optimization; (2) Most users are not familiar with the implementation details of these tools and often find it challenging to determine the optimal parameters for their specific datasets. The default parameters are often what most users would apply, thereby ensuring that our study reflects real-world usage scenarios; (3) High-quality software should exhibit robust performance under default settings, regardless of dataset variability; (4) In line with previously published guidelines for benchmarking computational methods and consistent with prior benchmark studies, default parameters or those recommended in the software documentation are commonly used^{9–14}. To acknowledge this

point, we have now revised our results and discussion on page 7 lines 234–236, and page 18 lines 668–671.

3) The authors mention using published long-read sequencing data. Most of the data collected from publications is outdated (published in 2022 or earlier, thus produced in 2021 or earlier). Long-read sequencing has dramatically changed since then. It is not always clear, which data is used. As a rough guideline, no conclusions should be drawn on data produced before summer of 2022.

We respectfully disagree with the assertion that the data we have collected should be considered “outdated” and therefore not suitable for drawing conclusions. The specific “dramatic changes” in long-read sequencing technology referred to by the reviewer (presumably the development of Nanopore R10.4) were not clearly defined. The datasets we used primarily employ Nanopore R9.4/R9.4.1 sequencing.

It is important to recognize that new techniques often require a period of validation and analysis before gaining widespread acceptance in the scientific community. Our review of recent publications in the NCBI public database indicates that a significant number of studies published after the summer of 2022 still utilize the R9.4/R9.4.1 version of Nanopore sequencing. Examples include datasets from [[https://www.ncbi.nlm.nih.gov/sra/SRX19788850\[accn\]](https://www.ncbi.nlm.nih.gov/sra/SRX19788850[accn])], [[https://www.ncbi.nlm.nih.gov/sra/SRX17942557\[accn\]](https://www.ncbi.nlm.nih.gov/sra/SRX17942557[accn])], and [[https://www.ncbi.nlm.nih.gov/sra/SRX19215447\[accn\]](https://www.ncbi.nlm.nih.gov/sra/SRX19215447[accn])]. Additionally, the algorithms we benchmarked, as well as other methods mentioned in your second comment, were mostly developed using datasets published before the summer of 2022, primarily based on Nanopore R9.4/R9.4.1 data.

These observations suggest that the newer version of Nanopore sequencing has not yet been widely adopted by the scientific community, and many researchers continue to rely on the older, more established versions. As such, the performance evaluations of various software tools on these datasets remain highly relevant and offer valuable guidance for researchers. This is particularly true until the newer sequencing approaches are thoroughly validated and novel methods are developed based on systematic investigations of large-scale datasets generated by these new techniques.

4) Regarding the use of -k14 in minimap ... it is not clear that this makes much of a difference from the default. Also, the use of distinct parameters should correlate with the used kits (and therefore quality that is achieved). That would require distinct parameters for different types of cDNA library.

We appreciate the reviewer’s attention to the details of our methodology. In our methods section, we indeed specify that we used three distinct combinations of parameters to align different types of long-read sequencing data (on page 22 lines 831–835). These choices were made in accordance with the recommendations provided in the documentation of minimap2 (<https://github.com/lh3/minimap2#map-long-splice>).

To clarify, the “-k 14” parameter was specifically employed for the five Nanopore direct RNA sequencing datasets of *C. elegans* that we analyzed. This adjustment was made following the

guidance from the minimap2 documentation, which suggests using a smaller k-mer size to increase the sensitivity for detecting splice junctions, particularly for the first or the last exons in noisy Nanopore direct RNA-seq reads.

Reviewer #2

I have read over the revised manuscript and the authors responses to the previous concerns. The authors have done a good job of responding to most of these concerns and the manuscript is much improved.

We sincerely thank the reviewer for their careful reading of our manuscript and for acknowledging the improvements we have made.

I still have several concerns regarding the clarity/accuracy of some analysis and the description of some results which do not agree well with the presented data. I have described these remaining issues below:

1. Read completeness in simulated data:

The authors description of the results page 8, lines 277-294 should be revised. There are a number of statements which are not an accurate description of the results. The key comparison is how tools perform compared to 100% complete transcripts.

The authors state “For the Nanopore datasets, most methods, except for Bambu (guided) and StringTie2 (both modes), exhibited inferior performance with less-complete sequencing reads”. – I don’t think this is correct. For example, StringTie2 does much worse in both precision and sensitivity when reads are incomplete at both ends. I think a more accurate overall summary would be that “most methods exhibited inferior performance with less-complete sequencing reads, although the precision of Bambu (guided) was only slightly impacted”.

Regarding the performance of various methods in relation to read completeness, we have updated our analysis to include IsoQuant in both guided and unguided modes. The new results indicate that the guided mode of IsoQuant consistently maintained the highest and most stable precision and sensitivity across all tested datasets, even when reads were truncated (as shown in Figures 2E,F; S4E,F, and Figures below).

Figure 2

Figure 2. Accuracy of software performance on simulated datasets of Nanopore R103, Pacbio SEQUEL CLR, and CCS.
E, F. Precision (E) and sensitivity (F) of the performance of tested methods obtained on simulated data of Nanopore R103, Pacbio SEQUEL CLR, and CCS with different read completeness (0.0_0.0: 100% complete, 0.1_0.1: 10% truncated from both ends; 0.2_0.2: 20% truncated from both ends; 0.2_0.0: 20% truncated from 5' end; 0.4_0.0: 40% truncated from 5' end; 0.0_0.2: 20% truncated from 3' end; 0.0_0.4: 40% truncated from 3' end, three replicates for each sequencing platform, n=63 in total). N and P represent datasets generated from the Nanopore and Pacbio platforms, respectively. All reported values are expressed as means, with Standard Deviation (SD) detailed in the Source Data file.

Supplementary Figure 4

Supplementary Figure 4. Accuracy of Software Performance on Simulated Datasets of Nanopore R94, Pacbio RSII CLR, and CCS.
E, F. Precision (E) and sensitivity (F) of the performance of tested methods obtained on simulated data of Nanopore R94, Pacbio RSII CLR, and CCS with different read completeness (0.0_0.0: 100% complete, 0.1_0.1: 10% truncated from both ends; 0.2_0.2: 20% truncated from both ends; 0.2_0.0: 20% truncated from 5' end; 0.4_0.0: 40% truncated from 5' end; 0.0_0.2: 20% truncated from 3' end; 0.0_0.4: 40% truncated from 3' end, three replicates for each sequencing platform, n=63 in total). N and P represent datasets generated from the Nanopore and Pacbio platforms, respectively. All reported values are expressed as means, with Standard Deviation (SD) detailed in the Source Data file.

Based on these new results, we have revised the statement as:

“We also evaluated software performance under varying degrees of read completeness. In all Nanopore and PacBio datasets, most methods exhibited inferior performance with less-complete sequencing reads, although the precision of IsoQuant (guided) and Bambu (guided) were only slightly impacted. In addition, IsoQuant (guided) also consistently demonstrated the highest and most stable sensitivity across different sequencing platforms (Figure 2E, F; S4E, F).”

The revised text could be found on page 8 lines 274–279.

Regarding Stringtie2 the authors state: “performance remained relatively stable when reads were truncated from the 5' end and its accuracy even improved noticeably when reads were truncated from the 3' end”. The 1st claim is not backed up by the data in Fig2E&F, in almost all cases StringTie performance was significantly decreased with 5' truncation (compared to 100% complete transcripts). The later claim should also be revised, 20% 3' truncation decreased StringTie2 performance and while further truncation unexpectedly improved it in some models,

as far as I can see, this improvement did not match the original performance on 100% complete RNAs.

“for the PacBio CCS datasets, the precision and sensitivity for StringTie2 (both modes) was not significantly impacted by the changes in read completeness from either the 3’ or 5’ truncation ” – Again this does not agree with the data presented, performance in truncated datasets was significantly reduced compared to 100% complete data.

As suggested, we have now revised the statement as:

“StringTie2 (in both modes) exhibited reduced precision and sensitivity as reads became incomplete from both the 3’ and 5’ ends in all datasets. Compared to fully complete reads, StringTie2 showed reduced precision and sensitivity with 20% truncation from the either 3’ or 5’ end, while further truncation unexpectedly improved it in certain models (Figure 2E, F; S4E, F).”

The revised text could be found on page 8 lines 284–288.

Page 6 lines 212, 213 state the authors tested a “10% or 20% truncation of 3’ end, and 10% or 20% truncation from 5’ ends of the TGS reads”. However, figure 2 and S4 show a 20% or 40% truncation. Please correct figure or text to correct values.

We’ve corrected this to 20% or 40% truncation in our revised manuscript which can be found on page 6 lines 216–219.

From what I understand the author’s transcript incompleteness model is uniform in the percentage of transcript missing from the 5’ and/or 3’ end. This is a simplification, as real data incompleteness is not uniform per read and the proportion missing increases as transcript length increases. I suggest the authors note this limitation in discussion and they may also wish to note that a simulator selecting a level of incompleteness based on a distribution would be more realistic.

We thank the reviewer for these comments, as suggested, we have now discussed this limitation in the discussion section as:

“Another potential limitation of our study is the use of a uniform model by YASIM to simulate read completeness. This approach simplifies the simulation compared to the non-uniform incompleteness observed in real data, where missing portions are likely to be pronounced in longer transcripts. Future iterations of YASIM will explore the integration of a more sophisticated distribution model to better simulate the variation in incomplete reads.”

The revised text could be found on page 18 lines 663–668.

2. Proportion of novel isoforms and TEs

This section is improved and the hypothesis that many intergenic isoforms represent the detection of expressed transposable elements (TEs) is interesting. However, the author don’t present the result that would help pin-point TEs as the cause. This should be achievable by calculating what proportion of intergenic transcripts overlap TEs. Is it the majority as the author appear to hypothesize?

We have conducted a thorough examination of the overlap between TE sequences and mono-exonic transcripts identified by the methods that previously produced a significant proportion of intergenic isoforms. Overall, 54%–60% of intergenic mono-exonic isoforms called by FLAIR (in both modes), Bambu (guided), and Freddie overlapped with TE regions.

A second important point here is if these TEs are real detections? TEs are prone to multi-mapping and false +ve detections. Removing mono-exonic TE transcripts implies the author think they are false +ves, but the paragraph starting line 391 suggests the opposite. The authors should be explicit here and if they believe these are real detections, (as a class) then some evidence to support this is required.

We appreciate the opportunity to clarify our analysis regarding transposable elements (TEs). We understand that TEs are complex genomic features that can complicate mapping and lead to potential multi-mapping and false positive detections. The reference annotation utilized by SQANTI3 is based on the protein-coding genes and does not include annotations specific to TE regions. Consequently, it classifies TE-related transcripts as intergenic, which does not necessarily imply that these transcripts are artefactual.

To avoid confusion, we have revised this statement as:

“Mono-exonic transcripts were excluded from classification since they lack splice junctions and may introduce bias into the results when analyzing their performance over protein-coding genes.” (on page 11 lines 405–407)

To assess the validity of these TE detections, we first selected the intergenic mono-exonic isoforms called by FLAIR (in both modes), Bambu (guided), and Freddie. We then performed random permutations over the intergenic regions, generating a set of random shuffled regions for each call as control for genomic average. Then, we tested the enrichment of the test regions (intergenic mono-exonic isoforms and its paired random shuffle regions) over TEs. This analysis revealed a significant enrichment of intergenic mono-exonic isoforms over TE regions. We have updated this analysis in Figure S11B (as shown below), and have made corresponding revisions in our main text as follows:

“Compared to randomly shuffled genomic regions, the mono-exonic isoforms detected by FLAIR (in both modes), Bambu (unguided), and Freddie showed a significant enrichment over TE regions (54% to 60% vs. 47%) (Figure S11B). Visualization of representative intergenic transcripts specific to naïve hESCs demonstrated a high degree of overlap with previously reported functional TE loci such as LTR5Hs and HERVH/LTR7Y, suggesting the capacity of these methods to detect TEs (Figure S11C, D).” (on page 12 lines 440–447)

Supplementary Figure 11

Supplementary Figure 11. Isoform Classification for Mono-exonic Transcripts.

B. The odds ratio (observed over expected) of overlapping regions between TEs and intergenic mono-exon transcripts identified by Bambu (guided), FLAIR (guided), FLAIR (unguided), Freddie compared with random shuffled intergenic regions. Statistical analysis was performed with Fisher exact test ($*P < 0.05$). Source data underlying B are provided as a Source Data file.

In their response to reviews the author appear to agree that the large proportion of novel transcripts found by FLAIR (both forms), and to a lesser extent by Freddie and StringTie2 (both forms), which remain even after removing mono-exonic transcripts likely contain many false positives. However, I don't think the authors actually state this. Instead, they largely imply the opposite in the discussion. I suggest the authors revise their discussion due to these 3 reasons.

1. Bambu (guided) has the highest precision and usually a high sensitivity only just below StringTie2 guided in their simulations.
2. Dong et al 2022 (who should be cited in this discussion section), showed Bambu has the best performance on ground truth spike-in data with higher true positives and lower false positives in their full dataset.
3. Figure 3 shows that StringTie2 (both forms), Freddie and FLAIR (both forms) commonly find ~50-75% of identified isoforms not to be full splice matches in mouse and humans. We should ask ourselves if this seems likely to be true.

Taken together the most logical conclusion is many of these novel transcripts are false +ves while at the same time Bambu is perhaps quite conservative in calling novel isoforms.

We are grateful for the reviewer's insightful observations and the opportunity to refine our discussion. In our revised manuscript, we have expanded our benchmark analysis to include precision and sensitivity assessments of various methods using 16 previously published sequins TGS RNA-seq datasets, as well as incorporating IsoQuant in both guided and unguided modes.

Our updated findings with simulated, sequins, and real data, indicated that IsoQuant (guided) and Bambu (guided) generally achieved top performance in precision, and IsoQuant (guided) and StringTie2 (guided) exhibited best sensitivity. Freddie and TAMA showed lowest precision, while Freddie, FLAME10, and TAMA showed lowest sensitivity.

For the analysis with sequins data, Freddie was not included due to its prohibitive computational demands, possibly stemming from the extreme depths of the sequins datasets. As depicted in the revised Figure 3, sequins data demonstrated that IsoQuant (guided), StringTie2 (guided), and Bambu (guided) all exhibit commendable sensitivity and precision, while FLAIR (in both

modes), StringTie2 (unguided), UNAGI, TALON, and TAMA exhibited low precision. FLAIR in both modes, TALON, and StringTie2 (unguided) reported a significant number of NIC or NNC isoforms, which suggests a potential for a high level of false positives. Conversely, Bambu (unguided) and IsoQuant (unguided) primarily detected FSM or ISMs, indicating a more conservative approach in the identification of novel isoforms.

Based on the new results, we have now clarified this point for sequins data in our main text as:

“Reviewing the composition of isoform types detected by each method, guided methods generally identified a higher number of FSM isoforms compared to unguided methods, with IsoQuant (guided) and Bambu (guided) detecting the largest proportion of FSM, followed by StringTie2 (guided) and FLAMES3 (Figure 3C). FLAIR (in both modes), TALON, and StringTie2 (unguided) reported a significant amount of NIC or NNC, indicating a potentially high level of false positives. Bambu (unguided) and IsoQuant (unguided) detected mostly FSM or ISMs, suggesting that these two methods may be quite conservative in calling novel isoforms.” (on page 10 lines 358–365)

Figure 3

Figure 3. Software Performance on Sequins Datasets.

A, B. Precision (A) and sensitivity (B) of the performance of methods tested on previously published TGS RNA-seq datasets spiked-in with Sequins DNA. Individual samples are denoted by gray dots, with gray lines connecting points corresponding to the same sample. The median is indicated by the central line, the boxes delineate the 25th (bottom) and 75th (top) percentiles, and whiskers extend to the furthest points within 1.5 times the interquartile range from the box. **C.** Pie charts display five different isoform types alongside the total counts of isoform events detected by the tested methods across 16 previously published TGS RNA-seq datasets spiked-in with Sequins DNA. FSM, ISM, NIC, and NNC correspond to Full Splice Match, Incomplete Splice Match, Novel In Catalog, and Novel Not In Catalog, respectively. The size of each circle is proportional to the logarithm of the count of isoform events detected by that method divided by logarithm of the sample-level maximal isoform event count for each sample. Hierarchical clustering was applied to the results from these methods. Source data underlying A and B are provided as a Source Data file.

We have also extensively revised the description for real data as:

“Most methods detected a major proportion of FSM and ISM, except for FLAIR (unguided). Consistent with the sequins datasets, Bambu (guided) and IsoQuant (guided) reported the highest proportion of FSMs, while FLAIR (unguided) detected the largest number of isoforms classified as NNC, potentially due to a high number of false positives called (Figure 4A, B). TALON and FLAIR (guided) detected a large proportion of NNC and NIC in sequins datasets, respectively. The performance of TALON on real data exhibited comparable levels of FSM and ISM detection as FLAMES3 and IsoQuant (unguided), whereas FLAIR (guided) showed similar proportions of FSM and ISM to StringTie2 (guided) (Figure 4A, B). This difference could potentially be explained by the varying sequencing depths for sequins and real data, as our simulated data suggested TALON and FLAIR (guided) may detect more false positives with increasing sequencing depths (Figure 2A; S4A). Bambu (unguided) tends to detect a large proportion of ISM in both sequins and real datasets (Figure 4A, 3C). NNC and NIC detected by Freddie remain questionable, given its low precision on simulated data. We also performed a comparative analysis of the results obtained from different methods and quantitatively assessed their similarities using Jaccard statistics, which represent the pairwise overlapping of detected isoforms at the base-pair resolution for each method. StringTie2 (in both modes) and FLAIR (guided) exhibited a notable level of concordance, which may be partly due to the higher number of NIC categorizations by these methods (Figure 4, S10).” (on pages 11–12 lines 407–427).

As suggested by the reviewer, Dong et al., 2023 have been cited in the discussion section as:

“In a recent study, Dong et al. applied sequins TGS RNA-seq datasets to evaluate the performance of six methods and identified Bambu and StringTie2 as the best isoform detection tools.” (on page 17 lines 623–625).

3. Differential isoform usage (DIU):

From reading the method lines (734-757), I'm not clear on how the ground truth DIU data was generated. I would like to think the authors directly created isoform counts that both did and did not correspond to DIU. As shown by Dong et al 2022 (ref 22) tools for calling DIU are not very accurate, so if the authors used experimental data and/or DIU calling tools to decide on a ground truth DIU dataset there is a danger this will give inaccurate results. This could then call into question the accuracy of the DIU benchmarking. The authors should consider how they can better explain what they did and ensure their DIU ground truth results are valid.

We thank the reviewer for bringing this to our attention and apologize for any lack of clarity in our initial description of the Differential Isoform Usage (DIU) simulation process. To elucidate our methodology, we have provided a detailed explanation below:

“Initially, 96 DIU genes were identified from the count matrices of actual Naïve and Primed hESC TGS RNA-seq datasets. Isoform counts for these genes were then extracted from the original count matrices. An additional 100 genes were randomly selected from the reference annotation, with the exclusion of the 96 DIU genes, and were incorporated into the count matrices. The resulting isoform count matrix, encompassing 196 genes, was used to create expression profiles in YASIM. Subsequently, four simulated TGS RNA-seq datasets were generated employing the Nanopore R103 error model. These simulated reads were aligned and processed using eleven benchmarked methods, followed by salmon quantification and DIU

calling using isoformSwitchAnalyzeR. Meanwhile, the corresponding GTF annotation of the 196 genes plus the four simulated datasets were also processed by salmon and isoformSwitchAnalyzeR, and the DIU gene list obtained were served as ground truth. The eleven resulting DIU gene lists were then compared with the ground truth DIUs to calculate the respective precision and sensitivity values.”

To better illustrate this process, we have included an additional figure, Figure S12, in the revised manuscript. This figure provides a visual framework of the DIU simulation process, which we believe will enhance understanding and verification of our approach.

Supplementary Figure 12

Supplementary Figure 12. Schematic Workflow for DIU Simulation.

This figure presents the process for simulating Differential Isoform Usage (DIU). Initially, 96 DIU genes were identified from the count matrices of actual Naive and Primed hESC TGS RNA-seq datasets. Isoform counts for these genes were then extracted from the original count matrices. An additional 100 genes were randomly selected from the reference annotation, with the exclusion of the 96 DIU genes, and were incorporated into the count matrices. The resulting isoform count matrix, encompassing 196 genes, was used to create expression profiles in YASIM. Subsequently, four simulated TGS RNA-seq datasets were generated employing the Nanopore R103 error model. These simulated reads were aligned and processed using eleven benchmarked methods, followed by salmon quantification and DIU calling using isoformSwitchAnalyzerR. Meanwhile, the corresponding GTF annotation of the 196 genes plus the four simulated datasets were also processed by salmon and isoformSwitchAnalyzerR, and the DIU gene list obtained were served as ground truth. The eleven resulting DIU gene lists were then compared with the ground truth DIUs to calculate the respective precision and sensitivity values.

We also appreciate the reviewer for providing insightful suggestions concerning Dong et al.'s study on DIU analysis. In Dong et al.'s study, they employed a uniform upstream quantification pipeline (salmon) and assessed the performance of various downstream tools for DIU calling. Their analysis suggested very little consistency among the methods, and no clear front-runner could be identified for downstream DIU calling.

In contrast, our study aims to benchmark the performance of different upstream isoform detection tools in the context of downstream DIU calling. Therefore, we applied different isoform detection tools in the upstream analysis and conducted DIU calling using a uniform downstream pipeline consisting of salmon for quantification and isoformSwitchR for DIU calling on the four simulated datasets. Given the difference in research scope, we believe our approach to DIU simulation aligns with our study's objectives.

To provide further clarity on this point, we have revised the main text and discussion accordingly, as shown below:

“As previously reported, there was no clear front-runner for downstream DIU analysis. Therefore, we adopted a consistent downstream workflow using isoformSwitchAnalyzeR for DIU calling, with the isoforms detected by different methods as input.” (in results section, on page 12, lines 454–458)

“While we benchmarked the performance of various upstream isoform detection tools on downstream DIU calling, Dong et al. tested different tools specifically designed for downstream DIU calling and noted limited consistency between methods.” (in discussion section, on page 6, lines 607–609)

4. What is the base simulation that each analysis varies a component of. Transcriptome complexity of 2 is specified in the Methods but the other default parameters don't seem to be (read depth, error rate). Please add.

The default depth was set as 20, complexity as 2, and the error rate as 15% (i.e., 0.85 accuracy) except for those involving varying levels of depths or error rates. The default setting for read truncation was set to 0 and the default setting for reference annotation completeness was set to 100% unless specified. “--low_cutoff” was set for 0.01 for PBSIM2 LLRG and 1 for PBSIM3 LLRG with “--high_cutoff_ratio” set as 200.

We've added the information to our revised manuscript on page 20, lines 747–752.

5. Transcriptome complexity index

Figure S2B shows the actual number of isoforms generated per gene for the “transcriptome complexity index” was a lot lower than targeted. I.e: the mean no. of isoforms per gene was 2.8 isoforms when 5 was targeted, 4.5 when 9 were targeted etc. I suggest it would be more accurate to name the transcriptome complexity index by the actual mean complexity, not the targeted complexity.

As suggested by the reviewer, we've now added the values on the X-axis label in our revised Figure 2C, D and Figure S4C,D (as shown below) to represent the actual mean complexity.

Figure 2

Figure 2. Accuracy of software performance on simulated datasets of Nanopore R103, Pacbio SEQUEL CLR, and CCS. C, D. Precision (C) and sensitivity (D) of the performance of tested methods obtained on simulated data of Nanopore R103, Pacbio SEQUEL CLR, and CCS across different transcriptome complexity indices (1, 3, 5, 7, 9; values in parentheses denote the actual mean number of isoforms per gene simulated), with three replicates per sequencing platform (n=45 in total). N and P represent datasets generated from the Nanopore and Pacbio platforms, respectively. All reported values are expressed as means, with Standard Deviation (SD) detailed in the Source Data file.

Supplementary Figure 4

Supplementary Figure 4. Accuracy of Software Performance on Simulated Datasets of Nanopore R94, Pacbio RSII CLR, and CCS. C, D. Precision (C) and sensitivity (D) of the performance of tested methods obtained on simulated data of Nanopore R103, Pacbio SEQUEL CLR, and CCS across different transcriptome complexity indices (1, 3, 5, 7, 9; values in parentheses denote the actual mean number of isoforms per gene simulated), with three replicates per sequencing platform (n=45 in total). N and P represent datasets generated from the Nanopore and Pacbio platforms, respectively. All reported values are expressed as means, with Standard Deviation (SD) detailed in the Source Data file.

Result re: transcriptome complexity index. Lines 256-275. The authors should revise parts of this section to ensure they are focusing on the key results and to avoid duplication. Eg: The author do not mention that Bambu (guided) has the highest precision in all simulated datasets. Eg: The authors state that TAMA and Bambu (unguided) for Nanopore & TAMA and TALON for PacBio CCS may increase slightly in precision with increased complexity, however these are some of the worst performing tools, so this isn't all that relevant. Eg: When different sequencing models show equivalent results the authors should combine their descriptions instead of describing each individually.

As suggested by the reviewer, we have extensively revised this part (on pages 7–8 lines 259–273).

6. Sequencing accuracy section. Again revise to ensure main result is stated and duplicate information is removed. I suggest:

Line 297 change “For the Nanopore datasets” to “For all datasets”

Line 307 – Delete sentence beginning “For PacBio CLR datasets...”.

We've revised our manuscript as suggested. The revised text could be found on page 8 lines 291–302.

7. Q-scores

Line 346/ Fig S5A – I'm concerned about the accuracy of the Q score values presented. All the nanopore values, except those from Li et al., 2020 are almost certainly too high. For example, the authors own hESC datasets all have mean Q-scores above 20 (see Sup Table 1), which represents a mean accuracy of over 99%. This is unrealistic for R9.4.1 flowcells, which the authors report using. As another example, the H1 and H2 samples are reported to have a Q-score >18 (accuracy >98%), but Fig S6A shows an error rate of ~10% or more. It would appear there is an error in the authors calculations. Can this please be identified and fixed.

We thank the reviewer for this point. We have fixed our code for visualizing the Q score, and the revised figure for both sequins and experimental data is shown below (and in Figure S5A, S7A).

Supplementary Figure 5

Supplementary Figure 5. Quality Control for Sequins Datasets.

A-F. Quality control analysis of Sequins TGS RNA-seq datasets was performed on average read quality score (Q-Score from FASTQ file) distribution (A).

Supplementary Figure 7

Supplementary Figure 7. GC Level, Read length, and Read Quality Distribution of Experimental Datasets.

A. Quality control analysis of experimental datasets was performed on average read quality score (Q-Score from FASTQ file) distribution. N.A. in A represents Q-Score not available. The publicly available experimental datasets originate from the following sources: C1: L1 larval stage of *C. elegans*, C2: mix stage of *C. elegans*, C3: young adult stage of *C. elegans*; C4: Wildtype *C. elegans* total RNA replicate 1, C5: Wildtype *C. elegans* total RNA replicate 2, D1: *Drosophila*, D2: *Drosophila* testis, M1: mouse activated CD8 T cell, M2: mouse naïve CD8 T cell, M3: mouse retinal cells (control), M4: mouse retinal cells (glaucomatous), M5: mouse CD4SP cells, M6: mouse CD8SP cells, M7: mouse neural stem cells (E15.5), M8: mouse neural stem cells (P1.5), M9: mouse cerebral cells, M10: mouse hippocampus cells. H1: human Beta cells, H2: human Beta cells treated with cytokines, H3: human Hela cells, H4: human iPSC cells. The TGS RNA-seq dataset on Naive and Primed hESCs was generated in this study.

8. Read length distributions.

FigS5B – Many PacBio samples show a flat distribution. This suggests that either these samples have been fractionated by length to enrich longer cDNAs, or the length includes sub-reads. The authors should check which of these applies and consider if this alters any of their other results or conclusions. They should also mention the explanation on line 352.

Upon further review of the relevant literature (Weirather et al., 2017)¹⁵, we have verified that the PacBio CLR libraries are indeed size-selected. This procedure results in a greater proportion of longer reads, as the libraries are fractionated by length to specifically enrich for longer cDNAs. We have included this information in our manuscript to provide context to the observed flat distribution of read lengths (on page 11 lines 385–388). Upon careful consideration, we have concluded that this size-selection process for PacBio libraries does not significantly impact the overall results and conclusions of our study.

9. Calculating read completeness in real data

The authors state (line 359-61) that most libraries had “high read completeness”. I’m not convinced this is supported by Fig S7. There are established metrics for quantifying the proportion of reads that “fully” cover a transcript (ie: see Gleeson et al 2022 in NAR). The authors should calculate this so they have a range of values to report.

We are thankful for the reviewer’s constructive feedback regarding our assessment of read completeness. Following your recommendation, we have adopted a more standardized metric for quantifying the proportion of reads that fully cover a transcript, as detailed in Gleeson et al., 2022, in *Nucleic Acids Research*¹⁶. Consequently, we have updated Figure S7 to align with the visualization approach used in Figure 1E of Gleeson et al. for both our sequins and experimental datasets. The corresponding new figures are now presented as revised Figures S6 and S9 (as shown below). These changes have allowed us to more accurately measure and describe read completeness across our datasets.

In light of these revised figures and the application of established metrics, we have adjusted the manuscript text concerning read completeness.

Sequins datasets: “*The sequins datasets exhibited an average coverage ranging from 300X to 13,000X (Figure S5F and Table S1), with lower read completeness representing samples with lower read quality (Figure S6).*” (on page 9 lines 331–333)

Experimental datasets: “*Read completeness analysis revealed certain datasets (D2, M5, M6, M7, M8, H3, and H4) exhibited poor read completeness for transcript longer than 5kb (Figure S9).*” (on page 11 lines 395–397)

Supplementary Figure 6

Supplementary Figure 6. Read Completeness Distribution of Seq uins Datasets.
Distribution of read completeness and transcript length for different Se quins TGS RNA-seq datasets.

Supplementary Figure 9

Supplementary Figure 9. Read Completeness Distribution of Experimental Datasets.

Distribution of read completeness and transcript length for different experimental datasets. The publicly available experimental datasets originate from the following sources: C1: L1 larval stage of *C. elegans*, C2: mix stage of *C. elegans*, C3: young adult stage of *C. elegans*; C4: Wildtype *C. elegans* total RNA replicate 1, C5: Wildtype *C. elegans* total RNA replicate 2, D1: *Drosophila*, D2: *Drosophila* testis, M1: mouse activated CD8 T cell, M2: mouse naive CD8 T cell, M3: mouse retinal cells (control), M4: mouse retinal cells (glaucomatous), M5: mouse CD4SP cells, M6: mouse CD8SP cells, M7: mouse neural stem cells (E15.5), M8: mouse neural stem cells (P1.5), M9: mouse cerebral cells, M10: mouse hippocampus cells. H1: human Beta cells, H2: human Beta cells treated with cytokines, H3: human Hela cells, H4: human iPSC cells. The TGS RNA-seq dataset on Naive and Primed hESCs was generated in this study.

10. YASIM online documentation is much improved. I suggest a couple of further improvements: Please add clear links between the labw.org documentation page and the github page. As much of the necessary information is found on one of the other.

We have added this as suggested.

YASIM (when you include the associated LLRG simulators) is a complex program with many options. I recommend expanding the FAQ to enable users to understand what important options do and how to choose the best value for their experiment. Examples would be “How do I specify read completeness?”, “How should I decide on the complexity of AS events (Transcriptome Complexity Index)?” “How should I choose the Sequencing Depth of Isoforms?” etc.

We have now provided an FAQ session to better help users with potential problems they might encounter. The FAQ session can be found at <https://labw.org/yasim-docs/src/faq.html>

Minor corrections:

Line 64,65 – remove “the most common PacBio data type”. This was previously true but I don’t believe it is any more.

We have revised this part as suggested.

Line 414. High similarity between unguided StringTie2 and Bambu does not suggest their results are “robust”. Simulated data makes it clear these unguided versions are not very accurate. Please revise.

Following the inclusion of IsoQuant in our analysis, we observed that the methods demonstrating the highest similarity are now StringTie2 (guided), StringTie2 (unguided), and FLAIR (guided). We agree with you that similarity in results alone is not a definitive indicator of robustness. Therefore, we have revised this part as:

“We also performed a comparative analysis of the results obtained from different methods and quantitatively assessed their similarities using Jaccard statistics, which represent the pairwise overlapping of detected isoforms at the base-pair resolution for each method. StringTie2 (in both modes) and FLAIR (guided) exhibited a notable level of concordance, which may be partly attributed to the higher number of NIC categorizations made by these methods (Figure 4, S10). However, whether these NICs are true positives remains in doubt.”

The revised text could be found on page 12 lines 421–427.

Line 495. Please add a citation for: “Incomplete reads introduce ambiguity in isoform assignment, posing challenges for accurate identification and analysis”.

As suggested, we have now added a citation (Kanitz et al., 2015, Genome Biology) to this statement¹⁷.

Line 498. Error rate of “>8-10%” is quite vague. “1-10%” would be reasonable. The authors could also define this from the error rates in their data.

As suggested, we have now revised this as 1-10%.

Line 550 – Sentence doesn’t make sense. Missing word?

We have now revised this sentence as:

“Additionally, inconsistent trends were observed in the results obtained by StringTie2 and IsoQuant (guided) from data generated by PBSIM2 and PBSIM3 reads that were truncated from the 5’ end.”

Line 787. Nanopore TGS, please confirm which kit was used to generate libraries as the ligation sequencing kit (LSK) is not usually used for whole transcriptome. Was polyA purification, or rRNA depletion performed on RNA? If so please specify.

We have collected additional information from previously published papers regarding library preparation kits, flow cells, technology, and basecaller into the revised Table S1.

Line 859 – FN should be “False Negatives”

We have revised this part as suggested.

Line 869 – Can't ISM be isoforms missing either 5' or 3' exons?

We have corrected this to 5' or 3' exons (on page 25 line 955).

Line 870 – revise to make it clear that for FSM the 5' start or 3' end of the first and last exon can vary so long as the internal splice site stays the same.

We have revised this part as suggested (on page 25 lines 952-954)

References

1. Zhu, C. *et al.* Single-molecule, full-length transcript isoform sequencing reveals disease-associated RNA isoforms in cardiomyocytes. *Nat Commun* **12**, 4203 (2021).
2. Wright, D. J. *et al.* Long read sequencing reveals novel isoforms and insights into splicing regulation during cell state changes. *BMC Genomics* **23**, 42 (2022).
3. Dong, X. *et al.* The long and the short of it: unlocking nanopore long-read RNA sequencing data with short-read differential expression analysis tools. *NAR Genomics and Bioinformatics* **3**, lqab028 (2021).
4. Gao, Y. *et al.* ESPRESSO: Robust discovery and quantification of transcript isoforms from error-prone long-read RNA-seq data. *Sci Adv* **9**, eabq5072 (2023).
5. Petri, A. J. & Sahlin, K. isONform: reference-free transcriptome reconstruction from Oxford Nanopore data. *Bioinformatics* **39**, i222–i231 (2023).
6. Xia, Y. *et al.* TAgET: a toolkit for analyzing full-length transcripts from long-read sequencing. *Nat Commun* **14**, 5935 (2023).
7. Lienhard, M. *et al.* IsoTools: a flexible workflow for long-read transcriptome sequencing analysis. *Bioinformatics* **39**, btad364 (2023).
8. Weber, L. M. *et al.* Essential guidelines for computational method benchmarking. *Genome Biology* **20**, 125 (2019).
9. Cameron, D. L., Di Stefano, L. & Papenfuss, A. T. Comprehensive evaluation and characterisation of short read general-purpose structural variant calling software. *Nat Commun* **10**, 3240 (2019).
10. Ye, S. H., Siddle, K. J., Park, D. J. & Sabeti, P. C. Benchmarking Metagenomics Tools for Taxonomic Classification. *Cell* **178**, 779–794 (2019).
11. Magnitov, M. D., Kuznetsova, V. S., Ulianov, S. V., Razin, S. V. & Tyakht, A. V. Benchmark of software tools for prokaryotic chromosomal interaction domain identification. *Bioinformatics* **36**, 4560–4567 (2020).
12. Xi, N. M. & Li, J. J. Benchmarking Computational Doublet-Detection Methods for Single-Cell RNA Sequencing Data. *Cell Systems* **12**, 176-194.e6 (2021).
13. Vlachos, C. *et al.* Benchmarking software tools for detecting and quantifying selection in evolve and resequencing studies. *Genome Biology* **20**, 169 (2019).
14. Nunn, A., Otto, C., Stadler, P. F. & Langenberger, D. Comprehensive benchmarking of software for mapping whole genome bisulfite data: from read alignment to DNA methylation analysis. *Briefings in Bioinformatics* **22**, bbab021 (2021).
15. Weirather, J. L. *et al.* Comprehensive comparison of Pacific Biosciences and Oxford Nanopore Technologies and their applications to transcriptome analysis. Preprint at <https://doi.org/10.12688/f1000research.10571.2> (2017).

16. Gleeson, J. *et al.* Accurate expression quantification from nanopore direct RNA sequencing with NanoCount. *Nucleic Acids Research* **50**, e19 (2022).
17. Kanitz, A. *et al.* Comparative assessment of methods for the computational inference of transcript isoform abundance from RNA-seq data. *Genome Biol* **16**, 150 (2015).

Reviewer #2 (Remarks to the Author):

The manuscript is now looking really good and the authors are to be congratulated for their hard work to address the various concerns I had about the paper.

I have a few minor comment for the authors to address:

- English has continually improved still needs some editing and use of spell checking, including in figure and sup figure legends and figure axes would be helpful.

- Lines 279-284: The claims in these two sentences appear contradictory as written. Suggest revising to make it clear the second sentence refers to an exception to the result in the 1st sentence.

- Something the authors may want to mention is that the Sup Fig10 Jaccard similarity statistics never seem to show IsoQuant (guided) and Bambu (guided) have high similarity. This is quite surprising as they are the two best performing tools. Do they perform well but produce quite different sets of probably true positive isoforms? The authors may have some thoughts on this.

- I think the paragraph (line 429-446) on TE detection could still be clearer. Which software performed well on this comparison? Nb: If a program identified TEs as expressed, it doesn't mean it is accurate for detecting which TEs are expressed given their sequence similarity.

Sup Fig11B - legend states fig shows Bambu "guided" but I believe it is actually "unguided".

Line 442 - Do you mean "enriched FOR TE regions"? Capitals used for emphasis.

- Line 551-552 - Suggest add qualification "sometimes" or "under some circumstance" to the discussion regarding Flames. Looking at Fig2H etc, it appears that Flames sensitivity decreased in some but not all circumstances as accuracy increased. Nb: This is also consistent with the Flames-only results in FigS13.

- Line 580, given the Stringtie2 results with single-end degradation vs dual end would it be more correct to say "transcription start site (TSS) or transcription termination site (TTS) are matched"? ie: "or" not "and"

- I think both Sup Table S2 and S3 spreadsheets are titled as S2. Please check and revise.

- Line 698 - either a word is missing, or an unnecessary "and" is present.

- Note the paper should cite the LRGASP pre-print:

<https://www.biorxiv.org/content/10.1101/2023.07.25.550582v1.full>

Reviewer #3 (Remarks to the Author):

In the previous round of review, I had 4 major comments (I) the lack of a ground truth (ideally spike-in data or alternatively qPCR data), (ii) missing evaluation of state-of-the-art software, (iii) age of the sequenced data and (iv) influence of parameters.

The authors have made very strong and positive efforts to address these issues and thus, three of the issues are now perfectly addressed. The fourth (age of sequencing data) is not, but I do understand that this is difficult to address (as it is a moving target). Also, one may argue that Nanopore quality will come closer and closer to PacBio quality (and PacBio quality will remain unchanged), and that thus

at some time in the future PacBio results may be used as a proxy for Nanopore.

Overall, the authors' advances now make this a very informative contribution to the field. I now support its publication.

Reviewer #2 (Remarks to the Author):

The manuscript is now looking really good and the authors are to be congratulated for their hard work to address the various concerns I had about the paper.

We thank the reviewer for this positive comment! We sincerely appreciate all the constructive feedback provided by the reviewer, which has greatly contributed to further improving this paper.

I have a few minor comment for the authors to address:

- English has continually improved still needs some editing and use of spell checking, including in figure and sup figure legends and figure axes would be helpful.

We thank the reviewer for raising this concern, and we have completely revised the manuscript as requested.

- Lines 279-284: The claims in these two sentences appear contradictory as written. Suggest revising to make it clear the second sentence refers to an exception to the result in the 1st sentence.

We have revised this part as suggested. (Lines 290–295)

- Something the authors may want to mention is that the Sup Fig10 Jaccard similarity statistics never seem to show IsoQuant (guided) and Bambu (guided) have high similarity. This is quite surprising as they are the two best performing tools. Do they perform well but produce quite different sets of probably true positive isoforms? The authors may have some thoughts on this.

We are grateful for the reviewer's constructive feedback. After further examining the IGV tracks, we found that Bambu (guided) and IsoQuant (guided) indeed appear to detect different sets of true positive isoforms. We think this discrepancy would likely explain their fewer overlapping isoforms despite their commendable performance. We have also added additional discussion on this in our revised manuscript. (Lines 444–448)

- I think the paragraph (line 429-446) on TE detection could still be clearer. Which software performed well on this comparison? Nb: If a program identified TEs as expressed, it doesn't mean it is accurate for detecting which TEs are expressed given their sequence similarity.

We appreciate the reviewer's great advice. From Supplementary Figure 11B it seems FLAIR (guided) and Freddie detected slightly more expressed TEs. But we also agree with what the reviewer mentioned that those TE identifications may not be considered as accurate given that there is a high similarity among TE sequences. We have further revised the paragraph as suggested. (Lines 465–467)

Sup Fig11B - legend states fig shows Bambu “guided” but I believe it is actually “unguided”.

We thank the reviewer for catching this, and we apologize for the confusion. We have revised this part as requested.

Line 442 - Do you mean “enriched FOR TE regions”? Capitals used for emphasis.

We thank the reviewer for raising this question and we have revised it as suggested. (Line 463)

- Line 551-552 - Suggest add qualification “sometimes” or “under some circumstance” to the discussion regarding Flames. Looking at Fig2H etc, it appears that Flames sensitivity decreased in some but not all circumstances as accuracy increased. Nb: This is also consistent with the Flames-only results in FigS13.

We thank the reviewer for noticing this point and we have revised this part as suggested. (Line 578–579)

- Line 580, given the Stringtie2 results with single-end degradation vs dual end would it be more correct to say “transcription start site (TSS) or transcription termination site (TTS) are matched”? ie: “or” not “and”

We appreciate the reviewer’s observation regarding this matter, and we have accordingly revised this section as requested. (Line 606)

- I think both Sup Table S2 and S3 spreadsheets are titled as S2. Please check and revise.

We apologize for this mistake and we have corrected it.

- Line 698 - either a word is missing, or an unnecessary “and” is present.

We apologize for this typo and we have corrected it.

- Note the paper should cite the LRGASP pre-print: <https://www.biorxiv.org/content/10.1101/2023.07.25.550582v1.full>

We have added the citation as suggested.

Reviewer #3 (Remarks to the Author):

In the previous round of review, I had 4 major comments (i) the lack of a ground truth (ideally spike-in data or alternatively qPCR data), (ii) missing evaluation of state-of-the-art software, (iii) age of the sequenced data and (iv) influence of parameters.

The authors have made very strong and positive efforts to address these issues and thus, three of the issues are now perfectly addressed. The fourth (age of sequencing data) is not, but I do understand that this is difficult to address (as it is a moving target). Also, one may argue that Nanopore quality will come closer and closer to PacBio quality (and PacBio quality will remain unchanged), and that thus at some time in the future PacBio results may be used as a proxy for Nanopore.

Overall, the authors' advances now make this a very informative contribution to the field. I now support its publication.

We are grateful to the reviewer for the positive comments! Additionally, we sincerely value the constructive feedback provided by the reviewer, enhancing the paper significantly.